# Is the Last Layer Sufficient for Uncertainty Quantification?

**Joseph Wilson** [1]   **Chris van der Heide** [2]   **Liam Hodgkinson** [3]   **Fred Roosta** [1 4]

## Abstract

Epistemic uncertainty quantification (UQ) for deep neural networks (DNNs) is a requirement for safe adoption of AI in mission-critical settings. Several leading methods for UQ linearize DNNs to form Bayesian Generalized Linear Models (GLMs), where epistemic uncertainty is modeled via the predictive posterior distribution. Linearizing around the parameters of the *final connected layer* of a DNN is a commonly used approximation for reducing the computational burden of such GLMs, though it is often believed to come at the cost of degraded performance. In this work, we compare GLMs arising from full-network and last-layer linearization using both theoretical and empirical approaches. We first employ tools from random matrix theory to conduct a theoretical comparison; this analysis reveals no meaningful improvement in the UQ capabilities of full linearization. Coupled with a large-scale empirical evaluation across a range of modern machine learning tasks, we arrive at the following conclusion: a last-layer approximation yields comparable UQ performance while offering substantially improved computational efficiency.

## 1. Introduction

While the predictive performance of deep neural networks (DNNs) continues to improve (Vaswani et al., 2017; Ray, 2023), a key missing component of modern DNNs is uncertainty quantification (UQ) (Abdar et al., 2021). The lack of ability to accurately quantify how certain a network is about its prediction is currently a roadblock to the use of DNNs

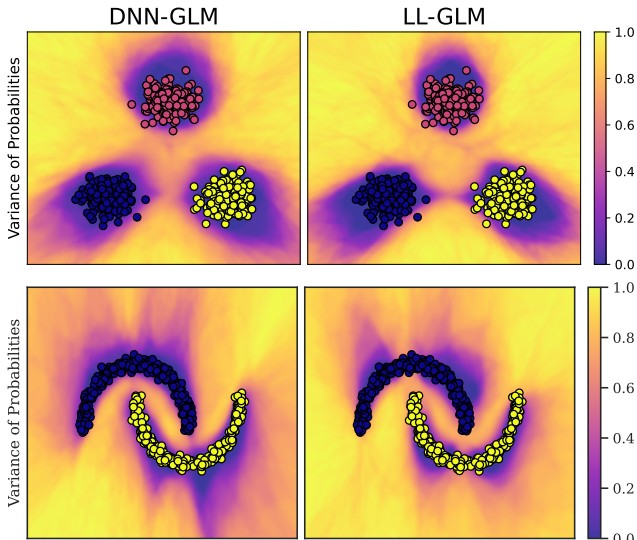

*Figure 1.* Variance of the maximum softmax probability, on the (top) Three Islands dataset and (bottom) Two Moons dataset. We see that a last-layer approximation does not affect quality of UQ. Excerpt from Figure 14a.

in mission-critical settings (Nemani et al., 2023). There has been significant progress in the field of UQ in recent years, mainly in three key areas: conformal predictions (Vovk et al., 2005; Papadopoulos et al., 2002; Lei & Wasserman, 2014), feature-space methods (Altieri et al., 2024; Dadalto et al., 2023; Granese et al., 2021), and Bayesian methods (Neal, 2012). While conformal predictions and feature-space methods display impressive results, Bayesian methods provide a very natural framework for modeling uncertainty, through the lens of the posterior predictive distribution.

There exist many Bayesian methods, such as the Laplace Approximation (MacKay, 1992; Ritter et al., 2018), Variational Inference (Hinton & Van Camp, 1993; Graves, 2011), SWAG (Maddox et al., 2019), MC-Dropout (Gal & Ghahramani, 2016), and Deep Ensembles (DE) (Lakshminarayanan et al., 2017). Unfortunately, these methods often suffer from poor performance and/or extremely high computational load. For example, the popular DE is currently considered the gold-standard for performance; however, it requires a prohibitive computational budget. Attempts to find methods that are competitive with DE in performance, yet require marginal cost, have only been partly successful.

[*]Equal contribution   [1] School of Mathematics and Physics, University of Queensland, Australia [2] Department of Electrical and Electronic Engineering, University of Melbourne, Australia [3] School of Mathematics and Statistics, University of Melbourne, Australia [4] ARC Training Centre for Information Resilience (CIRES), Brisbane, Australia. Correspondence to: Joseph Wilson <joseph.wilson1@uqconnect.edu.au>.

*Proceedings of the 43rd International Conference on Machine Learning*, Seoul, South Korea. PMLR 306, 2026. Copyright 2026 by the author(s).

A recent, promising line of work has focused on first linearizing a DNN around its parameters, and then applying Bayesian methods to the linearized model (Immer et al., 2021b; Maddox et al., 2021; Wilson et al., 2025). This approach improves tractability, while retaining competitive performance. However, an open question remains: *around which subset of parameters should one linearize?* Taking the subset to be parameters in the final connected layer of a DNN (termed a *last-layer* approximation) is a common approach (Snoek et al., 2015; Kristiadi et al., 2020) that *significantly* increases computational efficiency of these Bayesian methods (Daxberger et al., 2021). This approach was shown in Brosse et al. (2020) to perform comparably to a *full Bayesian Neural Network* (BNN), i.e. forming the posterior for the non-linear function. However, compared to linearizing around *all* parameters of the DNN, a last-layer approach is often thought to lead to decreased performance. In this work, we put this assumption to the test.

Under a Bayesian framework, these linearized models amount to Bayesian Generalized Linear Models (GLMs) (Immer et al., 2021b). By the equivalence of Bayesian GLMs in *weight* space to a Gaussian Process (GP) in *function* space (Rasmussen & Williams, 2005), a natural way to compare the quality of these approximations is through the corresponding *Bayes Free Energy* (BFE), i.e. the negative log of the marginal likelihood and a key measure of model quality (Hodgkinson et al., 2023a;b), of the equivalent GP using the induced kernel functions. Specifically, we note that the last-layer approximation amounts to the Conjugate Kernel (CK), while taking the full set of parameters induces the widely-studied Neural Tangent Kernel (NTK). We denote the GLM induced by a last-layer linearization of a DNN as an LL-GLM (last-layer GLM), and the GLM induced by the full linearization of a DNN as a DNN-GLM.

To analyze the BFE for both the CK and NTK, we take a randomly initialized DNN, with mild assumptions on the network components and data distributions. From this, we employ the tools from Random Matrix Theory (RMT) (Couillet & Liao, 2022; Fan & Wang, 2020) to characterize the deterministic limiting BFE, under the double-asymptotic regime where both DNN layer-width and number of training data points tend to infinity proportionally. This regime is appropriate for the large models & datasets used in modern machine learning. Note that DNNs at initialization provide important information about their trained counterparts. Firstly, the NTK at initialization converges to a deterministic limit as DNN width grows infinitely large, and remains constant thereafter during training (Jacot et al., 2018). Secondly, the underlying DNNs converge to a GP in this same limit, with kernel function given by the expected CK (Neal, 2012; Matthews et al., 2018; Lee et al., 2017). Further, randomly initialized DNNs display double-descent (Liao et al., 2020; Mei & Montanari, 2022). Finally, the *strong lottery ticket*

*hypothesis* (Ramanujan et al., 2020) proposes that randomly initialized DNNs contain *sub-networks* that at initialization attain the same accuracy as their trained super-network.

From the limiting BFE, we are able to directly contrast the LL-GLM against the DNN-GLM, by comparing the induced kernel functions. We tie the comparison of these kernels to two current areas of research in DNN theory: *robustness* (insensitivity to input perturbations) and *descent phenomena* (including double- and triple-descent behavior). Strikingly, we identify only a single regime in which the LL-GLM underperforms the DNN-GLM: when the number of data points is *much* larger than the network width. Even then, our experiments suggest this gap may be an artifact of random initialization rather than a fundamental limitation.

> Our theory predicts no intrinsic performance advantage of the DNN-GLM over the LL-GLM.

Motivated by this, we conduct a large-scale empirical investigation of the LL-GLM versus DNN-GLM in the context of UQ. Note that naïvely forming these GLMs is often still prohibitive for modern machine learning tasks, and many further approximations are required (Immer et al., 2021b; Daxberger et al., 2021; Antorán et al., 2022; Ortega et al., 2023; Deng et al., 2022). Motivated by the NUQLS method (Wilson et al., 2025), we provide a simple algorithm to sample from these Bayesian GLMs, that scales to large language models, implemented in a lightweight package, *LinearSampling*. We then compare these GLMs on a series of regression, classification, and language processing tasks. Across all domains, the LL-GLM consistently matches the UQ performance of the DNN-GLM, often rivaling that of the gold-standard DE. Crucially, the LL-GLM achieves this with substantially lower computational and memory costs. This work suggests that epistemic uncertainty in a DNN arises from uncertainty in how the network *features* are used, rather than in the features themselves.

**Contributions**

1. We derive, under mild conditions, the limiting BFE for GPs with randomly initialized CK and NTK, characterized as the unique solution to a system of implicit fixed-point equations.
2. We show theoretically that both kernels exhibit robustness to input perturbations as well as descent phenomena.
3. We demonstrate that the randomly initialized NTK yields superior fit in the highly over-sampled regime, and empirically show that this advantage disappears after training.
4. We conduct a large-scale empirical evaluation of LL-GLMs and DNN-GLMs for UQ, demonstrating equivalent performance across tasks.
5. We release a lightweight, scalable PyTorch package for sampling from these GLMs: *LinearSampling*.

## 2. Background

### 2.1. Bayesian UQ

For the sake of brevity, we do not include a comprehensive review of Bayesian UQ methods; rather, we refer the reader to Wilson et al. (2025) for a description of leading Bayesian UQ methods. Several important works to note are the LLA methods, such as base LLA (Immer et al., 2021b; Daxberger et al., 2021), Sampling-LLA (Antorán et al., 2022), VaLLA (Ortega et al., 2023), and ELLA (Deng et al., 2022). These are the leading methods for forming LL-GLM and DNN-GLM posteriors. However, the difference lies in the approximations used. Specifically, for regression, and moderate model-size, the base LLA method can sample from both the LL-GLM and DNN-GLM posteriors. For classification, a Laplace Approximation results in independent posteriors for each class. Further, when model-size or number of classes is large, LLA requires further approximations, on top of the last-layer approximations, such as KFAC and Diagonal approximations, which hinder performance. Sampling LLA uses the sample-then-optimize framework (Matthews et al., 2017) to be able to sample from the DNN-GLM for larger models, though run-time can be prohibitive (see Ortega et al. (2023) for practical run-times). VaLLA and ELLA employ variational learning and Nystrom (Martinsson & Tropp, 2020) techniques, respectively, to further approximate the DNN-GLM posterior. However, LLA and Sampling-LLA cannot efficiently sample from the posterior of a LL-GLM and DNN-GLM for larger models. Further, for classification, independent posteriors are required. VaLLA and ELLA, while showing good performance, do not sample from the posterior of a LL-GLM or DNN-GLM. Hence, we require novel frameworks to sample from these posteriors for large classification tasks.

**Last-Layer Approximation** Several works in the Bayesian UQ literature discuss the fidelity-cost trade-off of last-layer approximations. In Watson et al. (2021), the authors posit that last-layer features can overfit the feature space, limiting predictive UQ ability. They propose to amend Type-II maximum likelihood overfitting with an augmented marginal likelihood. Further, Calvo-Ordoñez et al. (2026) claim that the last-layer approximation misses "uncertainty induced by earlier layers"; a low-rank transformation between the full Jacobian and the last-layer features is proposed as a new ad-hoc set of features. Interestingly, a direct comparison of LLA (DNN-GLM) vs. last-layer LLA (LL-GLM) on a 1- toy-regression problem is provided in (Ortega et al., 2023), the only direct comparison we are aware of in the literature. Not all works are pessimistic; in Fiedler & Lucia (2023), the authors discuss the "promising compromise" of using Bayesian last-layer methods for DNNs, while Kristiadi et al. (2020) demonstrate that a last-layer approximation "already gives desirable benefits". However,

our work is the first to explicitly compare a last-layer approximation (LL-GLM) with the full-linearization in a large empirical evaluation, across a range of tasks.

### 2.2. DNNs

We primarily consider a supervised-learning problem, given some training data $(\mathbf{x}_i, \mathbf{y}_i)_{i=1}^n \in \mathbb{R}^{d_0} \times \mathbb{R}^c$, where $(\mathbf{x}_i, \mathbf{y}_i) \sim \mathcal{D}$, for some data-generating distribution $\mathcal{D}$. We concatenate the inputs as $\mathbf{X} = [\mathbf{x}_1 | \dots | \mathbf{x}_n] \in \mathbb{R}^{d_0 \times n}$ and outputs as $\mathbf{Y} = [\mathbf{y}_1, \dots, \mathbf{y}_n] \in \mathbb{R}^{c \times n}$. We define a fully-connected, feedforward, neural network $\mathbf{f}_{\boldsymbol{\theta}} : \mathbb{R}^{d_0} \to \mathbb{R}^c$, with NTK scaling Jacot et al. (2018) and $L$-layers as

$$\mathbf{f}_{\boldsymbol{\theta}}(\mathbf{x}) = \mathbf{w}^T \frac{1}{\sqrt{d_L}} \sigma \left( \cdots \sigma \left( \mathbf{W}_2 \frac{1}{\sqrt{d_1}} \sigma \left( \mathbf{W}_1 \mathbf{x} \right) \cdots \right) \right).$$

Here, $\sigma : \mathbb{R} \to \mathbb{R}$ is the activation function, $\boldsymbol{\theta} = \text{vec}\{\mathbf{W}_1, \dots, \mathbf{W}_L, \mathbf{w}\}$ is the vectorized collection of all layer-weights, where the weights are given by $\mathbf{W}_l \in \mathbb{R}^{d_l \times d_{l-1}}$ for $1 \le l \le L$, and $\mathbf{w} \in \mathbb{R}^{d_L \times c}$, with $d_1, \dots, d_L$ the number of neurons in each hidden layer. For some input vector $\mathbf{x} \in \mathbb{R}^{d_0}$, we denote the post-activations vectors as $\mathbf{x}_l = \sigma \left( W_l \mathbf{x}_{l-1} \right) / \sqrt{d_l}$, for $1 \le l \le L$, and $\mathbf{X}_l = [\mathbf{x}_{1,l}, \dots \mathbf{x}_{n,l}]$. Let $\mathbf{J}(\boldsymbol{\theta}, \mathbf{x}) \in \mathbb{R}^{c \times p}$ be the Jacobian of the network with respect to $\boldsymbol{\theta}$, where $p = \dim(\boldsymbol{\theta})$. Let $\mathbf{J}(\boldsymbol{\theta}, \mathbf{X}) \in \mathbb{R}^{nc \times p}$ be the concatenation of Jacobian matrices. If $\mathbf{f}$ is scalar-valued, we write the function as $f$.

We define the **Conjugate Kernel** function as

$$\kappa^{\text{CK}}(\mathbf{x}_i, \mathbf{x}_j) = \mathbf{x}_{i,L}^T \mathbf{x}_{j,L}.$$

The Gram matrix of the CK at the final layer is thus given by $\mathbf{K}_{\mathbf{X}}^{\text{CK}} = \mathbf{X}_L^T \mathbf{X}_L \in \mathbb{R}^{nc \times nc}$.

For a DNN undergoing gradient flow, the functional dynamics of the DNN undergo kernel gradient flow, where the kernel is the **Neural Tangent Kernel**. As shown in Jacot et al. (2018), the NTK becomes deterministic and constant during gradient flow as the width of the DNN approaches infinity. In this regime, the NTK is termed the analytic NTK. The finite-width NTK, or the *empirical* NTK, has garnered significant attention in recent years (Fort et al., 2020; Arora et al., 2019), with roles in generalization (Hodgkinson et al., 2023b), training (Du et al., 2019), robustness (Bombari et al., 2023), and uncertainty quantification (Immer et al., 2021b; Wilson et al., 2025). The NTK function is defined as

$$\kappa^{\text{NTK}}(\mathbf{x}_i, \mathbf{x}_j) = \mathbf{J}(\boldsymbol{\theta}, \mathbf{x}_i) \mathbf{J}(\boldsymbol{\theta}, \mathbf{x}_j)^T.$$

The Gram matrix of the NTK is given by $\mathbf{K}_{\mathbf{X}}^{\text{NTK}} = \mathbf{J}(\boldsymbol{\theta}, \mathbf{X}) \mathbf{J}(\boldsymbol{\theta}, \mathbf{X})^T \in \mathbb{R}^{nc \times nc}$.

**Kernel Machines** Both the CK and NTK have been used in kernel machines. Specifically, Arora et al. (2019) employed an NTK Support-Vector Machine for small-data

tasks, in which it outperformed competitors. Interestingly, Qadeer et al. (2023) compared the predictive performance of the CK and the NTK kernel machines, on a series of tasks. They find that in these experiments, the CK often performs as well as, or better, than the NTK.

### 2.3. Bayesian Models

**Gaussian Process** A GP is a distribution over functions that is determined by its mean and kernel functions, $m : \mathbb{R}^{d_0} \to \mathbb{R}$ and $\kappa : \mathbb{R}^{d_0} \times \mathbb{R}^{d_0} \to \mathbb{R}$. We take a Gaussian likelihood $y_i = f(\mathbf{x}_i) + \epsilon_i$, $\epsilon_i \sim \mathcal{N}(0, \tau)$, for noise $\tau > 0$, and prior $f \sim \mathcal{GP}(m, \lambda^{-1}k)$ for some regularization parameter $\lambda > 0$. The Gram matrix $\mathbf{K_X} \in \mathbb{R}^{n \times n}$ has elements $\mathbf{K_X}^{ij} = \kappa(\mathbf{x}_i, \mathbf{x}_j)$. We also define $\mathbf{K_{x^*}} = [\kappa(\mathbf{x}^*, \mathbf{x}_1), \dots, \kappa(\mathbf{x}^*, \mathbf{x}_n)] \in \mathbb{R}^n$, $\boldsymbol{\kappa}_{\mathbf{x}^*} = \kappa(\mathbf{x}^*, \mathbf{x}^*) \in \mathbb{R}$ and $m(\mathbf{X}) = [m(\mathbf{x}_1), \dots, m(\mathbf{x}_n)] \in \mathbb{R}^n$. Under these settings, the posterior predictive distribution of $f$ for some test point $\mathbf{x}^*$, given the training data $\mathcal{D} = \{\mathbf{X}, \mathbf{Y}\}$, is

$$f(\mathbf{x}^*)|\mathcal{D} \sim \mathcal{N}\bigg(\mathbf{K_{x^*}}^T (\mathbf{K_X} + \lambda\tau\mathrm{I})^{-1} (\mathbf{Y} - m(\mathbf{X})) + m(\mathbf{x}^*),$$
$$\lambda^{-1}(\boldsymbol{\kappa}_{\mathbf{x}^*} - \mathbf{K_{x^*}}^T (\mathbf{K_X} + \lambda\tau\mathrm{I})^{-1}\mathbf{K_{x^*}})\bigg). \quad (1)$$

**GLMs** Consider the linearization of $\mathbf{f}_{\hat{\boldsymbol{\theta}}}$ around parameters $\hat{\boldsymbol{\theta}}$ with respect to a subset $\boldsymbol{\theta}_S \subseteq \boldsymbol{\theta}$, $\boldsymbol{\theta}_S \in \mathbb{R}^{p_S}$,

$$\tilde{\mathbf{f}}_{\hat{\boldsymbol{\theta}}}(\boldsymbol{\theta}_S, \mathbf{x}) = \mathbf{f}_{\hat{\boldsymbol{\theta}}}(\mathbf{x}) + \mathbf{J}_S(\hat{\boldsymbol{\theta}}_S, \mathbf{x})\left(\boldsymbol{\theta}_S - \hat{\boldsymbol{\theta}}_S\right), \quad (2)$$

where $\mathbf{J}_S(\boldsymbol{\theta}, \mathbf{x}) \in \mathbb{R}^{c \times p_S}$ is the Jacobian of $\mathbf{f}_{\boldsymbol{\theta}}(\mathbf{x})$ w.r.t the parameters $\boldsymbol{\theta}_S$, evaluated at input $\mathbf{x}$ and parameters $\boldsymbol{\theta}$, and $\hat{\boldsymbol{\theta}}_S \subseteq \hat{\boldsymbol{\theta}}$ contains the corresponding subset of parameters in $\hat{\boldsymbol{\theta}}$ as $\boldsymbol{\theta}_S$. If we place a prior $\boldsymbol{\theta}_S \sim p(\boldsymbol{\theta}_S)$, then $\tilde{\mathbf{f}}_{\hat{\boldsymbol{\theta}}}(\boldsymbol{\theta}_S, \mathbf{x})$ is a GLM. By the equivalence of Bayesian GLMs in *weight* space to a GP in *function* space (Rasmussen & Williams, 2005), if we set $p(\boldsymbol{\theta}_S) = \mathcal{N}(0, \lambda^{-1}\mathrm{I})$, then the *predictive posterior distribution* of $\tilde{\mathbf{f}}_{\hat{\boldsymbol{\theta}}}(\boldsymbol{\theta}_S, \mathbf{x})$ is equal to (1), for mean function $m(x) = \mathbf{f}_{\hat{\boldsymbol{\theta}}}(\mathbf{x})$ and kernel function $\kappa(x, x') = \mathbf{J}_S(\hat{\boldsymbol{\theta}}_S, \mathbf{x})\mathbf{J}_S(\hat{\boldsymbol{\theta}}_S, \mathbf{x}')^T$. Hence, if $\boldsymbol{\theta}_S = \boldsymbol{\theta}$, $\kappa = \kappa^{\mathrm{NTK}}$. If we let $\boldsymbol{\theta}_S = \boldsymbol{\theta}_L$, the parameters in the final-connected layer[1], $\mathbf{J}_S(\hat{\boldsymbol{\theta}}_S, \mathbf{x}) = \mathbf{x}_{i,L}^T$, and hence $\kappa = \kappa^{\mathrm{CK}}$.

**BFE** The marginal likelihood, or model evidence, is the normalizing constant of the posterior distribution $\mathcal{Z}_n^{\tau,\lambda} = \int_f p(\mathbf{Y}|f, \mathbf{X})\, p(f)\, df$. The **Bayes Free Energy (BFE)** is defined as $\mathcal{F}_n^{\tau,\lambda} := -\log \mathcal{Z}_n^{\tau,\lambda}$, and under (1) is

$$\mathcal{F}_n^{\tau,\lambda} = \frac{1}{2}\lambda\,(\mathbf{Y} - m(\mathbf{X}))^T (\mathbf{K_X} + \lambda\tau\mathrm{I})^{-1} (\mathbf{Y} - m(\mathbf{X}))$$
$$+ \frac{1}{2}\log\det(\mathbf{K_X} + \lambda\tau\mathrm{I}) - \frac{n}{2}\log\left(\frac{\lambda}{2\pi}\right).$$

---

[1] $\boldsymbol{\theta}_L = \mathbf{w}$ in the notation of Section 2.

The marginal likelihood is the canonical measure of model fit for Bayesian systems. It is the probability of generating the data under the prior choice; for GPs, the prior choice corresponds to the choice in kernel, for a fixed mean function. It has been linked to generalization ability of model classes (Hodgkinson et al., 2023b). Interestingly, exhaustive leave-$k$-out cross-validation using the predictive log-likelihood $p(D_\mathrm{k} \mid D_\mathrm{n\text{-}k})$ under the posterior distribution as the score is equivalent to the marginal likelihood (Fong & Holmes, 2020). Further, the choice of prior that maximizes the marginal likelihood is equivalent to the prior choice that minimizes the celebrated PAC-Bayes bound (Germain et al., 2016; Hodgkinson et al., 2023c). For GPs, selection of kernel hyper-parameters by maximization of the marginal likelihood (a procedure called *empirical Bayes*) generally gives impressive predictive performance (Krivoruchko & Gribov, 2019). Hence, the marginal likelihood, or equivalently the BFE, is an effective quantity to consider for contrasting the modeling performance of different kernel functions. There are also explicit connections between BFE and UQ ability of a Bayesian model; see Appendix B. The limiting BFE[2] was characterized in Hodgkinson et al. (2023a) for inner-product and radial-basis kernels.

### 2.4. Robustness

A *robust* function is a function that has some lack of sensitivity to input manipulation, i.e. an adversarial attack (Carlini & Wagner, 2017). While previous theoretical works have attempted to quantify robustness through Lipschitzness of the function (Bubeck & Sellke, 2021; Bombari et al., 2023), we do not consider this the appropriate metric. Works from the benign overfitting literature (Bartlett et al., 2021; Haas et al., 2023) have demonstrated that successful, over-parameterized functions that interpolate the training data can be decomposed into two additive components; a *smooth* component, which maps the underlying data curve, and is responsible for generalization, and a *spike* component that deviates from the *smooth* component at training points in order to interpolate, often using near-discontinuous deviations with unbounded derivatives. Hence, these high-performing functions may have large Lipschitz constants; minimizing the Lipschitz constants of these functions may be inappropriate. Probabilistic interpretations of robustness also make use of the Lipschitz-ness of the function (Mangal et al., 2019).

As discussed in Section 2.3, the BFE is a key quantity for measuring the probabilistic performance of a kernel function for a given data distribution. Minimizing the BFE is equal to finding the kernel function that gives the maximum likelihood of generating the given dataset[3]. This will lead to both good predictive performance and good UQ. Conse-

---

[2] Under the double-asymptotic limit.
[3] This is true under a zero-mean prior.

quently, inputs that deviate from the training distribution induce higher epistemic uncertainty, enabling detection of adversarial or out-of-distribution examples.

> We quantify robustness of a **kernel function** by BFE.

We emphasize that we are considering the robustness of a kernel function that induces a distribution over functions, rather than the robustness of a predictive function.

### 2.5. Descent Curves

A related phenomena in machine learning is the double-descent curve (Belkin et al., 2019). This curve states that contrary to the classical bias-variance trade-off, heavily over-parameterized models may generalize better than under-parameterized models. This has been proved for a variety of models, such as linear regression (Bartlett et al., 2020), random trees (Belkin et al., 2019), random features (Liao et al., 2020), etc.. Importantly, it has been empirically observed in large-scale neural networks (Nakkiran et al., 2021). Extension to this have been observed, such as the triple-descent curve (Adlam & Pennington, 2020) for NTK machines. While double-descent is characterized by a local-maxima around the point of interpolation, $(p = n)$, we consider a generalization of this: *descent* curves. These are curves for which a generalization metric, here BFE, decreases in some manner with respect to number of parameters.

## 3. Theory

To contrast the performance of LL-GLM and DNN-GLM for UQ, we compare the BFE for GPs that employ the induced kernel functions of these two approaches, i.e. the CK and the NTK. To aid the tractability of an analysis, we consider the network in Section 2.2, for scalar outputs $(c = 1)$, randomly initialized. Consider a zero-mean prior $f \sim \mathcal{GP}(0, \lambda^{-1}\kappa)$ for some regularization parameter $\lambda > 0$, where $\kappa \in \{\kappa^{\mathrm{CK}}, \kappa^{\mathrm{NTK}}\}$, and $\mathbf{K_X} \in \{\mathbf{K_X^{CK}}, \mathbf{K_X^{NTK}}\}$. The limiting BFE is given as

$$\lim_{n\to\infty} \frac{1}{n}\mathbb{E}\mathcal{F}_n^{\tau,\lambda} = \lim_{n\to\infty} \left[\frac{\lambda}{2n}\mathbb{E}\mathbf{Y}^T(\mathbf{K_X} + \lambda\tau\mathbf{I})^{-1}\mathbf{Y} \quad (3)\right.$$
$$\left. + \frac{1}{2n}\mathbb{E}\log\det(\mathbf{K_X} + \lambda\tau\mathbf{I}) - \frac{1}{2}\log\left(\frac{\lambda}{2\pi}\right)\right].$$

We consider the limit $n, d_0, d_1, \ldots, d_L \to \infty$ such that $\lim n/d_l = \gamma_l$, for $\gamma_l \in (0, \infty)$, and $l = 0, 1, \ldots, L$. We define $\gamma = (\gamma_0, \ldots, \gamma_L)$. Assumptions underlying our theoretical analysis are given in Appendix A (Assumption 1). Note assumptions on the network are mild; the constraint on the first and second moments of the nonlinearity function can be achieved through shifting and scaling. Assump-

tions on input data covers inputs $\mathbf{X} = [\mathbf{x}_1, \ldots, \mathbf{x}_n]$, where $\mathbf{x}_i = \mathbf{z}_i/\sqrt{d_0}$, $\mathbf{z}_i \sim \mathcal{N}(\mathbf{0}, \mathbf{I})$, and $\mu_0$ corresponds to the Marchenko-Pastur (Marčenko & Pastur, 1967) distribution. Further, we note that our results can be extended to a non-zero mean prior by shifting our targets as $\tilde{\mathbf{Y}} = \mathbf{Y} - m(\mathbf{X})$, where $m(\mathbf{X}) = [m(\mathbf{x}_1)|\ldots|m(\mathbf{x}_n)] \in \mathbb{R}^{d_0 \times n}$.

**Theorem 1.** *Under Assumption 1,*

$$n^{-1}\mathbb{E}\mathcal{F}_n^{\tau,\lambda} \to \mathcal{F}_\infty^{\tau,\lambda},$$

*where $\mathcal{F}_\infty^{\tau,\lambda}$ is well-defined and is the unique solution to a system of fixed-point equations (see (General) in Appendix A.3).*

*If the input $\mathbf{X} = [\mathbf{x}_1, \ldots, \mathbf{x}_n]$ with $\mathbf{x}_i = \mathbf{z}_i/\sqrt{d_0}$ and $\mathbf{z}_i \sim \mathcal{N}(\mathbf{0}, \mathbf{I})$, then $\mathcal{F}_\infty^{\tau,\lambda} > (1 + \ln 2\pi)/2$, and*

$$\lim_{\gamma\to\gamma^*} \mathcal{F}_\infty^{\tau,\lambda^*} = \frac{1 + \ln 2\pi}{2}$$

***only** under the following conditions:*

- *For the **CK** kernel: $\gamma^* = (0, \ldots, 0)$, $\tau < 1$, and $\lambda^* = 1/(1 - \tau)$.*
- *For the **NTK** kernel, either*
  - *$\gamma^* = (0, \ldots, 0)$, $\tau < 1$, and $\lambda^* = \sum_{i=0}^L a_\sigma^i/(1 - \tau)$, or*
  - *$\gamma^* = (\infty, \ldots, \infty)$, $\tau < 1$, and $\lambda^* = r_+/(1 - \tau)$.*

*Further, for the **CK** kernel,*

$$\lim_{\tau\to 0} \lim_{\gamma\to\infty} \mathcal{F}_\infty^{\tau,\lambda} = \infty.$$

The form of $\mathcal{F}_\infty^\tau$ is given in (General) for inputs according to Assumption 1, and in (MP) for inputs $\mathbf{X} = [\mathbf{x}_1, \ldots, \mathbf{x}_n]$, where $\mathbf{x}_i = \mathbf{z}_i/\sqrt{d_0}$, $\mathbf{z}_i \sim \mathcal{N}(\mathbf{0}, \mathbf{I})$. The constants in Theorem 1 are defined in (6) in Appendix A.1. For the proof of Theorem 1, and empirical validation of convergence, see Appendix A. We now discuss consequences of Theorem 1, as well as further results. All figures in Section 3 and Appendix A can be reproduced with the following code.

### 3.1. Robustness

From Theorem 1, we see that as the network widths grow large proportionally to training data (i.e. parameters tend to infinity), and for noise $\tau < 1$, both the CK and the NTK approach the lower-bound on the BFE for appropriate regularization. For the CK, this result differs from the result given in Bombari et al. (2023), i.e. CK models can never be robust. This is due to the differing definition of robustness, that is, robustness of the kernel function as measured by the induced UQ ability (corresponding to BFE), rather than robustness of the mean predictor. Further, as training points grow large proportionally to width, the NTK can also approach the lower-bound on BFE for appropriate regularization. Hence, we diverge from the "universal

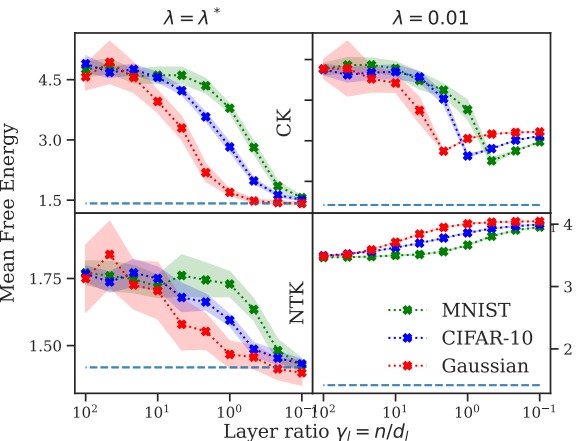

*Figure 2.* BFE as a function of $\gamma$ for the (top) CK and (bottom) NTK. Top left has $\lambda^* = 1/(1-\tau)$, bottom left has $\lambda^* = \sum_{i=0}^{L} a_\sigma^i/(1-\tau)$, and the right plots have $\lambda = 0.01$. The dotted blue line is the minimum BFE, $(1 + \ln 2\pi)/2$. We observe that at $\lambda*$, strong descent occurs.

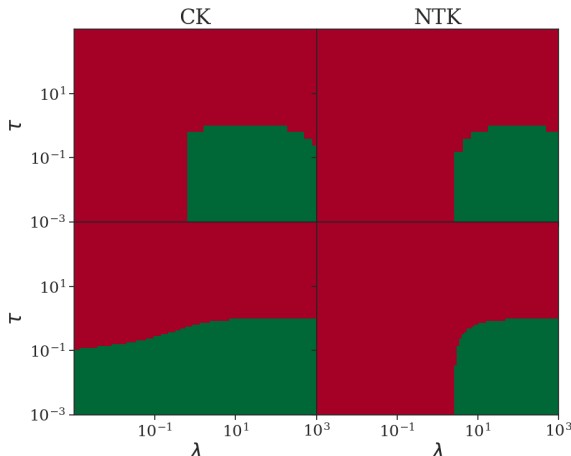

*Figure 3.* Heatmap of when (top) strong and (bottom) weak descent occurs for the CK and NTK, for varying values of $\tau, \lambda$. A green coloring shows a positive for descent occurring, while red indicates the negative.

law of robustness" for mean predictors given in Bubeck & Sellke (2021), which required functions to be sufficiently over-parameterized to be robust: kernel functions do not need to arise from over-parameterized DNNs to be robust. Note that the CK does not approach this lower-bound for $n/d_l \to \infty$, for any regularizer. We will discuss this point in a later section.

### 3.2. Descent

We first consider *strong descent*, where BFE reaches its minimum as the width of the DNN grows large:

**Definition 1** (Strong Descent). *Let $\tau = \tau(\boldsymbol{\gamma})$ and $\lambda = \lambda(\boldsymbol{\gamma})$ be curves parameterized by the layer ratios. Strong descent occurs for the BFE $\mathcal{F}_\infty^{\tau,\lambda}$ when*

$$\min_{\boldsymbol{\gamma}} \mathcal{F}_\infty^{\tau(\boldsymbol{\gamma}),\lambda(\boldsymbol{\gamma})} = \lim_{\boldsymbol{\gamma}\to 0} \mathcal{F}_\infty^{\tau(\boldsymbol{\gamma}),\lambda(\boldsymbol{\gamma})}.$$

**Theorem 2** (Strong Descent). *For $\tau < 1$, $\mathcal{F}_\infty^{\lambda,\tau}$ will undergo **strong descent** if $\lambda(\boldsymbol{\gamma})$ is a curve parameterized by $\boldsymbol{\gamma}$ such that for the CK, $\lim_{\boldsymbol{\gamma}\to 0} \lambda(\boldsymbol{\gamma}) = 1/(1-\tau)$, and for the NTK, $\lim_{\boldsymbol{\gamma}\to 0} \lambda(\boldsymbol{\gamma}) = \sum_{i=0}^{L} a_\sigma^i/(1-\tau)$ and $\lim_{\boldsymbol{\gamma}\to\infty} \lambda(\boldsymbol{\gamma}) \neq r_+/(1-\tau)$. For $\tau \geq 1$, $\mathcal{F}_\infty^{\lambda,\tau}$ will not undergo strong descent, for neither the CK nor the NTK.*

Hence, strong descent will occur for the both the CK and the NTK, for correctly chosen parameters. We provide empirical validation in Figure 2, for both Gaussian data, and for MNIST, CIFAR-10, after normalization and whitening.

We can also define a notion of *weak descent*, where we simply require that complete over-parameterization gives lower BFE than complete under-parameterization:

**Definition 2** (Weak Descent). *Let $\tau = \tau(\gamma_0,\ldots,\gamma_L) = \tau(\boldsymbol{\gamma})$ and $\lambda = \lambda(\gamma_0,\ldots,\gamma_L) = \lambda(\boldsymbol{\gamma})$ be curves parameterized by the layer ratios. Weak descent occurs for the BFE $\mathcal{F}_\infty^{\tau,\lambda}$ when*

$$\lim_{\boldsymbol{\gamma}\to 0} \mathcal{F}_\infty^{\tau(\boldsymbol{\gamma}),\lambda(\boldsymbol{\gamma})} < \lim_{\boldsymbol{\gamma}\to\infty} \mathcal{F}_\infty^{\tau(\boldsymbol{\gamma}),\lambda(\boldsymbol{\gamma})}.$$

We can show that weak descent occurs for a greater subset of parameters $\tau, \lambda$ for the CK, than for the NTK:

**Lemma 1** (Weak Descent). *For $K \in \{CK, NTK\}$, let $\bar{\lambda}(\tau)_K := \{\lambda \mid \lim_{\boldsymbol{\gamma}\to 0} \mathcal{F}_\infty^{\tau,\lambda} < \lim_{\boldsymbol{\gamma}\to\infty} \mathcal{F}_\infty^{\tau,\lambda}\}$. Then, $\bar{\lambda}(\tau)_{NTK} \subset \bar{\lambda}(\tau)_{CK}$.*

We empirically validate this by plotting a heatmap of parameters for which the CK and NTK display *weak* and *strong descent*, in Figure 3. We see that the area of parameters for the CK is larger than for the NTK, for both descent types.

### 3.3. Under-Parameterized Regime

A notable component of Theorem 1 is that as $\boldsymbol{\gamma} \to \infty$, then for certain $\tau, \lambda$, the BFE for the NTK reaches the lower-bound. In comparison, the CK never reaches this lower bound as $\boldsymbol{\gamma} \to \infty$, and in fact diverges in BFE as $\tau \to 0$. Hence, it seems that at initialization, the NTK is expressive enough to model proportionally infinite training-data, while the CK is not. This is an important use case for the NTK, as small-parameter models are much cheaper to compute. This regime is also common in practice; note that ResNet50 has a final connected layer of 2048 parameters, with 1.28 million training points for ImageNet ($\gamma_L \approx 626$).

However, we now empirically demonstrate that this inability

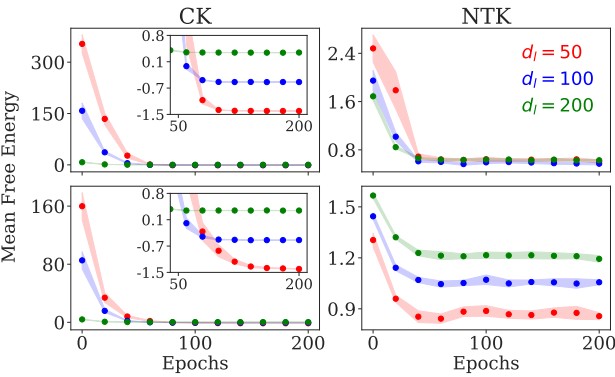

*Figure 4.* BFE as a function of epochs of training, for (top) Gaussian data with $\lambda = \lambda^*$ and (bottom) a teacher network $y = \sin(w^T x)$ with $\lambda = 1$, and $\tau = 10^{-3}$.

of the CK to model noiseless data in this regime is an artifact of initialized networks. Specifically, we generate training data, and then train an MLP to minimize the empirical risk using a squared-error loss. We form the CK and NTK from this trained MLP. Note that once the MLP has been trained, the mean function is $m(x) = f_\theta(x)$. We plot the BFE of the subsequent GP in Figure 4. We see that during training, the free energy for both kernels decreases. Incredibly, after training, the free energy for the CK is *lower* than that of the NTK, though it was initially much larger. Figure 8 and Figure 9 shows that this arises due to the data-fit term tending to zero, with the log-determinant remaining fixed at a lower value than that of the NTK. See Figure 11 for results with MNIST & CIFAR-10.

### 3.4. Connection to Trained Networks

While our results are derived for a randomly-initialized network, we can in certain instances make direct connection to a trained network. By Wang et al. (2023, Corollary 5.3), for a 2-layer DNN undergoing gradient descent for a student-teacher dataset, with a sufficiently small step-size, the spectrum of the kernel matrices $\mathbf{K}^{\text{CK}}_{\mathbf{X},t}$, $\mathbf{K}^{\text{NTK}}_{\mathbf{X},t}$ at gradient step $t$ converge to the spectrum of $\mathbf{K}^{\text{CK}}_{\mathbf{X},0}$, $\mathbf{K}^{\text{NTK}}_{\mathbf{X},0}$ (i.e. kernels at initialization) under the double-asymptotic regime we consider in our work. We can then consider the BFE for the trained kernel matrices on a down-stream task, i.e. the data setting we currently consider in Assumption 1. In this setting, as the spectral distributions after training are equivalent to those at initialization in the limiting regime, almost surely, our theoretical results would remain unchanged. This theory does not cover SGD/Adam. However, we can extend the result in Figure 4 to UCI regression datasets, for networks trained with Adam for 1500 epochs with a learning rate of 0.01, with $\lambda = 1, \tau = 0.01$. We see in Figure 12 that for each dataset, the free energy for the CK eventually decreases below the free energy for the NTK, during training.

## 4. Practical Consideration

To compare LL-GLM versus DNN-GLM on a series of machine learning tasks, we provide a lightweight schema for sampling from such GLMs. Consider a DNN of arbitrary structure, with output dimension $c \in \mathbb{N}_{\geq 0}$. We take the training dataset $(\mathbf{X}, \mathbf{Y}) \sim \mathcal{D}$, where we no longer assume $\mathbf{X}, \mathbf{Y}$ are Gaussian. Consider the trained parameters $\hat{\boldsymbol{\theta}}$,

$$\hat{\boldsymbol{\theta}} = \arg \min_{\boldsymbol{\theta}} \left\{ \sum_{i=1}^{n} \ell(\mathbf{f}_{\boldsymbol{\theta}}(\mathbf{x}_i), \mathbf{y}_i) + \mathcal{R}(\boldsymbol{\theta}) \right\},$$

for some loss-function $\ell : \mathbb{R}^c \times \mathbb{R}^c \to \mathbb{R}_{\geq 0}$, and regularizer $\mathcal{R} : \mathbb{R}^p \to \mathbb{R}_{\geq 0}$. Employing the linearized function in (2), we define two optimization problems:

$$\mathcal{L}_{\text{R}} : \min_{\boldsymbol{\theta}_S} \sum_{i,j=1}^{n,c} \left( \tilde{\mathbf{f}}_{\hat{\boldsymbol{\theta}}}(\boldsymbol{\theta}_S, \mathbf{x}_i)_j - \mathbf{y}_{i,j} \right)^2 \quad \text{(Regression)}$$

$$\mathcal{L}_{\text{C}} : \min_{\boldsymbol{\theta}_S} \sum_{i,j=1}^{n,c} \left( \tilde{\mathbf{f}}_{\hat{\boldsymbol{\theta}}}(\boldsymbol{\theta}_S, \mathbf{x}_i)_j - \mathbf{f}_{\hat{\boldsymbol{\theta}}}(\mathbf{x}_i)_j \right)^2 \quad \text{(Classification)}$$

We now present the *LinearSampling* framework:

---
**Algorithm 1** *LinearSampling*
---
**Input:** number of samples $T$, weights $\hat{\boldsymbol{\theta}}$, parameters $\boldsymbol{\theta}_S$, optimization problem $\mathcal{L}$, scale $\eta^2$.

**for** $t = 1$ **to** $T$ **do**

  Initialize $\boldsymbol{\theta}^0_{S,t} \leftarrow \mathcal{N}(\hat{\boldsymbol{\theta}}_S, \eta^2 \mathbf{I})$

  $\boldsymbol{\theta}^*_{S,t} \leftarrow$ Run (S)GD, initialized at $\boldsymbol{\theta}^0_{S,t}$, to

    (approximately) solve $\mathcal{L}$ and obtain $\boldsymbol{\theta}^*_{S,t}$

**end for**

**return** $\{\tilde{\mathbf{f}}_{\hat{\boldsymbol{\theta}}}(\boldsymbol{\theta}^*_{S,t}, .)\}_{t=1}^T$

---

**Lemma 2** (GLM Sampling). *If $p_S < nc$, and $\mathbf{J}_S(\hat{\boldsymbol{\theta}}, \mathbf{X}) \in \mathbb{R}^{nc \times p_S}$ is rank-deficient, or if $p_S > nc$, then Algorithm 1 for $\boldsymbol{\theta}_S$, $\mathcal{L} \in \{\mathcal{L}_R, \mathcal{L}_C\}$ returns samples from (1) for regression or classification respectively, for the kernel induced by the feature map $\mathbf{J}_S(\hat{\boldsymbol{\theta}}, \mathbf{x})$, with regularizer $\lambda = \eta^{-2}, \tau \to 0$.*

To sample from the predictive posterior of a DNN-GLM, we hence employ Algorithm 1 with $\boldsymbol{\theta}_S = \boldsymbol{\theta}$ (the regression case was shown in Wilson et al. (2025)). For an LL-GLM, we use $\boldsymbol{\theta}_S = \boldsymbol{\theta}_L$. The proof is given in Appendix D. Note that for many modern datasets and models, $\mathbf{J}_L(\hat{\boldsymbol{\theta}}, \mathbf{X})$ is *nearly* rank-deficient, with many small singular-values; in this case, early stopping of the (S)GD scheme in Algorithm 1 will return samples from (1), for an equivalent Gram matrix with some small singular-values truncated to zero. See Appendix E for an in-depth discussion of this. Further, note that we are sampling from a GLM with noise $\tau \to 0$; for modern models and datasets, such a noise level may be more appropriate (Hodgkinson et al., 2023b). We provide a lightweight *PyTorch* implementation, *LinearSampling*.

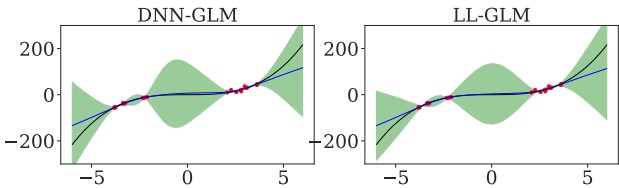

*Figure 5.* Toy regression problem. Here, the red points are training points, the black line is the underlying curve, the blue line is the mean prediction, and the green shading represents $\pm 2\sigma(x)$, where $\sigma(x)$ is the computed variance.

# 5. Empirical Evaluation

Employing the sampling scheme from Section 4, we now provide a large-scale comparison of LL-GLM versus DNN-GLM on a series of machine learning tasks, on regression (Section 5.1), classification (Section 5.2), and language modeling (Section 5.3) tasks. Code to reproduce the figures in this section, Appendix E and Appendix G can be found here. For implementation details, see Appendix H.

## 5.1. Regression

We first obtain a pictorial comparison on a toy regression problem. We train a small MLP on a 1D problem, where training points are taken from the curve $y = x^3 + \epsilon$, where $\epsilon \sim \mathcal{N}(0, 9)$ and $x \in [-4, 2] \cup [2, 4]$. We plot the mean prediction and confidence intervals for DNN-GLM and LL-GLM in Figure 5. Both methods capture the underlying curve within the confidence interval, and show growing uncertainty with distance from training points.

We also compare LL-GLM vs DNN-GLM on a range of UCI regression problems. We provide the values in Table 1. Surprisingly, we see no benefit gained from using the full DNN-GLM over the LL-GLM. In fact, UQ performance from the LL-GLM is sometimes *better* than from the the full DNN-GLM, as measured by Expected Calibration Error (ECE) and Gaussian Negative-Log Likelihood (NLL).

## 5.2. Classification

It was argued in Liu et al. (2020) that uncertainty measures should be 'distance-aware' for classification. To this end, we compare the LL-GLM and DNN-GLM on a toy example, where we form $C = 3$ clusters of training points in $\mathbb{R}^2$, a dataset we term 'Three Islands'. We train a small MLP to classify the points, and perform posterior inference using the GLMs. We plot the predictive variance in Figure 1. We see that both GLMs possess this 'distance-aware' property. See Appendix G.1 for comparison against competing methods.

Evaluation of UQ ability for large-scale classification tasks is notoriously difficult; see Wilson et al. (2025) for an in-depth discussion. To summarize, common metrics such

*Table 1.* Comparison of GLMs on a series of UCI regression tasks. Entries where DNN-GLM outperforms LL-GLM are highlighted in magenta, and teal for vice-versa. Performance of the LL-GLM is as good as that of the DNN-GLM.

| Dataset | Method | RMSE ↓ | NLL ↓ | ECE ↓ |
|---|---|---|---|---|
| **Energy** | DNN-GLM | $0.047\pm_{0.006}$ | $-2.400\pm_{0.209}$ | $0.002\pm_{0.002}$ |
| | LL-GLM | $0.042\pm_{0.005}$ | $-2.560\pm_{0.186}$ | $0.002\pm_{0.002}$ |
| **Kin8nm** | DNN-GLM | $0.252\pm_{0.005}$ | $-0.796\pm_{0.025}$ | $0.000\pm_{0.000}$ |
| | LL-GLM | $0.247\pm_{0.005}$ | $-0.832\pm_{0.017}$ | $0.000\pm_{0.000}$ |
| **Protein** | DNN-GLM | $0.623\pm_{0.005}$ | $0.209\pm_{0.047}$ | $0.002\pm_{0.000}$ |
| | LL-GLM | $0.630\pm_{0.011}$ | $0.250\pm_{0.025}$ | $0.002\pm_{0.000}$ |
| **Concrete** | DNN-GLM | $0.330\pm_{0.047}$ | $-0.316\pm_{0.501}$ | $0.003\pm_{0.001}$ |
| | LL-GLM | $0.310\pm_{0.020}$ | $-0.561\pm_{0.223}$ | $0.002\pm_{0.002}$ |
| **Naval** | DNN-GLM | $0.049\pm_{0.012}$ | $-2.546\pm_{0.134}$ | $0.002\pm_{0.002}$ |
| | LL-GLM | $0.037\pm_{0.004}$ | $-2.656\pm_{0.173}$ | $0.000\pm_{0.000}$ |
| **CCPP** | DNN-GLM | $0.244\pm_{0.008}$ | $-0.885\pm_{0.020}$ | $0.000\pm_{0.000}$ |
| | LL-GLM | $0.243\pm_{0.004}$ | $-0.902\pm_{0.030}$ | $0.000\pm_{0.000}$ |
| **Wine** | DNN-GLM | $0.789\pm_{0.042}$ | $0.284\pm_{0.066}$ | $0.001\pm_{0.000}$ |
| | LL-GLM | $0.796\pm_{0.044}$ | $0.309\pm_{0.067}$ | $0.001\pm_{0.001}$ |
| **Yacht** | DNN-GLM | $0.042\pm_{0.013}$ | $-1.561\pm_{2.319}$ | $0.012\pm_{0.010}$ |
| | LL-GLM | $0.043\pm_{0.018}$ | $-2.197\pm_{1.210}$ | $0.008\pm_{0.007}$ |
| **Song** | DNN-GLM | $0.839\pm_{0.014}$ | $0.646\pm_{0.056}$ | $0.001\pm_{0.000}$ |
| | LL-GLM | $0.829\pm_{0.005}$ | $0.397\pm_{0.016}$ | $0.000\pm_{0.000}$ |

as Expected Calibration Error (ECE), Negative-Log Likelihood (NLL) or Area-Under the Receiver-Operator Characteristic (AUCROC) are either flawed measurements, based on poor measures of uncertainty, or are measuring predictive ability. To combat this, we use the following metrics:

- AUCROC with variance of softmax predictions as the score, for correctly predicted vs. incorrectly predicted points (VARROC-ID), and for correctly predicted vs. OOD points (VARROC-OOD).

- The same metrics but with Mutual Information (MI) as the score (VARROC-MI-ID and VARROC-MI-OOD).

- The log-predictive point density (LPPD), with predictive probabilities computed using the multi-class probit approximation (Gibbs, 1998).

See Appendix F for in-depth discussion and definitions.

We compare LL-GLM and DNN-GLM on a range of large image classification tasks, using these metrics. We compare also to DE, SWAG, LLA, MC-Dropout, and SMS-UBU (Paulin et al., 2024) to showcase the strong performance of these GLMs as UQ methods. We display bar plots of the VARROC / VARROC-MI results in Figure 6, for ResNet9 trained on FashionMNIST, ResNet50 trained on CIFAR-10, ResNet50 trained on CIFAR-100, and ResNet50 trained on ImageNet. The datasets used for OOD were MNIST, CIFAR-100, CIFAR-10, and iNaturalist OOD Plants (Huang & Li, 2021) respectively. Note that for ImageNet, DE, LLA, SWAG and SMS-UBU were excluded due to computational constraints. Numerical values for Figure 6, as well as values for LPPD, execution time and memory usage are found in

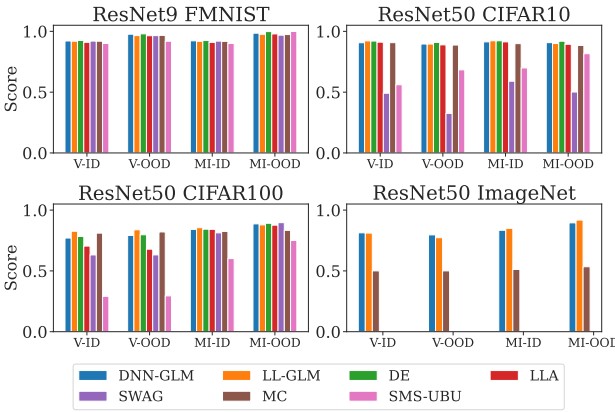

*Figure 6.* VARROC-ID (V-ID), VARROC-OOD (V-OOD), VARROC-MI-ID (MI-ID) and VARROC-MI-OOD (MI-OOD) results for image classification tasks.

*Table 2.* Comparison of LL-GLM vs. DNN-GLM on a GPT2 model, fine-tuned on the IMDB dataset. Value for VARROC-ID is mean and standard deviation over 5 independent runs.

| Method | VARROC-ID | Wall-Time | Memory |
|--------|-----------|-----------|--------|
| DNN-GLM | $0.842 \pm 0.015$ | 2.55 hr | 71.23 GB |
| LL-GLM | $0.823 \pm 0.025$ | 0.26 hr | 20.97 GB |

Table 4. We see that both DNN-GLM and LL-GLM perform equally well, and that they are consistently among the strongest performing methods. Note that DE performs the strongest; this is not surprising, considering the huge computational requirement. We provide the relative difference between the GLMs in Table 5. We see that across tasks, difference in performance is minimal, and that neither GLM consistently outperforms the other. Further, we observe that reduction in time and memory is significant for LL-GLM vs. DNN-GLM.

### 5.3. Language Modeling

We now compare the DNN-GLM vs. LL-GLM on a language processing task, using the 124m parameter GPT2 (Radford et al., 2019), with pre-trained weights. We fine-tune a classification head on the IMDB sentiment dataset (Maas et al., 2011), sample from the posterior of a LL-GLM and DNN-GLM, and use these samples to compute the VARROC-ID metric. We report this in Table 2, along with the wall-time on an H100 GPU for inference and sampling, and the maximum memory used. We observe no statistical difference between the methods for VARROC-ID, but observe a $10\times$ speed-up for wall-time and a $3.5\times$ decrease in memory use.

## 6. Conclusion

Last-layer approximations for Bayesian methods bring computational speed-ups, yet have often been thought to hinder

performance. From a theoretical point of view, comparing a last-layer to a full linearization can be achieved by considering the BFE of a GP employing the induced kernel functions. By employing tools from RMT, and for a randomly-initialized DNN, we have found no specific regimes where a last-layer approximation decreases model-fit. Through the introduction of a lightweight sampling algorithm, we were able to compare the UQ performance of the LL-GLM and the DNN-GLM on a series of modern machine learning tasks; no practical difference in performance was found. This is a surprising result; specifically, this suggests that epistemic uncertainty in a DNN arises not in the feature representations, but rather in how the features are *used* for prediction. For the UQ community, we hope that these results spur the adoption of the last-layer linearization.

**Limitations & Future Work** Note that the conditions in Assumption 1 prevent direct connection to theoretical results for trained DNNs on realistic datasets. While there is evidence that the theory-practice gap may be small in certain regimes, extending Theorem 1 and Theorem 2 to trained DNNs under practical assumptions, for example through the spectral distributions derived in Hodgkinson et al. (2025), is an exciting, yet challenging, avenue for future work. Further, the sampling technique of Algorithm 1 only allows sampling from the posterior of a noiseless GLM, and hence in our work we only model the *epistemic* uncertainty of a DNN. Extending Algorithm 1 to include noise models on the data (allowing for *aleatoric* uncertainty) would enable a more complete comparison of the DNN-GLM vs. LL-GLM for UQ. Finally, use of MCMC techniques to sample from the last-layer posterior may bring further performance and computational improvements.

## Acknowledgments

Liam Hodgkinson is supported by the Australian Research Council through a Discovery Early Career Researcher Award (DE240100144). Fred Roosta was partially supported by the Australian Research Council through an Industrial Transformation Training Centre for Information Resilience (IC200100022).

## Impact Statement

This paper presents work whose goal is to advance the field of machine learning. There are many potential societal consequences of our work, none of which we feel must be specifically highlighted here.

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

# A. Limiting Free Energy

A key contribution of this work is the characterization of the limit of (3) under certain assumptions on the network and the data, for the GP specified in Section 2. In the following sections of the appendix, we will detail both the resulting limiting quantity for (4) as well as the necessary derivations. In this section, we first discuss the general strategy. We then provide the necessary assumptions on the network and the data. In Appendix A.1 we reproduce the work of Fan & Wang (2020), who characterized the Stieltjes transform of the Gram matrix for the CK and NTK. From there, we detail in Appendix A.2 the derivation of the limiting log-determinant term for the limiting energy. Finally, in Appendix A.3 we combine the preceding sections to fully characterize the free energy, as well as detail how one would numerically evaluate it, and provide empirical validation of the limit. In Appendix A.4 we characterize a lower bound on the BFE, and in Appendix A.5 we characterize explicit expression for the limiting BFE when $\gamma \to 0$ or $\gamma \to \infty$. We combine these preceding sections in Appendix A.6 to prove Theorem 1. Finally, we prove results on strong-descent and weak-descent in Appendix A.7 and Appendix A.8.

We take the following definition from (Fan & Wang, 2020) for the inputs we are considering:

**Definition 3.** *Let $\epsilon, B > 0$. A matrix $\mathbf{X} \in \mathbb{R}^{d \times n}$ is $(\epsilon, B)$-orthonormal if its columns satisfy, for every $\alpha \neq \beta \in \{1, \ldots, n\}$,*

$$\left| ||\mathbf{x}_\alpha||^2 - 1 \right| \leq \epsilon, \quad \left| \mathbf{x}_\alpha^T \mathbf{x}_\beta \right| \leq \epsilon, \quad ||\mathbf{X}|| \leq B, \quad \sum_{\alpha=1}^n (||\mathbf{x}_\alpha||^2 - 1)^2 \leq B^2.$$

We take the following assumptions, which have been modified from Fan & Wang (2020):

**Assumption 1.** *1. The number of layers $L \geq 1$ in the network $f_\theta$ is fixed, and $n, d_0, d_1, \ldots, d_L \to \infty$ such that:*

*1.a. The weights are i.i.d. and distributed as $\mathcal{N}(0, 1)$.*

*1.b. The activation $\sigma(\mathbf{x})$ is twice-differentiable, with finite first- and second-derivatives. For $\xi \sim \mathcal{N}(0, 1)$, we have $\mathbb{E}[\sigma(\xi)] = 0$ and $\mathbb{E}[\sigma^2(\xi)] = 1$.*

*1.c. Input $\mathbf{X} \in \mathbb{R}^{d_0 \times n}$ is $(\epsilon_n, B)$-orthonormal, where $B$ is a constant, and $\epsilon_n n^{1/4} \to 0$ as $n \to \infty$.*

*1.d. As $n \to \infty$, $\lim spec\, \mathbf{X}^T \mathbf{X} = \mu_0$ for a probability distribution $\mu_0$ on $[0, \infty)$, and $\lim n/d_l = \gamma_l$ for constants $\gamma_l \in (0, \infty)$ and each $l = 1, 2, \ldots, L$.*

*2. Outputs $y_1, y_2, \ldots$ are distributed as $y_i \overset{i.i.d}{\sim} \mathcal{N}(0, 1)$, $i = 1, \ldots, n$.*

As an example, note that by Definition 3, Assumption 1 covers i.i.d input data $X_0 = [\mathbf{x}_1, \ldots, \mathbf{x}_n]$ where $\mathbf{x}_i \sim \mathcal{N}(\mathbf{0}, \mathbf{C})$ with Tr $\mathbf{C} = 1$. In this case, $\mu_0$ corresponds to the Marchenko-Pastur distribution.

We are interested in the limiting $\mathcal{F}_n^{\tau, \lambda}$ for large sample size and dimension, and where the kernel covariance function $\kappa$ is defined as either $\kappa = \kappa^{\text{CK}}$ or $\kappa = \kappa^{\text{NTK}}$. We define $\mathcal{F}_\infty^{\tau, \lambda} := \lim_{n \to \infty} n^{-1} \mathbb{E} \mathcal{F}_n^{\tau, \lambda}$. Looking at the normalized expected energy,

$$\frac{1}{n} \mathbb{E} \mathcal{F}_n^{\tau, \lambda} = \frac{\lambda}{2n} \mathbb{E} \mathbf{Y}^T (\mathbf{K_X} + \lambda \tau I)^{-1} \mathbf{Y} + \frac{1}{2n} \mathbb{E} \log \det(\mathbf{K_X} + \lambda \tau I) - \frac{1}{2} \log(\frac{\lambda}{2\pi})$$

$$= \underbrace{\frac{\lambda}{2n} \mathbb{E} \text{Tr}(\mathbf{K_X} + \lambda \tau I)^{-1}}_{(a)} + \underbrace{\frac{1}{2n} \mathbb{E} \log \det(\mathbf{K_X} + \lambda \tau I)}_{(b)} - \frac{1}{2} \log(\frac{\lambda}{2\pi}), \tag{4}$$

where we have used that $\mathbb{E}[\mathbf{Y}^T \mathbf{K_X} \mathbf{Y}] = \mathbb{E}[\text{Tr}\, \mathbf{K_X}]$ when $\mathbf{K_X}$ is independent of $\mathbf{Y}$. To compute the limiting mean free energy, we need to evaluate $(a)$ and $(b)$ as $n, d_0, d_1, \ldots, d_L \to \infty$ according to Assumption 1. As this will require evaluating functions of the limiting spectrum of the respective random Gram matrices, we first need to characterize the respective limiting spectrum.

## A.1. Stieltjes Transform of Kernel Gram Matrices

For a random matrix $\mathbf{M} \in \mathbb{R}^{n \times n}$, the empirical spectral measure of $\mathbf{M}$ is $\mu_{\mathbf{M}} = \frac{1}{n} \sum_{i=1}^n \delta_{\lambda_i(\mathbf{M})}$. We define the Stieltjes transform of $\mu_{\mathbf{M}}$ as $m_{\mu_{\mathbf{M}}}(z) := \int 1/(t - z) \mu_{\mathbf{M}}(dt)$, for $z \notin \text{spec}(\mathbf{M})$. The importance of the Stieltjes transform can be seen from the following relationship:

$$m_{\mu_{\mathbf{M}}}(z) = \frac{1}{n} \text{Tr}(\mathbf{M} - zI)^{-1}. \tag{5}$$

We require the Stieltjes transform $m_{\mathbf{K}}(z)$ for $\mathbf{K} \in \{\mathbf{K}_{\mathbf{X}}^{CK}, \mathbf{K}_{\mathbf{X}}^{NTK}\}$, for the distribution of $\lim \operatorname{spec}(\mathbf{K}_{\mathbf{X}})$. This has already been characterized in the work of Fan & Wang (2020). We firstly need to define some constants:

$$b_\sigma = \mathbb{E}[\sigma'(\xi)], \quad a_\sigma = \mathbb{E}[\sigma'(\xi)^2], \quad q_l = (b_\sigma^2)^{L-l}, \quad r_l = a_\sigma^{L-l}, \quad r_+ = \sum_{l=0}^{L-1} r_l - q_l. \tag{6}$$

We will make use of the following lemma in the proofs:

**Lemma 3** (Lemma 3.5 Fan & Wang (2020)). *Under Assumption 1, and with $r_+$ and $q_l$ defined as above,*

$$\lim \operatorname{spec} \mathbf{K}_{\mathbf{X}}^{NTK} = \lim \operatorname{spec} \left( r_+ I + \mathbf{X}_L^T \mathbf{X}_L + \sum_{l=0}^{L-1} q_l \mathbf{X}_l^T \mathbf{X}_l \right). \tag{7}$$

Hence, we see that the limiting spectrum of the NTK is the limiting spectrum of a linear combination of the intermediate Conjugate Kernel at each layer.

Let $\mathbf{z} = (z_{-1}, z_0, \ldots, z_L) \in \mathbb{C}^{L+2}$, and $\mathbf{w} = (w_{-1}, w_0, \ldots, w_L) \in \mathbb{C}^{L+2}$. For $l \geq 1$, we define the functions $s_l$ and $t_l$ recursively by

$$s_l(\mathbf{z}) = 1/z_l + \gamma_l t_{l-1} \left( \mathbf{z}_{\text{prev}}(s_l(\mathbf{z}), \mathbf{z}), (1 - b_\sigma^2, 0, \ldots, 0, b_\sigma^2) \right), \tag{8}$$

$$t_l(\mathbf{z}, \mathbf{w}) = w_l/z_l + t_{l-1}(\mathbf{z}_{\text{prev}}(s_l(\mathbf{z}), \mathbf{z}), \mathbf{w}_{\text{prev}}), \tag{9}$$

where

$$\mathbf{z}_{\text{prev}}(s_l(\mathbf{z}), \mathbf{z}) := \left( z_{-1} + \frac{1 - b_\sigma^2}{s_l(\mathbf{z})}, z_0, \ldots, z_{l-2}, z_{l-1} + \frac{b_\sigma^2}{s_l(\mathbf{z})} \right) \in \mathbb{C}^- \times \mathbb{R}^{l-1} \times \mathbb{C}^*, \tag{10}$$

$$\mathbf{w}_{\text{prev}} := (w_{-1}, \ldots, w_{l-1}) - (w_l/z_l) \cdot (z_{-1}, \ldots, z_{l-1}) \in \mathbb{C}^{l+1}.$$

For $l = 0$, we define the first function $t_0$ by

$$t_0 \left( (z_{-1}, z_0), (w_{-1}, w_0) \right) = \frac{z_0 w_{-1} - z_{-1} w_0}{z_0^2} m_0 \left( -\frac{z_{-1}}{z_0} \right) + \frac{w_0}{z_0}.$$

Finally, we define the following notation:

$$\mathbf{z}^\xi = (z_{-1} - \xi, z_0, \ldots, z_l).$$

We combine the relevant results from (Fan & Wang, 2020) into the following theorem:

**Theorem 3.** *(Fan & Wang, 2020) Suppose $b_\sigma \neq 0$. Under Assumption 1, for any fixed values $z_{-1}, z_0, \ldots, z_L \in \mathbb{R}$ where $z_L \neq 0$, we have $\lim \operatorname{spec}(z_{-1} I + z_0 \mathbf{X}_0^T \mathbf{X}_0 + \cdots + z_L \mathbf{X}_L^T \mathbf{X}_L) = \nu$ where $\nu$ is the probability distribution with Stieltjes transform $m_\nu(z) = t_L((-z + z_{-1}, z_0, \ldots, z_L), (1, 0, \ldots, 0))$. For any $\mathbf{z} \in \mathbb{C}^- \times \mathbb{R}^l \times \mathbb{C}^*$ and $l \geq 1$, there is a unique solution to the fixed point equation (8), such that we can evaluate $t_L(\mathbf{z}, \mathbf{w})$, for some $\mathbf{w} \in \mathbb{C}^{L+2}$. In particular, $\lim \operatorname{spec} \mathbf{K}_{\mathbf{X}}^{CK}$ and $\lim \operatorname{spec} \mathbf{K}_{\mathbf{X}}^{NTK}$ are the probability distributions with Stieltjes transforms*

$$m_{CK}(z) = t_L((-z, 0, \ldots, 0, 1), (1, 0, \ldots, 0))$$

$$m_{NTK}(z) = t_L((-z + r_+, q_0, \ldots, q_{L-1}, 1), (1, 0, \ldots, 0)).$$

We thus note that by (5), $\frac{\lambda}{2n} \mathbb{E} \operatorname{Tr}(\mathbf{K}_{\mathbf{X}} + \lambda \tau I)^{-1} \to \frac{\lambda}{2} m_{\mathbf{K}}(-\lambda \tau)$, for $\mathbf{K} \in \{\mathbf{K}_{\mathbf{X}}^{CK}, \mathbf{K}_{\mathbf{X}}^{NTK}\}$.

### A.2. Derivation of Log-Determinant Term

For the log-determinant term, we note that by the law of total expectation, combined with (7),

$$\frac{1}{n} \mathbb{E} \log \det(\mathbf{K}_{\mathbf{X}} + \lambda \tau I) = \frac{1}{n} \mathbb{E}_0 \mathbb{E}_1 \ldots \mathbb{E}_{L-1} \log \det(r_+ I + \mathbf{X}_L^T \mathbf{X}_L + \sum_{l=0}^{L-1} q_l \mathbf{X}_l^T \mathbf{X}_l + \lambda \tau I),$$

where $\mathbb{E}_l = \mathbb{E}[\,.\,|\mathbf{X}_l, \ldots, \mathbf{X}_0]$. Hence, we will attempt to find the log-determinant for each layer $l$, conditioned on the previous $l-1$ layers. We will then iterate through the layers to find the total expected log-determinant. We further abstract this process by considering an abstracted NTK gram matrix, $\mathbf{L}_l : \mathbb{R}^{l+2} \to \mathbb{R}^{n \times n}$, and a constant matrix $\mathbf{S}_l : \mathbb{R}^{l+2} \to \mathbb{R}^{n \times n}$, defined for $\mathbf{z}_l = (z_{-1}, z_0, \ldots, z_l) \in \mathbb{R}^{l+2}$ as

$$\mathbf{L}_l(\mathbf{z}_l) := z_l \mathbf{X}_l^T \mathbf{X}_l + \mathbf{S}_l(\mathbf{z}_l) \tag{11}$$

$$\mathbf{S}_l(\mathbf{z}_l) := \sum_{i=0}^{l-1} z_i \mathbf{X}_i^T \mathbf{X}_i + z_{-1} \mathbf{I}.$$

Hence, we would like to find the following:

$$\lim_{n \to \infty} \frac{1}{n} \mathbb{E}_{l-1} \log \det \left( \mathbf{L}_l(\mathbf{z}_l) + x\mathbf{I} \right),$$

for $x \in \mathbb{R}$. To aid us in computing this limiting log-determinant, we employ the Shannon transform (Tulino et al., 2004) $\mathcal{V}_{\mathbf{L}_l}(x; \mathbf{z}_l)$ of $\mathbf{L}_l(\mathbf{z}_l)$, for $x > 0$:

$$\mathcal{V}_{\mathbf{L}_l}(x; \mathbf{z}_l) = \frac{1}{n} \mathbb{E}_{l-1} \log \det \left( I + \frac{1}{x} \mathbf{L}_l(\mathbf{z}_l) \right) = \int_0^\infty \log \left( 1 + \frac{t}{x} \right) \mu_{\mathbf{L}_l}(dt; \mathbf{z}_l) = \int_x^{+\infty} \left( \frac{1}{z} - m_{\mathbf{L}_l}(-z; \mathbf{z}_l) \right) dz, \tag{12}$$

where $m_{\mathbf{L}_l}(x; \mathbf{z}_l)$ is the Stieltjes transform of the empirical spectral distribution, $\mu_{\mathbf{L}_l}(.; \mathbf{z}_l)$, of $\mathbf{L}_l(\mathbf{z}_l)$. Thus, to find $\lim_{n \to \infty} \frac{1}{n} \mathbb{E}_{l-1} \log \det \left( \mathbf{L}_l(\mathbf{z}_l) + x\mathbf{I} \right)$, we simply need to find some $\tilde{D}_l(z)$ such that $\frac{d}{dz} \tilde{D}_l(z) = 1/z - m_{\mathbf{L}_l}(-z; \mathbf{z}_l)$, i.e. the anti-derivative of the integrand on the right-hand side of (12).

For this, we first need to find the Stieltjes transform, $m_{\mathbf{L}_l}(x; \mathbf{z}_l)$, of the empirical spectral distribution, $\mu_{\mathbf{L}_l}(.\ :\ \mathbf{z}_l)$, for $\mathbf{L}_l(\mathbf{z}_l)$, conditioned on the previous $l-1$ layers. As we take Assumption 1, we employ Fan & Wang (2020, Theorem 3.7, Lemma G.6), as well as the definition in (9), to note that

$$\begin{aligned}
m_{\mathbf{L}_l}(x; \mathbf{z}_l) &\to \lim_{n \to \infty} \frac{1}{n} \mathrm{Tr} \left( z_l \mathbf{X}_l^T \mathbf{X}_l + \mathbf{S}_l(\mathbf{z}_l) - x\mathbf{I} \right)^{-1} \\
&= t_l(\mathbf{z}_l^x, (1, 0, \ldots, 0)) \\
&= t_{l-1}(\mathbf{z}_{\mathrm{prev}}(s_l(\mathbf{z}_l^x), \mathbf{z}_l^x), (1, 0, \ldots, 0)) \\
&= \lim_{n \to \infty} \frac{1}{n} \mathrm{Tr} \left( \mathbf{S}_l(\mathbf{z}_l) + s_l(\mathbf{z}_l^x)^{-1} \tilde{\mathbf{\Phi}}_l - x\mathbf{I} \right)^{-1},
\end{aligned}$$

where $\tilde{\mathbf{\Phi}}_l = b_\sigma^2 \mathbf{X}_{l-1}^T \mathbf{X}_{l-1} + (1 - b_\sigma^2)\mathbf{I}$ is deterministic conditioned on $\mathbf{X}_{l-1}, \ldots, \mathbf{X}_0$. We note also that, by (8),

$$\begin{aligned}
s_l(\mathbf{z}_l^x) &= \frac{1}{z_l} + \gamma_l t_{l-1}(\mathbf{z}_{\mathrm{prev}}(s_l(\mathbf{z}_l^x), \mathbf{z}_l^x), (1 - b_\sigma^2, 0, \ldots, 0, b_\sigma^2)) \\
&= \frac{1}{z_l} + \gamma_l \lim_{n \to \infty} \frac{1}{n} \mathrm{Tr}\, \tilde{\mathbf{\Phi}}_l \left( \mathbf{S}_l(\mathbf{z}_l) + s_l(\mathbf{z}_l^x)^{-1} \tilde{\mathbf{\Phi}}_l - x\mathbf{I} \right)^{-1}. \tag{13}
\end{aligned}$$

Hence,

$$\frac{z_l s_l(\mathbf{z}_l^x) - 1}{\gamma_l z_l} = \lim_{n \to \infty} \frac{1}{n} \mathrm{Tr}\, \tilde{\mathbf{\Phi}}_l \left( \mathbf{S}_l(\mathbf{z}_l) + s_l(\mathbf{z}_l^x)^{-1} \tilde{\mathbf{\Phi}}_l - x\mathbf{I} \right)^{-1}. \tag{14}$$

As we will be taking derivatives with respect to the input $x$ of $m_{\mathbf{L}_l}(x; \mathbf{z}_l)$, and as this will involve $s_l(\mathbf{z}_l^x)$ which is a function of $x$, we employ the short-hand notation

$$s_l(x) \equiv s_l(\mathbf{z}_l^x) = \frac{1}{z_l} + \gamma_l \lim_{n \to \infty} \frac{1}{n} \mathrm{Tr}\, \tilde{\mathbf{\Phi}}_l \left( \mathbf{S}_l + s_l(x)^{-1} \tilde{\mathbf{\Phi}}_l - x\mathbf{I} \right)^{-1},$$

where we have also dropped the dependence of $\mathbf{S}_l$ on $\mathbf{z}_l$.

**Lemma 4** (Shannon Transform). *Let $\mathbf{z}_l = (z_{-1}, z_0, \ldots, z_l) \in \mathbb{R}^{l+2}$, and take $\mathbf{L}_l$ as defined in (11). The anti-derivative of $1/z - m_{\mathbf{L}_l}(-z; \mathbf{z}_l)$, in the limit as $n \to \infty$, and under Assumption 1, is given by*

$$\tilde{D}_l(z) = -\frac{1}{n} \log \det \left( \mathbf{S}_l + s_l^{-1}(-z)\tilde{\boldsymbol{\Phi}}_l + z I_n \right) - \frac{1}{\gamma_l} \ln s_l(-z) - \frac{1}{z_l \gamma_l s_l(-z)} + \ln z.$$

*Further, we have that*

$$\mathcal{V}_l(x) = \frac{1}{n} \log \det \left( I + \frac{1}{x} \mathbf{L}_l(\mathbf{z}_l) \right) \to \frac{1}{n} \log \det \left( I_n + \frac{1}{x} \left( \mathbf{S}_l + s_l^{-1}(-x)\tilde{\boldsymbol{\Phi}}_l \right) \right)$$

$$+ \frac{1}{\gamma_l} \ln s_l(-x) + \frac{1}{z_l \gamma_l s_l(-x)} + \frac{1}{\gamma_l} \ln z_l - \frac{1}{\gamma_l}.$$

*Proof.* We will show that $d/dz \; \tilde{D}_l(z) = 1/z - m_{\mathbf{L}_l}(-z)$. We use Jacobi's formula,

$$\frac{d}{dz} \log \det \mathbf{A}(z) = \mathrm{Tr} \left( \mathbf{A}^{-1}(z) \frac{d\mathbf{A}(z)}{dz} \right).$$

We take the derivative of the log-determinant term first:

$$\frac{1}{n} \frac{d}{dz} \log \det \left( \mathbf{S}_l + s_l^{-1}(-z)\tilde{\boldsymbol{\Phi}}_l + z I_n \right) = \lim_{n \to \infty} \frac{1}{n} \mathrm{Tr} \left( \mathbf{S}_l + s_l^{-1}(-z)\tilde{\boldsymbol{\Phi}}_l + z I_n \right)^{-1} \left( -s_l^{-2}(-z)s_l'(-z)\tilde{\boldsymbol{\Phi}}_l + I_n \right)$$

$$= s_l^{-2}(-z)s_l'(-z) \lim_{n \to \infty} \frac{1}{n} \mathrm{Tr}\, \tilde{\boldsymbol{\Phi}}_l \left( \mathbf{S}_l + s_l^{-1}(-z)\tilde{\boldsymbol{\Phi}}_l + z I_n \right)^{-1}$$

$$+ \lim_{n \to \infty} \frac{1}{n} \mathrm{Tr} \left( \mathbf{S}_l + s_l^{-1}(-z)\tilde{\boldsymbol{\Phi}}_l + z I_n \right)^{-1}$$

$$= s_l^{-2}(-z)s_l'(-z) \frac{z_l s_l(\mathbf{z}_l) - 1}{\gamma_l z_l} + m_{\mathbf{L}_l}(-z)$$

$$= -\frac{s_l'(-z)}{s_l^2(-z)\gamma_l z_l} + \frac{s_l'(-z)}{\gamma_l s_l(-z)} + m_{\mathbf{L}_l}(-z),$$

where we used (14). Combining this with the rest of the derivatives we have

$$\frac{d}{dz} \tilde{D}_l(z) = \frac{s_l'(-z)}{s_l^2(-z)\gamma_l z_l} - \frac{s_l'(-z)}{\gamma_l s_l(-z)} - m_{\mathbf{L}_l}(-z) + \frac{s_l'(-z)}{\gamma_l s_l(-z)} - \frac{s_l'(-z)}{s_l^2(-z)\gamma_l z_l} + \frac{1}{z}$$

$$= \frac{1}{z} - m_{\mathbf{L}_l}(-z).$$

We also need to look at the asymptotics of $\tilde{D}_l(z)$, as by (12), we need to evaluate this anti-derivative at $z = +\infty$. Now, by Fan & Wang (2020, Proposition E.3), we know that $s_l$ is bounded. Further, Fan & Wang (2020, Corollary G.5) states that there is a unique solution $s_l(\mathbf{z}_l)$ in $\mathbb{C}^+$, as defined by (8). Hence, if we can find a bounded solution to (8), it must be the unique solution. We have that,

$$\lim_{x \to +\infty} s_l(-x) = \frac{1}{z_l} + \gamma_l \lim_{n \to \infty, x \to +\infty} \frac{1}{n} \mathrm{Tr}\, \tilde{\boldsymbol{\Phi}}_l \left( \mathbf{S}_l(\mathbf{z}_l) + s_l(-x)^{-1}\tilde{\boldsymbol{\Phi}}_l + x I \right)^{-1}.$$

Now if $s_l(-x)$ is bounded (as is $\mathbf{S}_l$ and $\tilde{\boldsymbol{\Phi}}_l$ by the $(\epsilon, B)-$orthonormality of Assumption 1), then as $x \to \infty$, $\mathrm{Tr}\, \tilde{\boldsymbol{\Phi}}_l \left( \mathbf{S}_l(\mathbf{z}_l) + s_l(-x)^{-1}\tilde{\boldsymbol{\Phi}}_l + x I \right)^{-1}$ is dominated by the $x I$ term, and hence this trace-term will go to 0. Hence,

$$\lim_{x \to +\infty} s_l(-x) = \frac{1}{z_l}.$$

Then,

$$\lim_{z \to \infty} \tilde{D}_l(z) = -\frac{1}{n} \lim_{z \to \infty} \log \det \left( \frac{1}{z} \left( \mathbf{S}_l + s_l^{-1}(-z)\tilde{\boldsymbol{\Phi}}_l \right) + I_n \right) - \lim_{z \to \infty} \frac{1}{\gamma_l} \ln s_l(-z) - \lim_{z \to \infty} \frac{1}{z_l \gamma_l s_l(-z)}$$

$$= \frac{1}{\gamma_l} \ln z_l - \frac{1}{\gamma_l}.$$

Hence, we can finally conclude that

$$\mathcal{V}_l(x) = \frac{1}{n} \log \det \left( \mathbf{I} + \frac{1}{x} \mathbf{L}_l(\mathbf{z}_l) \right) \to \frac{1}{n} \log \det \left( \mathbf{I}_n + \frac{1}{x} \left( \mathbf{S}_l + s_l^{-1}(-z) \tilde{\mathbf{\Phi}}_l \right) \right)$$
$$+ \frac{1}{\gamma_l} \ln s_l(-z) + \frac{1}{z_l \gamma_l s_l(-x)} + \frac{1}{\gamma_l} \ln z_l - \frac{1}{\gamma_l}.$$

□

Before we iterate through the layers, we firstly redefine the notation for inputs $\mathbf{z}_l$. Firstly, let $\mathbf{z}_L = (z_{-1}, z_0, \dots, z_L) \in \mathbb{R}^{L+2}$. Further, define $\mathbf{z}_{l-1}^{\xi} := \mathbf{z}_{\mathrm{prev}}(s_l(\mathbf{z}_l^{\xi}), \mathbf{z}_l^{\xi})$ and $\mathbf{z}_{l-1} := \mathbf{z}_{\mathrm{prev}}(s_l(\mathbf{z}_l), \mathbf{z}_l)$. The elements of the vector $\mathbf{z}_{l-1}$ and $\mathbf{z}_{l-1}^{\xi}$ will then be written as

$$\mathbf{z}_{l-1} = (z_{-1}^{(l-1)}, z_0^{(l-1)}, \dots, z_{-1}^{(l-2)}, z_{-1}^{(l-1)})$$
$$\mathbf{z}_{l-1}^{\xi} = (z_{-1}^{(l-1)} - \xi, z_0^{(l-1)}, \dots, z_{-1}^{(l-2)}, z_{-1}^{(l-1)}).$$

Further, we define the log-determinant of $\mathbf{L}_l$ for input $\mathbf{z}_L$ as

$$D_L(x, \mathbf{z}_L) = \lim_{n, d_0, \dots, d_L \to \infty} \frac{1}{n} \mathbb{E} \log \det(x + z_L \mathbf{X}_L^T \mathbf{X}_L + \sum_{l=0}^{L-1} z_l \mathbf{X}_l^T \mathbf{X}_l + z_{-1} \mathbf{I}_n).$$

**Theorem 4.** *Define the following quantity:*

$$C_l(x, \mathbf{z}_l, s_l(\mathbf{z}_l^{-x})) := \frac{1}{\gamma_l} \left( \ln(s_l(\mathbf{z}_l^{-x}) z_l^{(l)}) + \frac{1}{s_l(\mathbf{z}_l^{-x}) z_l^{(l)}} - 1 \right).$$

*Then, under Assumption 1,*

$$D_L(x, \mathbf{z}_L) = \sum_{l=1}^{L} C_l(x, \mathbf{z}_l, s_l(\mathbf{z}_l^{-x})) + \mathcal{V}_0 \left( \frac{x + z_{-1}^{(0)}}{z_0^{(0)}} \right) + \ln \left( 1 + \frac{z_{-1}^{(0)}}{x} \right),$$

*where $s_l(\mathbf{z}_l^{-x})$ is given as the solution of (8) for $1 \le l \le L$, and $\mathcal{V}_0(x)$ is the Shannon transform of the input spectral distribution $\mu_0$. Further, the log-determinant for the CK and NTK is given by $D_L(x, (0, \dots, 0, 1))$ and $D_L(x, (r_+, q_0, \dots, q_{L-1}, 1))$ respectively.*

*Proof.* We note from Lemma 4 that

$$\frac{1}{n} \log \det \left( \mathbf{I} + \frac{1}{x} \mathbf{L}_l(\mathbf{z}_l) \right) \to \frac{1}{n} \log \det \left( \mathbf{I}_n + \frac{1}{x} \left( \mathbf{S}_l + s_l^{-1}(-x) \tilde{\mathbf{\Phi}}_l \right) \right)$$
$$+ \frac{1}{\gamma_l} \ln s_l(-x) + \frac{1}{z_l^{(l)} \gamma_l s_l(-x)} + \frac{1}{\gamma_l} \ln z_l^{(l)} - \frac{1}{\gamma_l}.$$

Now,

$$\mathbf{S}_l + s_l^{-1}(-x) \tilde{\mathbf{\Phi}}_l = \sum_{i=0}^{l-1} z_i^{(l)} \mathbf{X}_i^T \mathbf{X}_i + z_{-1}^{(l)} \mathbf{I} + \frac{b_{\sigma}^2}{s_l(-x)} \mathbf{X}_{l-1}^T \mathbf{X}_{l-1} + \frac{1 - b_{\sigma}^2}{s_l(-x)} \mathbf{I}$$
$$= \left( z_{l-1}^{(l)} + \frac{b_{\sigma}^2}{s_l(-x)} \right) \mathbf{X}_{l-1}^T \mathbf{X}_{l-1} + \sum_{i=0}^{l-2} z_i^{(l)} \mathbf{X}_i^T \mathbf{X}_i + \left( z_{-1}^{(l)} + \frac{1 - b_{\sigma}^2}{s_l(-x)} \right) \mathbf{I}$$
$$= \left( z_{l-1}^{(l)} + \frac{b_{\sigma}^2}{s_l(-x)} \right) \mathbf{X}_{l-1}^T \mathbf{X}_{l-1} + \mathbf{S}_{l-1}(\mathbf{z}_{\mathrm{prev}}(s_l(\mathbf{z}_l^{-x}), \mathbf{z}_l))$$
$$= \mathbf{L}_{l-1}(\mathbf{z}_{\mathrm{prev}}(s_l(\mathbf{z}_l^{-x}), \mathbf{z}_l)),$$

where we have employed the short-hand notation from (10). Hence,

$$
\mathbb{E}_{l-2}\mathbb{E}_{l-1}\frac{1}{n}\log\det\left(\mathbf{I}+\frac{1}{x}\mathbf{L}_l(\mathbf{z}_l)\right) \sim \mathbb{E}_{l-2}\frac{1}{n}\log\det\left(\mathbf{I}_n+\frac{1}{x}\left(\mathbf{S}_l+s_l^{-1}(-x)\tilde{\mathbf{\Phi}}_l\right)\right)
$$

$$
+\frac{1}{\gamma_l}\ln s_l(-x)+\frac{1}{z_l^{(l)}\gamma_l s_l(-x)}+\frac{1}{\gamma_l}\ln z_l^{(l)}-\frac{1}{\gamma_l}
$$

$$
\sim \mathbb{E}_{l-2}\frac{1}{n}\log\det\left(\mathbf{I}_n+\frac{1}{x}\mathbf{L}_{l-1}(\mathbf{z}_{l-1})\right)+C_l(x,\mathbf{z}_l,s_l(\mathbf{z}_l^{-x}))
$$

$$
\sim \frac{1}{n}\log\det\left(\mathbf{I}_n+\frac{1}{x}\mathbf{L}_{l-2}(\mathbf{z}_{l-2})\right)
$$

$$
+C_{l-1}(x,\mathbf{z}_{l-1},s_{l-1}(\mathbf{z}_{l-1}^{-x}))+C_l(x,\mathbf{z}_l,s_l(\mathbf{z}_l^{-x})).
$$

We finally need to define the log-determinant for layer $l=0$. Note that at the 0th-layer,

$$
\frac{1}{n}\log\det\left(\mathbf{I}_n+\frac{1}{x}\mathbf{L}_0(\mathbf{z}_0)\right) = \frac{1}{n}\log\det\left(\mathbf{I}_n+\frac{1}{x}(z_0^{(0)}\mathbf{X}_0^T\mathbf{X}_0+z_{-1}^{(0)}\mathbf{I}_n)\right)
$$

$$
= \frac{1}{n}\log\det\left(\left(1+\frac{z_{-1}^{(0)}}{x}\right)\mathbf{I}_n+\frac{z_0^{(0)}}{x}\mathbf{X}_0^T\mathbf{X}_0\right)
$$

$$
= \frac{1}{n}\log\det\left(\mathbf{I}_n+\frac{z_0^{(0)}}{x+z_{-1}^{(0)}}\mathbf{X}_0^T\mathbf{X}_0\right)+\ln\left(\frac{x+z_{-1}^{(0)}}{x}\right)
$$

$$
= \mathcal{V}_0\left(\frac{x+z_{-1}^{(0)}}{z_0^{(0)}}\right)+\ln\left(\frac{x+z_{-1}^{(0)}}{x}\right),
$$

where $\mathcal{V}_0(x)$ is the Shannon transform for the input spectral distribution, $\mu_0$.

We can therefore put everything together to form the total log-determinant:

$$
D_L(x,\mathbf{z}_L) = \lim_{n,d_0,\ldots,d_L\to\infty}\frac{1}{n}\mathbb{E}\log\det(x+z_L\mathbf{X}_L^T\mathbf{X}_L+\sum_{l=0}^{L-1}z_l\mathbf{X}_l^T\mathbf{X}_l+z_{-1}\mathbf{I}_n)
$$

$$
= \sum_{l=1}^{L}C_l(x,\mathbf{z}_l,s_l(\mathbf{z}_l^{-x}))+\mathcal{V}_0\left(\frac{x+z_{-1}^{(0)}}{z_0^{(0)}}\right)+\ln\left(1+\frac{z_{-1}^{(0)}}{x}\right).
$$

$\square$

**Lemma 5** (Marchenko-Pastur Input Distribution). *Let the input $\mathbf{X}=d_0^{-1/2}\mathbf{Z}$, with $\mathbf{Z}$ having i.i.d. columns that are zero mean and unit covariance. Then,*

$$
D_L(x,\mathbf{z}_L) = \sum_{l=0}^{L}C_l(x,\mathbf{z}_l,s_l(\mathbf{z}_l^{-x}))+\ln(x+1/s_0(\mathbf{z}_0^{-x})+z_{-1}^{(0)}). \tag{15}
$$

*Proof.* We want to compute $\frac{1}{n}\log\det\left(\mathbf{I}_n+\frac{1}{x}(z_0^{(0)}\mathbf{X}_0^T\mathbf{X}_0+z_{-1}^{(0)}\mathbf{I}_n)\right)$. We employ Couillet et al. (2011, Theorem 1) to write the Stieltjes transform of $\mathbf{L}_0(\mathbf{z}_0)$ as

$$
m_{\mathbf{L}_0}(z;\mathbf{z}_0) = \frac{1}{n}\text{Tr}\left(\mathbf{S}_0+\frac{z_0^{(0)}}{1+z_0^{(0)}\gamma_0 m_{\mathbf{L}_0}(z;\mathbf{z}_0)}\mathbf{I}_n-z\mathbf{I}_n\right)^{-1} \tag{16}
$$

$$
= \frac{1}{n}\text{Tr}\left(z_{-1}\mathbf{I}_n+s_0^{-1}(z)\mathbf{I}_n-z\mathbf{I}_n\right)^{-1} \tag{17}
$$

$$
s_0(z) = \frac{1+z_0^{(0)}\gamma_0 m_{\mathbf{L}_0}(z;\mathbf{z}_0)}{z_0^{(0)}}.
$$

We then employ Lemma 4 to compute the Shannon transform of $m_{\mathcal{L}_0}(z; \mathbf{z}_0)$,

$$\mathcal{V}_0(x) \to \frac{1}{n} \log \det \left( \mathbf{I}_n + \frac{1}{x} \left( z_{-1}^{(0)} \mathbf{I}_n + s_0^{-1}(-x) \mathbf{I}_n \right) \right) + \frac{1}{\gamma_0} \ln s_0(-x) + \frac{1}{z_0^{(0)} \gamma_0 s_0(-x)} + \frac{1}{\gamma_0} \ln z_0^{(0)} - \frac{1}{\gamma_0}$$

$$= \ln \left( 1 + \frac{s_0(-x) + z_{-1}^{(0)}}{x} \right) + \frac{1}{\gamma_0} \ln s_0(-x) + \frac{1}{z_0^{(0)} \gamma_0 s_0(-x)} - \gamma_0^{-1} + \frac{1}{\gamma_0} \ln z_0^{(0)}$$

We can explicitly compute $s_0$ by noting from (16) that

$$m_{\mathbf{L}_0}(x; \mathbf{z}_0) = \left( z_{-1}^{(0)} + \frac{z_0^{(0)}}{1 + z_0^{(0)} \gamma_0 m_{\mathbf{L}_0}(x; \mathbf{z}_0)} - x \right)^{-1}$$

$$z_0^{(0)} m_{\mathbf{L}_0}(x; \mathbf{z}_0) = \left( \frac{z_{-1}^{(0)} - x}{z_0^{(0)}} + \frac{1}{1 + \gamma_0 z_0^{(0)} m_{\mathbf{L}_0}(x; \mathbf{z}_0)} \right)^{-1}.$$

The solution of this is the same as the solution of the Marchenko-Pastur equation,, i.e. the Stieltjes transform $m_0$ of the Marchenko-Pastur law (Couillet & Debbah, 2011, Page 51.):

$$m_0(z) = \left( -z + \frac{1}{1 + \gamma_0 m_0(z)} \right)^{-1}, \quad z = -\frac{z_{-1} - x}{z_0}.$$

Hence,

$$s_0(-x) = \frac{1}{z_0^{(0)}} + \frac{\gamma_0}{z_0^{(0)}} m_0\left( -\frac{z_{-1}^{(0)} + x}{z_0^{(0)}} \right) = \frac{1}{z_0^{(0)}} + \gamma_0 t_0(\mathbf{z}_0^{-x}, (1, 0)).$$

Hence, by Theorem 4,

$$D_L(x, \mathbf{z}_L) = \sum_{l=0}^{L} C_l(x, \mathbf{z}_l, s_l(\mathbf{z}_l^{-x})) + \ln(x + 1/s_0(\mathbf{z}_0^{-x}) + z_{-1}^{(0)}).$$

$\square$

## A.3. Expression for Limiting BFE

We can now present the limiting BFE for the CK and NTK, under the assumptions in Assumption 1. For the CK, let $m_K = m_{\text{CK}}$ and $\mathbf{z}_L = (0, 0, \dots, 0, 1)$, and for the NTK, let $m_K = m_{\text{NTK}}$ and $\mathbf{z}_L = (r_+, q_0, \dots, q_{L-1}, 1)$. Under $(\epsilon, B)-$orthonormal input data $\mathbf{X}$, with Shannon-transform $\mathcal{V}_0$, by Theorem 4 we have that

$$\mathcal{F}_\infty^{\tau, \lambda} = \frac{\lambda}{2} m_K(-\lambda\tau) + \frac{1}{2} \sum_{l=1}^{L} C_l(x, \mathbf{z}_l, s_l(\mathbf{z}_l^{-x})) + \frac{1}{2} \mathcal{V}_0 \left( \frac{x + z_{-1}^{(0)}}{z_0^{(0)}} \right) + \frac{1}{2} \ln \left( 1 + \frac{z_{-1}^{(0)}}{x} \right) - \frac{1}{2} \ln \frac{\lambda}{2\pi}. \qquad \text{(General)}$$

For input $\mathbf{X} = d_0^{-1/2} \mathbf{Z}$, with $\mathbf{Z}$ having $i.i.d.$ columns that are zero mean and unit covariance, we have by Lemma 5 that

$$\mathcal{F}_\infty^{\tau, \lambda} = \frac{\lambda}{2} m_K(-\lambda\tau) + \frac{1}{2} \sum_{l=0}^{L} C_l(x, \mathbf{z}_l, s_l(\mathbf{z}_l^{-x})) + \frac{1}{2} \ln(x + 1/s_0(\mathbf{z}_0^{-x}) + z_{-1}^{(0)}) - \frac{1}{2} \ln \frac{\lambda}{2\pi}. \qquad \text{(MP)}$$

A method of evaluating $s_L, s_{L-1}, \dots, s_1$, and hence (General), is given in (Fan & Wang, 2020, Appendix A).

We numerically evaluate the limiting BFE for Gaussian input data, i.e. (MP). We use 1000 input vectors, and take $\lambda = 0.5, \gamma = 0.1$. Layer-widths were homogeneous across the network layers, i.e. $d_0 = d_1 = \cdots = d_L = d$. We use 20 log-spaced layer-widths between $d = 200$ and $d = 10000$. Mean and $95\%$ confidence intervals are plotted from $N = 10$ repeats. Code can be found here. The results are presented in Figure 7. We see in Figure 7 that the limiting energy shows great agreement with the random empirical quantity, for both the NTK and the CK.

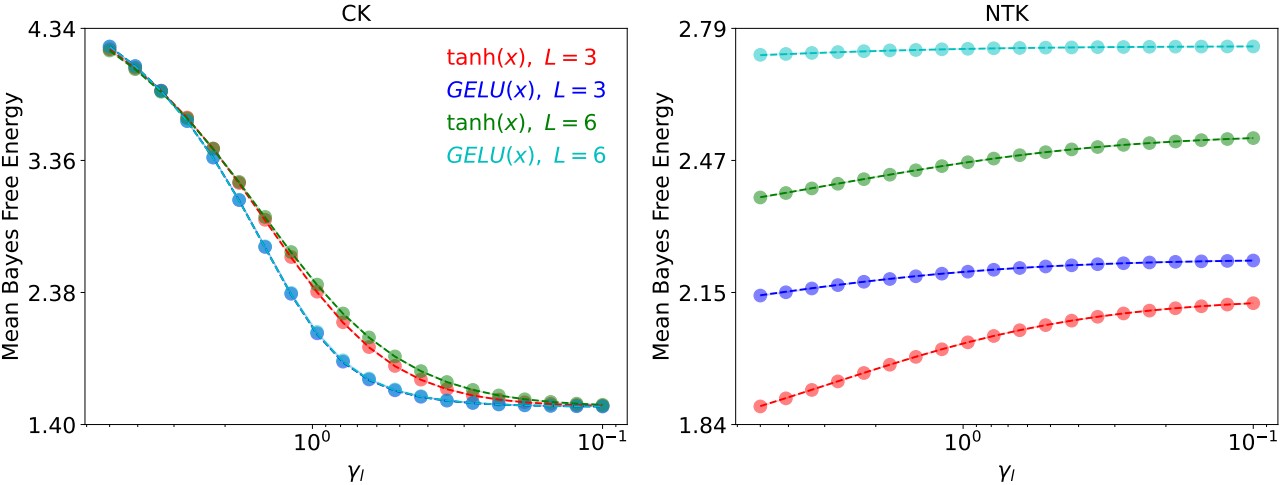

*Figure 7.* BFE for (left) CK and (right) NTK for normalized Gaussian input and output data, as a function of layer width $\gamma_l = \gamma_0 = \cdots = \gamma_L$, for $\tau = 0.1, \lambda = 0.5$. The empirical BFE is calculated with Gaussian input data, and is represented with circles, while the limiting BFE in (MP) is represented with dashed lines.

### A.4. Bound on Free Energy

**Lemma 6.** *For input $X_0 = d_0^{-1/2} Z$, with $Z$ having i.i.d. columns that are zero mean and with unit covariance,*

$$\mathcal{F}_\infty^{\tau,\lambda} > \min \mathcal{F}_\infty^{\tau,\lambda}.$$

*Proof.* Note that for a normalized Gaussian input, the free energy can be written as

$$\mathcal{F}_\infty^{\tau,\lambda} = \frac{\lambda}{2} m_K(\lambda\tau) + \frac{1}{2} \left[ \sum_{l=0}^{L} \mathcal{C}_l(x, \mathbf{z}_l, s_l(\mathbf{z}_l^{-x})) + \ln(x + 1/s_0(\mathbf{z}_0^{-x}) + z_{-1}) \right] - \frac{1}{2} \ln(\frac{\lambda}{2\pi}) \tag{18}$$

$$= \frac{\lambda}{2} t_0(\mathbf{z}_0^{-\lambda\tau}) + \frac{1}{2} \ln(x + 1/s_0(\mathbf{z}_0^{-x}) + z_{-1}) \tag{19}$$

$$+ \frac{1}{2} \sum_{l=0}^{L} \frac{1}{\gamma_l} \left( \frac{1}{s_l(\mathbf{z}_l^{-\lambda\tau}) \left[ \mathbf{z}_l^{-\lambda\tau} \right]_l} - \ln \left( \frac{1}{s_l(\mathbf{z}_l^{-\lambda\tau}) \left[ \mathbf{z}_l^{-\lambda\tau} \right]_l} \right) - 1 \right) - \frac{1}{2} \ln(\frac{\lambda}{2\pi}). \tag{20}$$

Now by the Marchenko-Pastur law:

$$m_0(z) = \left( -z + \frac{1}{1 + \gamma_0 m_0(z)} \right)^{-1}$$

$$\frac{1}{z_0} m_0(-\frac{x + z_{-1}}{z_0}) = \left( x + z_{-1} + \frac{1}{\frac{1}{z_0} + \gamma_0 \frac{1}{z_0} m_0(-\frac{x + z_{-1}}{z_0})} \right)^{-1}$$

$$t_0(\mathbf{z}_0^{-\lambda\tau}) = \left( x + 1/s_0(\mathbf{z}_0^{-x}) + z_{-1} \right)^{-1}$$

where we have used the previous definitions of $\mathbf{z}_l, t_0, s_0$. Hence, the free energy can be written as:

$$\mathcal{F}_\infty^{\tau,\lambda} = \frac{1}{2} \left[ \lambda t_0(\mathbf{z}_0^{-\lambda\tau}) - \ln(\lambda t_0(\mathbf{z}_0^{-\lambda\tau})) \right.$$

$$\left. + \frac{1}{2} \sum_{l=0}^{L} \frac{1}{\gamma_l} \left( \frac{1}{s_l(\mathbf{z}_l^{-\lambda\tau}) \left[ \mathbf{z}_l^{-\lambda\tau} \right]_l} - \ln \left( \frac{1}{s_l(\mathbf{z}_l^{-\lambda\tau}) \left[ \mathbf{z}_l^{-\lambda\tau} \right]_l} \right) - 1 \right) \right] + \frac{1}{2} \ln(2\pi).$$

Note that we have terms in the form of $x - \ln x$, which reaches a minimum of 1 at $x = 1$. We also note that

$$s_l(\mathbf{z}_l^{-\lambda\tau}) z_l^{(l)} = 1 + z_l^{(l)} \gamma_l t_{l-1}(\mathbf{z}_{l-1}, (1 - b_\sigma^2, 0, \ldots, 0, b_\sigma^2)).$$

We also appeal to (Couillet et al., 2011)[Theorem 1] to see that $t_l(\mathbf{z}_l^{-x}, \mathbf{w}) > 0$ for $x > 0$. Further, as $s_l > 0$ (by the definition of $s_l$ given in (Fan & Wang, 2020)), then $z_l^{(l)} > 0$, by the definition of $s_l$. Then,

$$s_l(\mathbf{z}_l^{-\lambda\tau}) \left[ \mathbf{z}_l^{-\lambda\tau} \right]_l > 1.$$

Hence,

$$\mathcal{C}_l(x, \mathbf{z}_l, s_l(\mathbf{z}_l^{-x})) > 0.$$

Therefore, we see that

$$\mathcal{F}_\infty^{\tau,\lambda} > \min \mathcal{F}_\infty^{\tau,\lambda}.$$

$\square$

### A.5. Under- and Over-Parameterized Models

By taking $\gamma_l \to 0$ and $\gamma_l \to \infty$ for $l = 0, 1, \ldots, L$, (i.e. very over- or under-parameterized models), we can derive closed form solutions for the limiting energy.

**Theorem 5.** *For input* $\mathbf{X} = d_0^{-1/2}\mathbf{Z}$*, with* $\mathbf{Z}$ *having i.i.d. columns that are zero mean and unit covariance, the limiting BFE* $\mathcal{F}_\infty^{\tau,\lambda}$ *converges to an explicit expression as* $\gamma_l \to 0$ *or* $\gamma_l \to \infty$*, for* $l = 0, 1, \ldots, L$*. For the CK, the expressions are given by*

$$\lim_{\gamma_0, \gamma_1, \ldots, \gamma_L \to 0} \mathcal{F}_\infty^{\tau,\lambda} := \mathcal{F}_{\infty,0}^{\tau,\lambda} = \frac{1}{2} \left( \frac{\lambda}{1 + \lambda\tau} - \ln \frac{\lambda}{1 + \lambda\tau} + \ln 2\pi \right)$$

$$\lim_{\gamma_0, \gamma_1, \ldots, \gamma_L \to \infty} \mathcal{F}_\infty^{\tau,\lambda} := \mathcal{F}_{\infty,\infty}^{\tau,\lambda} = \frac{1}{2} \left( \frac{1}{\tau} - \ln \frac{1}{\tau} + \ln 2\pi \right),$$

*while for the NTK,*

$$\lim_{\gamma_0, \gamma_1, \ldots, \gamma_L \to 0} \mathcal{F}_\infty^{\tau,\lambda} := \mathcal{F}_{\infty,0}^{\tau,\lambda} = \frac{1}{2} \left( \frac{\lambda}{\sum_{i=0}^L a_\sigma^i + \lambda\tau} - \ln \frac{\lambda}{\sum_{i=0}^L a_\sigma^i + \lambda\tau} + \ln 2\pi \right)$$

$$\lim_{\gamma_0, \gamma_1, \ldots, \gamma_L \to \infty} \mathcal{F}_\infty^{\tau,\lambda} := \mathcal{F}_{\infty,\infty}^{\tau,\lambda} = \frac{1}{2} \left( \frac{\lambda}{\lambda\tau + r_+} - \ln \frac{\lambda}{\lambda\tau + r_+} + \ln 2\pi \right).$$

*Proof.* We first analyze the case where $\gamma_l \to 0$, $l = 0, 1, \ldots, L$. For $\mathbf{z}_l = (z_{-1}, z_0, \ldots, z_l)$,

$$s_l(\mathbf{z}_l) = \frac{1}{z_l} + \gamma_l t_{l-1} \left( z_{-1} + \frac{1 - b_\sigma^2}{s_l(\mathbf{z}_l)}, z_0, \ldots, z_{l-2}, z_{l-1} + \frac{b_\sigma^2}{s_l(\mathbf{z}_l)} \right).$$

By (Fan & Wang, 2020, Proposition 3.6), any solution $s_l^*$ in $\mathbb{C}^+$ to (8) is the unique solution. Hence, we only need to find *a* solution. We guess that as $\gamma_l \to 0$, then $s_l \to 1/z_l$. Then,

$$\mathbf{z}_{\text{prev}}(s_l(\mathbf{z}_l), \mathbf{z}_l) \to (z_{-1} + (1 - b_\sigma^2)z_l, z_0, \ldots, z_{l-2}, z_{l-1} + (b_\sigma^2)z_l).$$

This is constant in $\gamma_l$, and hence the RHS of (8) converges to $1/z_l$, for $l = 1, \ldots, L$. Further, $s_0(z_{-1}, z_0) = 1/z_0 + \gamma_0 t_0((z_{-1}, z_0), (1, 0)) \to 1/z_0$.

Denote by $\mathbf{z}_{l-1} = \mathbf{z}_{\text{prev}}(s_l(\mathbf{z}_l), \mathbf{z}_l)$, $l = 0, \ldots, L$. Then, if $s_l \to 1/z_l$, for $l = 0, \ldots, L$, we have that

$$\mathbf{z}_L = (z_{-1}, z_0, \ldots, z_L)$$
$$\mathbf{z}_{L-1} \to (z_{-1} + (1 - b_\sigma^2)z_L, z_0, \ldots, z_{L-1} + b_\sigma^2 z_L)$$
$$\mathbf{z}_{L-2} \to (z_{-1} + (1 - b_\sigma^2)z_L + (1 - b_\sigma^2)(z_{L-1} + b_\sigma^2 z_L), z_0, \ldots, z_{L-2} + b_\sigma^2(z_{L-1} + b_\sigma^2 z_L)).$$

We can generalize this pattern to $\mathbf{z}_0$:

$$\mathbf{z}_0 \to \left( z_{-1} + (1 - b_\sigma^2) \sum_{j=1}^{L} \sum_{i=0}^{j-1} z_{L-i}(b_\sigma^2)^{j-1-i}, \sum_{i=0}^{L} z_i(b_\sigma^2)^i \right).$$

For the CK, $\mathbf{z}_L = (0, \ldots, 0, 1)$, and hence,

$$\mathbf{z}_0^{\mathrm{CK}} \to ((1 - b_\sigma^2) \sum_{j=0}^{L-1} (b_\sigma^2)^j, b_\sigma^{2L}) = (1 - b_\sigma^{2L}, b_\sigma^{2L}) = (z_{-1,\mathrm{CK}}^*, z_{0,\mathrm{CK}}^*).$$

For the NTK, $\mathbf{z}_L = (r_+, q_0, \ldots, q_{L-1}, 1)$, and so $\mathbf{z}_0^{\mathrm{NTK}} \to (z_{-1,\mathrm{NTK}}^*, z_{0,\mathrm{NTK}}^*)$, where some calculations show

$$z_{0,\mathrm{NTK}}^* = \sum_{i=0}^{L} q_i(b_\sigma^2)^i = \sum_{i=0}^{L} (b_\sigma^2)^{L-i}(b_\sigma^2)^i = (L+1)b_\sigma^{2L}$$

$$z_{-1,\mathrm{NTK}}^* = r_+ + (1 - b_\sigma^2) \sum_{j=1}^{L} \sum_{i=0}^{j-1} q_{L-i}(b_\sigma^2)^{j-1-i}$$

$$= r_+ + (1 - b_\sigma^2) \sum_{j=1}^{L} \sum_{i=0}^{j-1} (b_\sigma^2)^i (b_\sigma^2)^{j-1-i}$$

$$= r_+ + (1 - b_\sigma^2) \sum_{j=1}^{L} \sum_{i=0}^{j-1} (b_\sigma^2)^{j-1}$$

$$= r_+ + (1 - b_\sigma^2) \sum_{j=1}^{L} j(b_\sigma^2)^{j-1}$$

$$= r_+ + \sum_{j=1}^{L} j(b_\sigma^2)^{j-1} - \sum_{j=1}^{L} j(b_\sigma^2)^j$$

$$= r_+ + 1 + \sum_{j=1}^{L-1} (j+1)(b_\sigma^2)^j - \sum_{j=1}^{L-1} j(b_\sigma^2)^j - L(b_\sigma^{2L})$$

$$= r_+ + 1 + \sum_{j=1}^{L-1} (b_\sigma^2)^j - L(b_\sigma^{2L})$$

$$= \sum_{j=0}^{L-1} a_\sigma^{L-j} - \sum_{j=0}^{L-1} (b_\sigma^2)^{L-j} + 1 + \sum_{j=1}^{L-1} (b_\sigma^2)^j - Lb_\sigma^{2L}$$

$$= \sum_{j=0}^{L} a_\sigma^j - (L+1)b_\sigma^{2L}$$

The Stieltjes transform for both the CK and NTK can be written as $m_K(z) = t_L(\mathbf{z}_L^z) = t_0(\mathbf{z}_0^{\mathrm{K}}, (1, 0)) = m_0(-(z_{-1,\mathrm{K}}^* - z)/z_{0,\mathrm{K}}^*)/z_{0,\mathrm{K}}^*$. Now, by the assumption on $\mathbf{X}$, we have that $m_0$ follows the Marchenko-Pastur law. We have the following relationship for the Marchenko-Pastur law,

$$m_0(z) = \frac{1}{1 - \gamma_0 - z - \gamma_0 z m_0(z)}.$$

Hence, as $\gamma_0 \to 0, m_0(z) \to 1/(1 - z)$. Thus,

$$t_0(\mathbf{z}_0^z, (1, 0)) = \frac{1}{z_0^*} \cdot \frac{1}{1 + (z_{-1}^* - z)/z_0^*} = \frac{1}{z_{-1}^* + z_0^* - z}.$$

We now consider the log-determinant term, $D_l(x, \mathbf{z}_l)$. First consider the constant term:

$$C_l(x, \mathbf{z}_l, s_l(\mathbf{z}_l^{-x})) = \frac{1}{\gamma_l} \left( \ln s_l(\mathbf{z}_l^{-x}) z_l + \frac{1}{s_l(\mathbf{z}_l^{-x}) z_l} - 1 \right)$$

$$\rightarrow \frac{1}{\gamma_l} \left( \ln \frac{z_l}{z_l} + \frac{1}{\frac{z_l}{z_l}} - 1 \right) = 0, \quad l = 0, 1, \ldots, L.$$

We are then left with $D_{-1}(x, \mathbf{z}_0)$, i.e.

$$\ln \left( x + 1/s_0(\mathbf{z}_0^{-x}) + z_{-1}^* \right) = \ln(z_{-1}^* + z_0^* - x).$$

We can then substitute in the respective values for CK and NTK to get the limiting energy in the case where $\gamma_l \rightarrow 0$.

We now analyze the case when $\gamma_l \rightarrow \infty$, $l = 0, \ldots, L$. For $\gamma_l \rightarrow \infty$, $l = 0, 1, \ldots, L$, we guess that $s_l(\mathbf{z}_l) \rightarrow \infty$. Under this,

$$\mathbf{z}_{\text{prev}}(s_l(\mathbf{z}_l), \mathbf{z}_l) \rightarrow (z_{-1}, z_0, \ldots, z_{l-1}),$$

i.e. a constant in $\gamma_l$. By definition (Fan & Wang, 2020), $t_l(\mathbf{z}_l) \in \mathbb{R}^+$ for $\mathbf{z}_l \in \mathbb{R}_+^{l+2}$ (we note that for both the CK and NTK, and if $s_l(\mathbf{z}_l) \rightarrow \infty$, then $\mathbf{z}_l \in \mathbb{R}_+^{l+2}$). Thus, as $\gamma_l \rightarrow +\infty$, $\gamma_l t_l(\mathbf{z}_{\text{prev}}(s_l(\mathbf{z}_l), \mathbf{z}_l)) \rightarrow +\infty$. Hence, if $s_l(\mathbf{z}_l) \rightarrow \infty$, then the LHS of (8) converges to the RHS of (8).

Now, for $s_l(\mathbf{z}_l) \rightarrow \infty$, we have for $\mathbf{z}_L = (z_{-1}, z_0, \ldots, z_L)$ that

$$\mathbf{z}_0 \rightarrow (z_{-1}, z_0).$$

To characterise the Stieltjes transform, we need to analyze the MP-law for $\gamma_0 \rightarrow \infty$:

$$m_0(z) = \frac{1 - \gamma_0 - z - \sqrt{(z - \gamma_0 - 1)^2 - 4\gamma_0}}{2z\gamma_0}$$

$$= \frac{1}{2z} \left( \frac{1}{\gamma_0} - 1 - \frac{z}{\gamma_0} - \sqrt{\frac{1}{\gamma_0^2}(z^2 - 2z(\gamma_0 + 1) + \gamma_0^2 + 2\gamma_0 + 1 - 4\gamma_0)} \right)$$

$$\rightarrow -\frac{1}{z}.$$

Hence, $m_K(z) = t_L(\mathbf{z}_L^z) = t_0(\mathbf{z}_0, (1, 0)) = m_0(-(z_{-1} - z)/z_0)/z_0 = 1/(z_{-1} - z)$.

For the log-determinant term, the constant term is given by

$$C_l(x, \mathbf{z}_l, s_l(\mathbf{z}_l^{-x})) = \frac{1}{\gamma_l} \left( \ln s_l(\mathbf{z}_l^{-x}) z_l - \frac{1}{s_l(\mathbf{z}_l^{-x}) z_l} - 1 \right).$$

We note that $\ln x$ will grow slower than $x$ for $x > 0$, and so as both $\gamma_l$ and $s_l(\mathbf{z}_l^{-x})$ grow, $C_l \rightarrow 0$. Hence,

$$D_L(x, \mathbf{z}_L) \rightarrow \ln(x + z_{-1}).$$

Again, substituting the respective values for the CK and NTK will give the BFE as $\gamma_l \rightarrow 0$, $l = 0, 1, \ldots, L$. $\qquad \square$

**Remark 1** (Equivalence of Linear and Conjugate Kernels). *Note that the limiting expressions for the CK in Theorem 5 are not functions of $L$. Hence, we can make the following conclusion: the BFE for the linear and conjugate kernel is asymptotically equivalent as $\gamma_l \rightarrow 0$ or $\gamma_l \rightarrow \infty$. However, we see that the BFE for the NTK is layer-dependent.*

Due to the closed-form expressions in Theorem 5, we can characterize the behavior of these expressions in terms of $\lambda$ and $\tau$:

**Lemma 7.** *When $\tau < 1$, there exists $\lambda_0^*$ that minimizes $\mathcal{F}_{\infty,0}^{\tau,\lambda}$ for the CK and NTK, and $\lambda_\infty^*$ that minimizes $\mathcal{F}_{\infty,\infty}^{\tau,\lambda}$ for the NTK. Specifically, for the CK,*

$$\lambda_0^* = \frac{1}{1-\tau},$$

*and for the NTK,*

$$\lambda_0^* = \frac{\sum_{i=0}^{L} a_\sigma^i}{1-\tau}, \quad \lambda_\infty^* = \frac{r_+}{1-\tau}.$$

*When $\tau \geq 1$, there is no optimal $\lambda^*$. In this case, $\mathcal{F}_{\infty,0}^{\tau,\lambda}$ is decreasing in $\lambda$ for the CK and NTK, as well as $\mathcal{F}_{\infty,\infty}^{\tau,\lambda}$ for the NTK. Furthermore, when $\tau > 1$, then $\mathcal{F}_{\infty,0}^{\tau,\lambda} > \mathcal{F}_{\infty,\infty}^{\tau,\lambda}$ for the CK.*

*Proof.* The proof of this lemma is immediate upon noting that $x - \ln x$ is minimized at $x = 1$. $\qquad \square$

### A.6. Proof of Theorem 1

For the limiting expression for the BFE, under Assumption 1, we take the result of Theorem 4, where the explicit form for the BFE is given in (General). The lower bound on the BFE is given in Lemma 6. For the convergence of the BFE to its lower bound, we note that by Theorem 5, the limiting form in Theorem 4 converges to explicit expressions as $\gamma \to 0$ and $\gamma \to \infty$. Further, by Lemma 7, these explicit expressions are optimized, at which point they attain the value in Lemma 6, for the optimal regularizers given in Lemma 7. Further, we can see from Theorem 5, that the BFE for the CK diverges for $\gamma \to \infty, \tau \to 0$.

### A.7. Strong Descent & Feature-Attainment

We now prove Theorem 2:

*Proof of Theorem 2.* Note that by Lemma 7 and Theorem 5, when $\tau < 1$, $\mathcal{F}_{\infty,0}^{\tau,\lambda_0^*}$ attains the minimum free energy from Lemma 6, for both the CK and the NTK. Hence, for strong-descent, we simply need to choose $\lambda(\gamma)$ such that $\mathcal{F}_{\infty,\infty}^{\tau,\lambda(\gamma)} > \frac{1+\log(2\pi)}{2}$. Then, by Theorem 5, $\mathcal{F}_\infty^{\tau,\lambda(\gamma)}$ will approach this limiting bound as $\gamma \to 0$, and *strong-descent* will have occurred. For $\gamma \geq 1$, neither $\mathcal{F}_{\infty,0}^{\tau,\lambda^*}$ nor $\mathcal{F}_{\infty,\infty}^{\tau,\lambda^*}$ can attain the bound in Lemma 6, by Lemma 7. $\qquad \square$

### A.8. Weak Descent

In this section we will present and give the proof of Lemma 8:

**Lemma 8** (Weak Descent). *If $\tau < 1$, then weak descent will occur for the CK if*

$$\lambda(\tau)_{CK} > -\frac{W_0\left(-\frac{1}{\tau}e^{-\frac{1}{\tau}}\right)}{1 + \tau W_0\left(-\frac{1}{\tau}e^{-\frac{1}{\tau}}\right)},$$

*where $W_0(.)$ is the principal branch of Lambert's W function. For weak descent to occur for the NTK,*

$$\lambda(\tau)_{NTK} > \frac{r_+ + \sum_{i=1}^{L} a_\sigma^i}{r_+ - \sum_{i=1}^{L} a_\sigma^i} \ln\left(\frac{r_+}{\sum_{i=1}^{L} a_\sigma^i}\right)$$

$$\lambda(\tau)_{NTK} > \lambda(\tau)_{CK}.$$

*If $\tau \geq 1$, then weak descent will not occur for either the CK or the NTK.*

*Proof of Lemma 8.* For weak descent to occur, we require $\mathcal{F}_{\infty,0}^{\tau,\lambda} < \mathcal{F}_{\infty,\infty}^{\tau,\lambda}$. We note that when $\tau \geq 1$, $\lambda/(C + \lambda\tau) < 1$ for $C > 0, \lambda > 0$. This corresponds to $0 < x < 1$ for the function $g(x) = x - \ln x$. In this domain, $g(x)$ is decreasing. As $\mathcal{F}_{\infty,0}^{\tau,\lambda} = g(\lambda/(C_0 + \lambda\tau))$, and $\mathcal{F}_{\infty,\infty}^{\tau,\lambda} = g(\lambda/(C_\infty + \lambda\tau))$, for $C_0 > C_\infty$ (where specific values are dependent on kernel

function, yet ordering holds), and as $\lambda/(C_0 + \lambda\tau) < \lambda/(C_\infty + \lambda\tau)$, it is clear that when $\tau \geq 1$, $\mathcal{F}_{\infty,0}^{\tau,\lambda} > \mathcal{F}_{\infty,\infty}^{\tau,\lambda}$.

For the weak-descent bound on $\lambda$ for the CK, we are trying to find $\lambda$ such that $\mathcal{F}_{\infty,0}^{\tau,\lambda} < \mathcal{F}_{\infty,\infty}^{\tau,\lambda}$, i.e.

$$\frac{\lambda}{1+\lambda\tau} - \ln\left(\frac{\lambda}{1+\lambda\tau}\right) < \frac{1}{\tau} - \ln\left(\frac{1}{\tau}\right)$$

$$e^{\frac{\lambda}{1+\lambda\tau}} e^{-\frac{1}{\tau}} < \frac{\tau\lambda}{1+\lambda\tau}$$

$$-\frac{\tau\lambda}{1+\lambda\tau} e^{-\frac{\lambda}{1+\lambda\tau}} < -\frac{1}{\tau} e^{-\frac{1}{\tau}}.$$

As $z \mapsto z e^z$ is an increasing function for $z > -1$, than if $z e^z > C$, $z > W(C)$, where $W$ is the Lambert W function. Hence,

$$-\frac{\tau\lambda}{1+\lambda\tau} = W(-\frac{1}{\tau} e^{-\frac{1}{\tau}}).$$

This can be rearranged to give the condition:

$$\lambda(\tau) > -\frac{W(-\frac{1}{\tau} e^{-\frac{1}{\tau}})}{1 + \tau W(-\frac{1}{\tau} e^{-\frac{1}{\tau}})}$$

where $\lambda(\tau)$ is a value, given $\tau$, such that if $\lambda > \lambda(\tau)$, weak descent occurs. We note that for $x \leq -1$, $W_{-1}(x e^x) = x$. As $\tau < 1$, this would result in dividing by zero in the above expression. Hence, $W(-\frac{1}{\tau} e^{-\frac{1}{\tau}}) = W_0(-\frac{1}{\tau} e^{-\frac{1}{\tau}})$, i.e. the principal branch. For $\tau < 1$, this is well-defined.

We now give the bound for the NTK. Firstly, as $\tau \to 0$, $\lambda/(a + \lambda\tau) \to \lambda/a$. Hence, for $\mathcal{F}_{\infty,0}^{\tau,\lambda} < \mathcal{F}_{\infty,\infty}^{\tau,\lambda}$,

$$\frac{\lambda}{\sum_{i=0}^{L} a_i} - \ln\left(\frac{\lambda}{\sum_{i=0}^{L} a_i}\right) < \frac{\lambda}{r_+} - \ln\left(\frac{\lambda}{r_+}\right)$$

$$\lambda < \frac{r_+ \sum_{i=0}^{L} a_i}{r_+ - \sum_{i=0}^{L} a_i} \ln\left(\frac{r_+}{\sum_{i=0}^{L} a_i}\right).$$

For this to be a bound for all $\tau \in (0,1)$, we require that the bound $\lambda(\tau)$ such that $\mathcal{F}_{\infty,0}^{\tau,\lambda(\tau)} < \mathcal{F}_{\infty,\infty}^{\tau,\lambda(\tau)}$ be increasing in $\tau$; then, the bound $\lambda(0)$ is a conservative bound for all $\tau \in (0,1)$. We now show that $\lambda(\tau)$ is increasing.

Note that $\mathcal{F}_{\infty,0}^{\tau,\lambda(\tau)} < \mathcal{F}_{\infty,\infty}^{\tau,\lambda(\tau)}$ is equivalent to $g(h_{\sum_{i=0}^{L} a_i}(\lambda,\tau)) < g(h_{r_+}(\lambda,\tau))$, where $g$ is defined as above, and $h_C(\lambda,\tau) = \lambda/(C + \lambda\tau)$. Also note that if $g(h_{\sum_{i=0}^{L} a_i}(\lambda,\tau)) = g(h_{r_+}(\lambda,\tau))$, with $h_{\sum_{i=0}^{L} a_i}(\lambda,\tau) \neq h_{r_+}(\lambda,\tau)$, then $h_{\sum_{i=0}^{L} a_i}(\lambda,\tau) \leq 1 \leq h_{r_+}(\lambda,\tau)$. In other words, if we denote $x_1 = h_{\sum_{i=0}^{L} a_i}(\lambda,\tau)$, and $x_2 = h_{r_+}(\lambda,\tau)$, then the line that intersects $g(x_1)$ and $g(x_2)$ has zero-gradient (as $g(x_1) = g(x_2)$), and $x_1$ and $x_2$ must be on either side (left and right respectively) of the minimum of $g(x)$ at $x = 1$, by the convexity of $g$.

Further, we also recognize that $h_C(\lambda,\tau)$ is an increasing function of $\lambda$, and a decreasing function of $\tau$, by the fact that

$$\frac{\partial}{\partial\lambda} h_C(\lambda,\tau) = \frac{C}{(C + \lambda\tau)^2} > 0, \quad \frac{\partial}{\partial\tau} h_C(\lambda,\tau) = -\frac{\lambda^2}{(C + \lambda\tau)^2} < 0.$$

We also have that $g$ is a decreasing function of $x$ for $0 < x < 1$, and an increasing function of $x$ for $x > 1$.

Say we have found some $\lambda_0, \tau_0$ such that $g(h_{\sum_{i=0}^{L} a_i}(\lambda_0, \tau_0)) = g(h_{r_+}(\lambda_0, \tau_0))$. Then increasing $\tau$, i.e. letting $\tau = \tau_0 + \epsilon$, $\epsilon > 0$, will decrease $h_{\sum_{i=0}^{L} a_i}(\lambda_0, \tau)$ and $h_{r_+}(\lambda_0, \tau)$. This will lead to a decrease in $g(h_{r_+}(\lambda_0, \tau))$ (as it is to the right

of $x = 1$) and an increase in $g(h_{\sum_{i=0}^{L} a_i}(\lambda_0, \tau))$ (as it is to the left of $x = 1$). To equate these again (remember we are trying to find the boundary of values such that $\mathcal{F}_{\infty,0}^{\tau,\lambda(\tau)} = \mathcal{F}_{\infty,\infty}^{\tau,\lambda(\tau)}$), we need to increase $\lambda$ to $\lambda = \lambda_0 + \delta$, $\delta > 0$, such that $g(h_{\sum_{i=0}^{L} a_i}(\lambda, \tau))$ will decrease, and $g(h_{r_+}(\lambda, \tau))$ will increase, for some $\delta$ such that these functions will equal again at $\lambda = \lambda_0 + \delta$, $\tau = \tau_0 + \epsilon$. As increasing $\lambda$ past this point will send $g(h_{\sum_{i=0}^{L} a_i}(\lambda, \tau))$ to a lower value than $g(h_{r_+}(\lambda, \tau))$, we see that the function $\lambda(\tau)$, which gives the values of $\lambda$, for a given $\tau$, such that $\mathcal{F}_{\infty,0}^{\tau,\lambda(\tau)} < \mathcal{F}_{\infty,\infty}^{\tau,\lambda(\tau)}$, is an increasing function of $\tau$.

For the comparison bound with the CK, we note that at $\lambda(\tau)_{\text{CK}}$,

$$\frac{\lambda(\tau)_{\text{CK}}}{1 + \tau\lambda(\tau)_{\text{CK}}} - \ln\left(\frac{\lambda(\tau)_{\text{CK}}}{1 + \tau\lambda(\tau)_{\text{CK}}}\right) = \frac{1}{\tau} - \ln\left(\frac{1}{\tau}\right).$$

As $1/\tau > 1$ for $\tau < 1$, this means that $\lambda(\tau)_{\text{CK}}/(1 + \tau\lambda(\tau)_{\text{CK}}) < 1$. We note that $\sum_{i=0}^{L} a_i > 1$ and $r_+ > 0$, by the definition of the constants, and the assumption that $b_\sigma \neq 0$. Hence,

$$\frac{\lambda(\tau)_{\text{CK}}}{\sum_{i=0}^{L} a_i + \tau\lambda(\tau)_{\text{CK}}} < \frac{\lambda(\tau)_{\text{CK}}}{1 + \tau\lambda(\tau)_{\text{CK}}} \quad \rightarrow \quad h_{\sum_{i=0}^{L} a_i}(\lambda(\tau)_{\text{CK}}, \tau) > h_1(\lambda(\tau)_{\text{CK}}, \tau)$$

$$\frac{\lambda(\tau)_{\text{CK}}}{r_+ + \tau\lambda(\tau)_{\text{CK}}} < \frac{1}{\tau} \quad \rightarrow \quad \begin{cases} h_{r_+}(\lambda(\tau)_{\text{CK}}, \tau) \leq h_1(\lambda(\tau)_{\text{CK}}, \tau) & \text{if } r_+ \leq 1 \\ h_{\sum_{i=0}^{L} a_i}(\lambda(\tau)_{\text{CK}}, \tau) > h_{r_+}(\lambda(\tau)_{\text{CK}}, \tau) > h_1(\lambda(\tau)_{\text{CK}}, \tau) & \text{if } r_+ > 1 \end{cases}$$

Hence, we have that at $\lambda(\tau)_{\text{CK}}$, $\mathcal{F}_{\infty,0,\text{NTK}}^{\tau,\lambda(\tau)_{\text{CK}}} > \mathcal{F}_{\infty,\infty,\text{NTK}}^{\tau,\lambda(\tau)_{\text{CK}}}$. As per the previous arguments, $\lambda(\tau)_{\text{NTK}}$ must increase in order for $\mathcal{F}_{\infty,0,\text{NTK}}^{\tau,\lambda(\tau)_{\text{NTK}}} < \mathcal{F}_{\infty,\infty,\text{NTK}}^{\tau,\lambda(\tau)_{\text{NTK}}}$. $\square$

This gives us a convenient method for comparing the CK and the NTK, i.e. comparing the area of parameters $\lambda(\tau)$ such that weak descent occurs. We find a bound on the difference in parameter volume,

$$\Delta A > \int_0^1 (\lambda(\tau)_{\text{NTK}} - \lambda(\tau)_{\text{CK}}) d\tau$$

$$= \frac{r_+ \sum_{i=1}^{L} a_\sigma^i}{r_+ - \sum_{i=1}^{L} a_\sigma^i} \ln\left(\frac{r_+}{\sum_{i=1}^{L} a_\sigma^i}\right)(\tau_0 - 1) - \int_0^{\tau_0} \lambda(\tau)_{\text{CK}} d\tau,$$

where $\tau_0$ is the value such that $(r_+ \sum_{i=1}^{L} a_\sigma^i) \ln\left(\frac{r_+}{\sum_{i=1}^{L} a_\sigma^i}\right)/(r_+ - \sum_{i=1}^{L} a_\sigma^i) = \lambda(\tau_0)_{\text{CK}}$. We compute both $\tau_0$ and the above integral numerically, and present the results in Appendix A.8 for different neural architectures. Note that $\Delta A > 0$ by Lemma 8.

| Architecture | 3-Layer Tanh | 5-Layer GeLU |
|---|---|---|
| $\Delta A$: | 2.11463 | 11.0042 |

*Table 3.* Difference in parameter volumes for which weak descent *doesn't occur*, for the CK versus NTK, for two neural architectures. A larger positive value corresponds to a less robust NTK kernel, in comparison to the CK.

## B. Connection Between BFE & UQ

Epistemic uncertainty quantification for Bayesian models arises from posterior variance; for a test input, there may be many functions with high posterior weight, yet they all give different outputs for this input. We note that marginal likelihood is the likelihood of seeing the training data under our prior distribution (i.e. the given kernel function), i.e.

$$\mathcal{Z}_n^{\tau,\lambda} = \int_f p(\mathbf{Y}|f, \mathbf{X}) \, p(f) \, df.$$

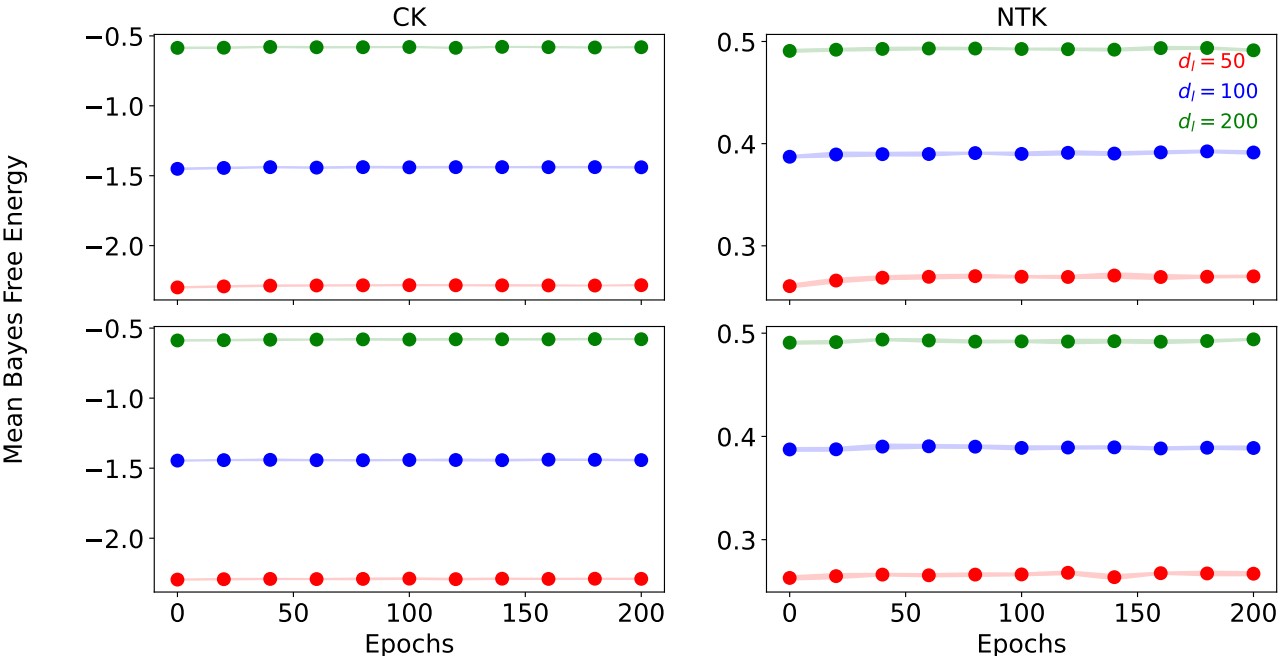

*Figure 8.* Log-determinant of $K_X + \tau\lambda$ as a function of epochs of training, for (top) Gaussian data with $\lambda = \lambda^*$ and (bottom) a teacher network $y = \sin(w^T x)$ with $\lambda = 1$, and $\tau = 10^{-3}$.

Therefore, if we assume we have a kernel function with large $\mathcal{Z}_n^{\tau,\lambda}$, then this means that at the time of posterior inference, functions with (relatively) high posterior weight will give a better fit to the training data, compared to posterior functions for a kernel function with small $\mathcal{Z}_n^{\tau,\lambda}$. Hence, the posterior functions for a kernel function with large $\mathcal{Z}_n^{\tau,\lambda}$ will more faithfully represent the possible functions one may expect to see given the training data. Thus, the variance in the outputs of these posterior function for a new test point will be a better estimate of epistemic uncertainty. The connection to BFE is then immediate. Note that this connection is not novel; see (Immer et al., 2021a; Daxberger et al., 2021).

## C. Free Energy After Training

We note that the BFE is the sum of two components: the data-fit term $(Y - m(X))^T (K_X + \tau\lambda)^{-1} (Y - m(X))$ and the log-determinant $\log\det(K_X + \tau\lambda)$. We plot the log-determinant, the data-fit, and the data-fit under $m(X) = 0$, in Figure 8, Figure 9, and Figure 10 respectively. We see that interestingly the log-determinant term does not change during training, and instead seems to be determined at initialization. In comparison, the data-fit goes to zero during training. Under a zero-mean prior, the data-fit still decreases.

We also display the BFE after training for MNIST and CIFAR-10, after normalization and whitening, in Figure 11. We again see the exact same relationship: the BFE for the CK decreases below that of the NTK after training.

Interestingly, we can also plot the free energy for the CK and NTK for a network that has been trained on various UCI regression datasets using Adam. In Figure 12, we see that eventually, the free energy for the CK drops below that for the NTK. Here, the network size is given in Table 1.

## D. Linear Sampling

To show that Algorithm 1 allows for sampling from Bayesian GLMs, we first consider the following optimization problem:

$$\min_{\boldsymbol{\theta}_S} \sum_{i=1}^{n} \ell_2(\tilde{\mathbf{f}}_{\hat{\boldsymbol{\theta}}}(\boldsymbol{\theta}_S, \mathbf{x}_i), \tilde{\mathbf{y}}_i), \tag{21}$$

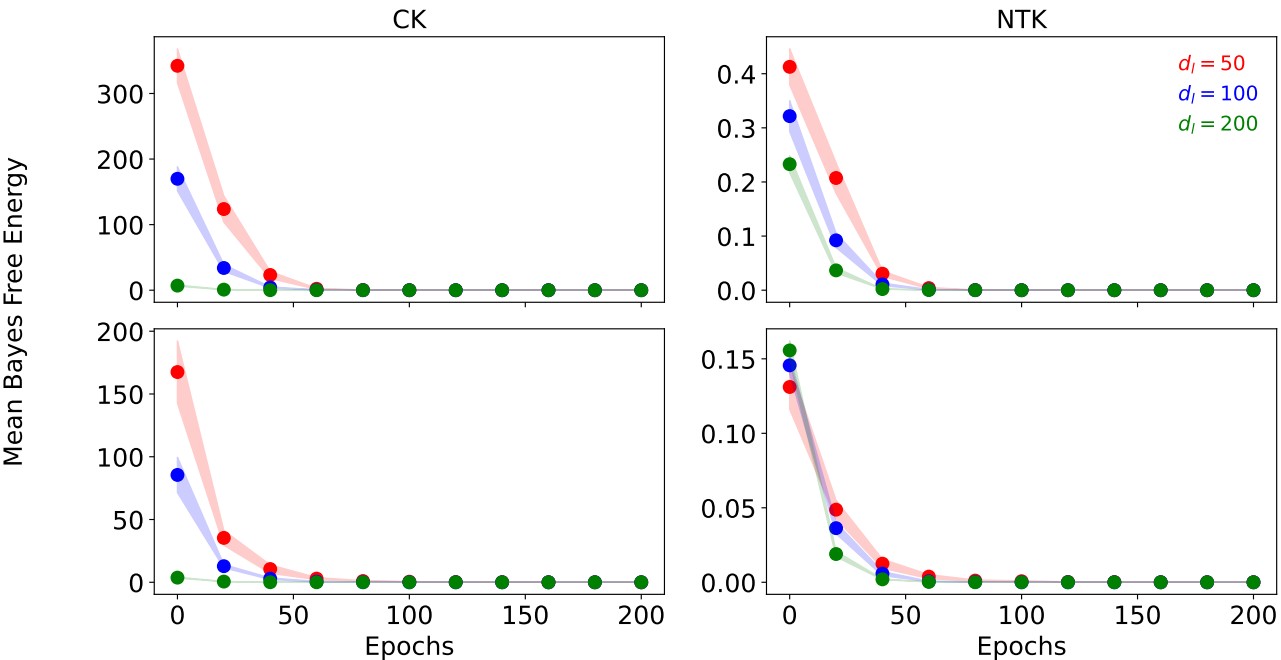

*Figure 9.* Data-fit term for $K_X + \tau\lambda$ as a function of epochs of training, for (top) Gaussian data with $\lambda = \lambda^*$ and (bottom) a teacher network $y = \sin(w^T x)$ with $\lambda = 1$, and $\tau = 10^{-3}$.

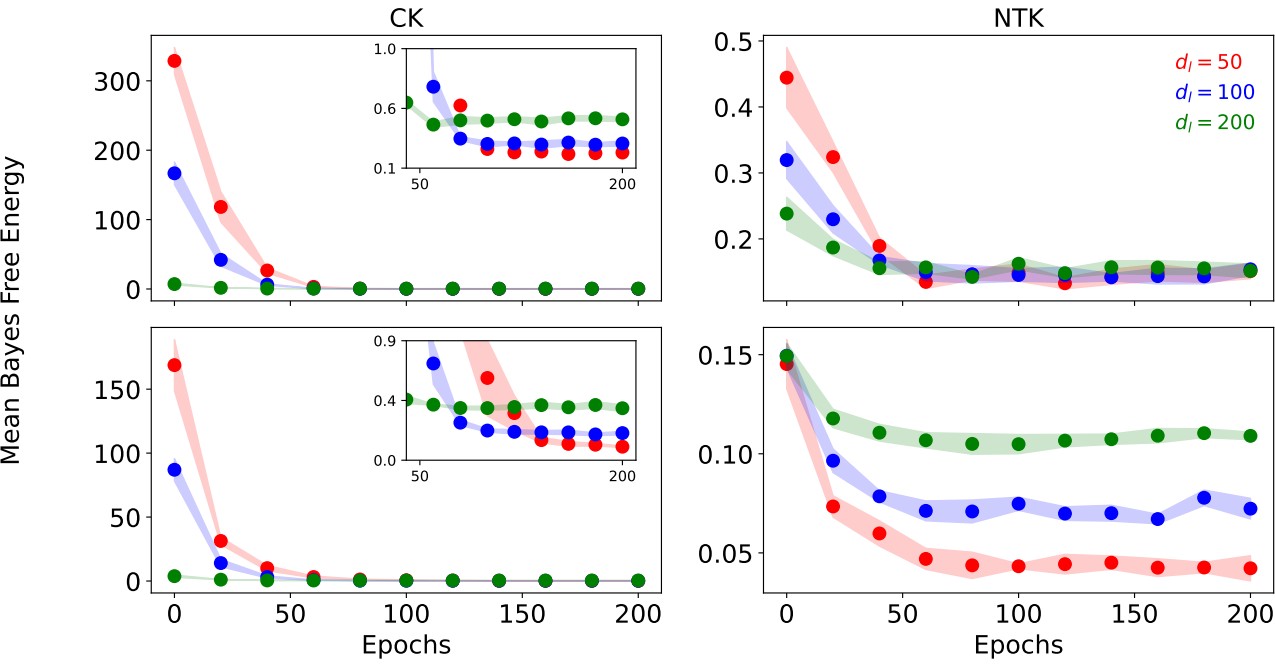

*Figure 10.* Data-fit term for $K_X + \tau\lambda$, under a **zero-mean** prior, as a function of epochs of training, for (top) Gaussian data with $\lambda = \lambda^*$ and (bottom) a teacher network $y = \sin(w^T x)$ with $\lambda = 1$, and $\tau = 10^{-3}$.

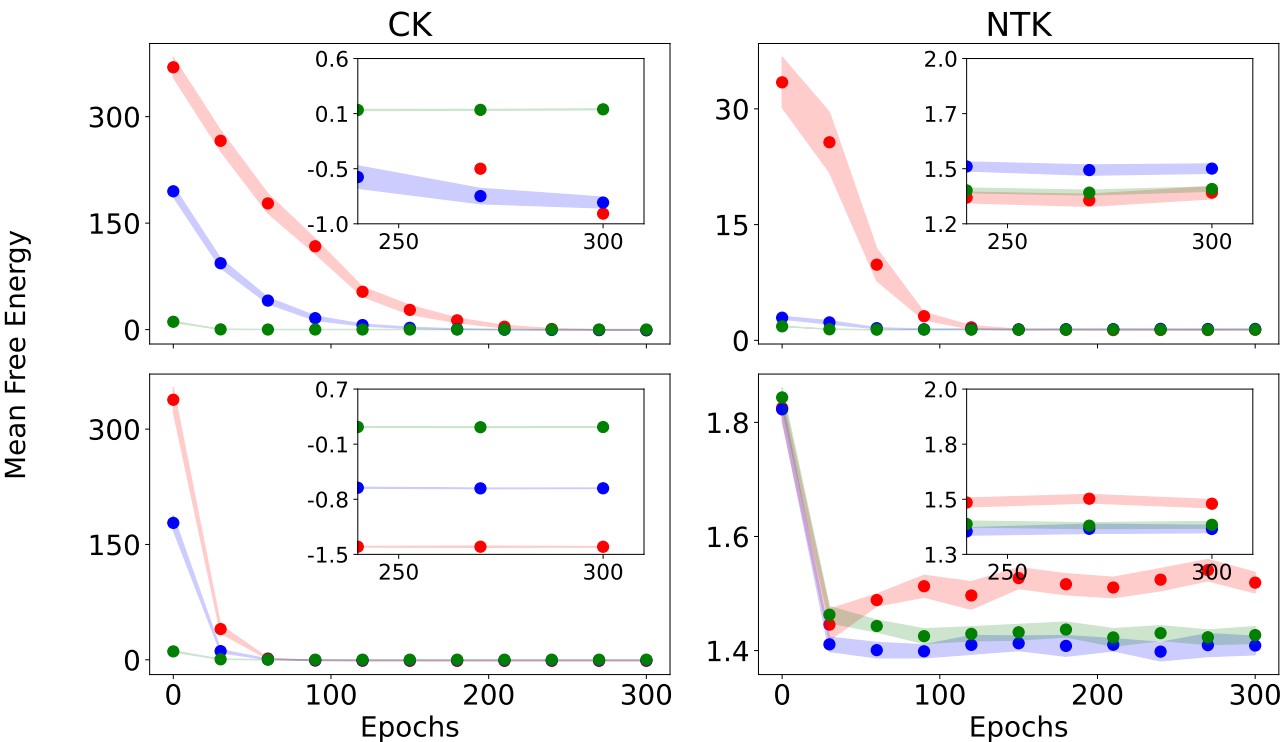

*Figure 11.* BFE as a function of epochs of training, for (top) MNIST and (bottom) CIFAR-10, with $\lambda = 1$, and $\tau = 10^{-3}$.

where $\ell_2(f, y) = (f - y)^2$, and $\tilde{\mathbf{y}}_i \in \mathbb{R}^c$, $i = 1, \dots, n$. Suppose $\boldsymbol{\theta}_S^{\ddagger}$ is any solution to (21). Using $\boldsymbol{\theta}_S^{\ddagger}$, one can construct a family of solutions to (21) as

$$\boldsymbol{\theta}_{S,\mathbf{z}}^{\star} = \boldsymbol{\theta}_S^{\ddagger} + \left(\mathbf{I} - \mathbf{J}_S(\hat{\boldsymbol{\theta}}_S, \mathbf{X})^{\dagger} \mathbf{J}_S(\hat{\boldsymbol{\theta}}_S, \mathbf{X})\right) \mathbf{z}, \quad \forall \mathbf{z} \in \mathbb{R}^{p_S}. \tag{22}$$

The second term in (22) consists of all vectors in the null space of $\mathbf{J}_S(\hat{\boldsymbol{\theta}}_S, \mathbf{X})$. By Wilson et al. (2025, Lemma 3.1), we take $\boldsymbol{\theta}_S^{\ddagger}$ as the unique solution to (21) that is orthogonal to the null space of $\mathbf{J}_S(\hat{\boldsymbol{\theta}}_S, \mathbf{X})$.

Further, by Wilson et al. (2025, Theorem 3.2), solutions of the form (22) can be found by employing Gradient Descent (GD) or Stochastic Gradient Descent (SGD), initialized at $\mathbf{z} \in \mathbb{R}^{p_S}$. The assumptions of Wilson et al. (2025, Theorem 3.2) for GD require a strictly convex loss with locally Lipschitz continuous gradient, which is satisftied for $\ell_2$, and for SGD requires a strongly convex loss with Lipschitz continuous gradient (which is satisfied for $\ell_2$, an interpolating function $\tilde{\mathbf{f}}_{\hat{\boldsymbol{\theta}}}$, and that $\mathbf{J}_S(\hat{\boldsymbol{\theta}}_S, \mathbf{X})$ is full row-rank. For sufficiently over-parameterized DNNs, and when using $\boldsymbol{\theta}_S = \boldsymbol{\theta}$, this rank assumption is reasonable (see Liu et al. (2022)). When $\boldsymbol{\theta}_S = \boldsymbol{\theta}_L$, then $\mathbf{J}_S(\hat{\boldsymbol{\theta}}_S, \mathbf{X})$ is often *not* full-row rank. In this case, we require that either $\mathbf{J}_S(\hat{\boldsymbol{\theta}}_S, \mathbf{X})$ be rank-deficient, or that $p_L > nc$, for the existence of a null-space. The full-row rank assumption required in the proof of the SGD component, so that when a solution to (21) is found, the loss-gradient (with respect to input, not parameters) will be zero. However, for the $\ell_2$ loss, when a solution to (21) is found, the loss-gradient will automatically be zero. Hence, for this particular loss function, we do not require full-row rank.

At solutions of the form (22), setting $\mathbf{z} = \hat{\boldsymbol{\theta}}_S - \mathbf{z}_0$, and using the fact that for the $\ell_2$ loss,

$$\boldsymbol{\theta}_S^{\ddagger} = \mathbf{J}_S(\hat{\boldsymbol{\theta}}_S, \mathbf{X})^{\dagger} \left(\tilde{\mathbf{Y}} - \mathbf{f}_{\boldsymbol{\theta}}(\mathbf{X}) + \mathbf{J}_S(\hat{\boldsymbol{\theta}}_S, \mathbf{X})\hat{\boldsymbol{\theta}}_S\right),$$

where we concatenate the outputs and predictions as $\tilde{\mathbf{Y}} = [\tilde{\mathbf{y}}_1, \dots, \tilde{\mathbf{y}}_n] \in \mathbb{R}^{nc}$, $\mathbf{f}_{\boldsymbol{\theta}}(\mathbf{X}) = [\mathbf{f}_{\boldsymbol{\theta}}(\mathbf{x}_1), \dots, \mathbf{f}_{\boldsymbol{\theta}}(\mathbf{x}_n)] \in \mathbb{R}^{nc}$, we

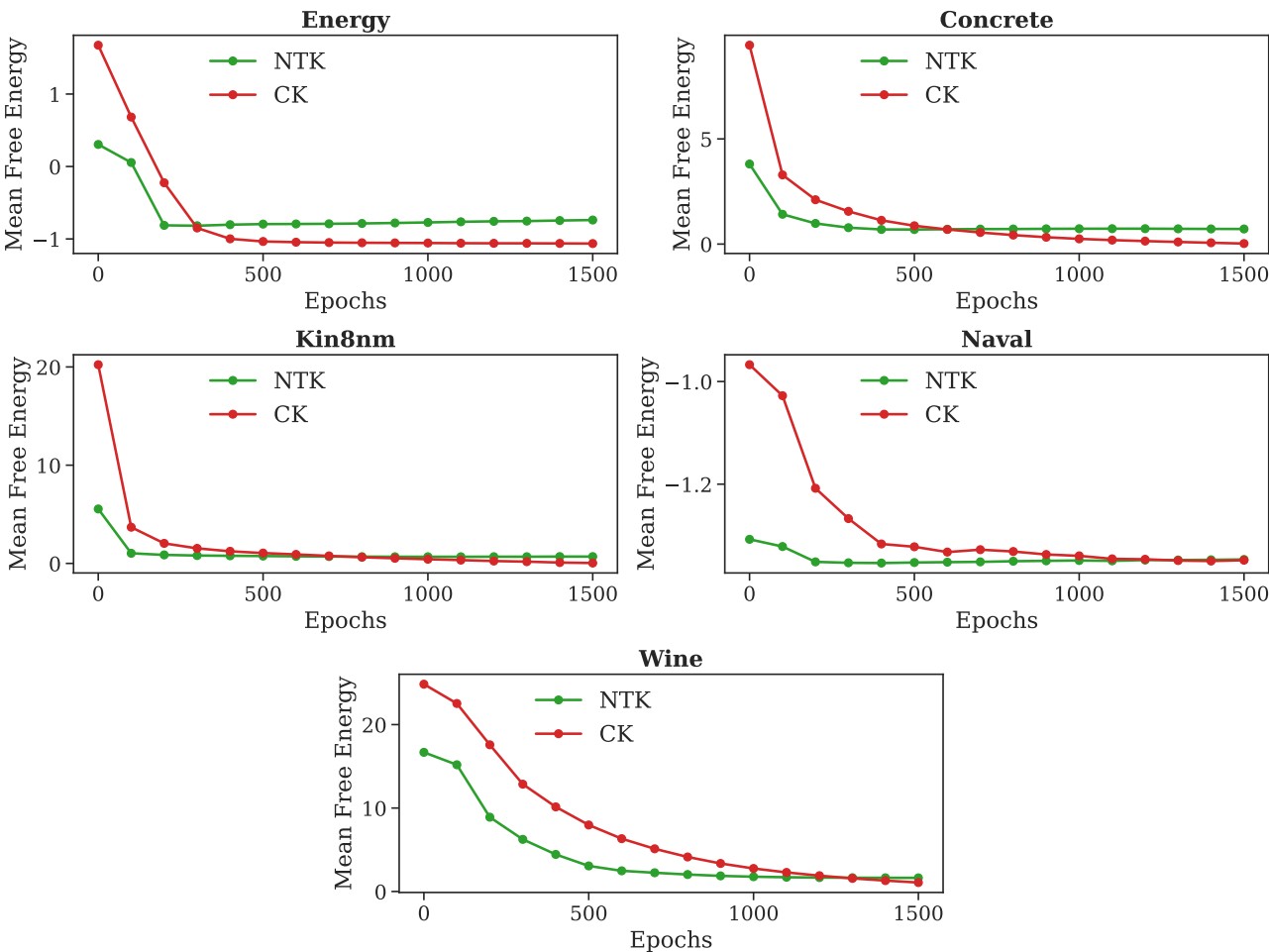

*Figure 12.* BFE as a function of epochs of training, for UCI regression datasets, for a small MLP trained with Adam for 1500 epochs using a learning rate 0.01, with $\lambda = 1, \tau = 0.001$. We see that BFE for CK drops below that of the NTK eventually.

find that

$$\tilde{\mathbf{f}}_{\hat{\boldsymbol{\theta}}}(\boldsymbol{\theta}_{S,\mathbf{z}}^{\star}, \mathbf{x}) = \mathbf{f}_{\hat{\boldsymbol{\theta}}}(\mathbf{x}_i) + \mathbf{J}_S(\hat{\boldsymbol{\theta}}_S, \mathbf{x}) \left( \boldsymbol{\theta}_{S,\mathbf{z}}^{\star} - \hat{\boldsymbol{\theta}}_S \right)$$

$$= \mathbf{f}_{\hat{\boldsymbol{\theta}}}(\mathbf{x}_i) + \mathbf{J}_S(\hat{\boldsymbol{\theta}}_S, \mathbf{x}) \left( \boldsymbol{\theta}_S^{\ddagger} + \left( \mathbf{I} - \mathbf{J}_S(\hat{\boldsymbol{\theta}}_S, \mathbf{X})^{\dagger} \mathbf{J}_S(\hat{\boldsymbol{\theta}}_S, \mathbf{X}) \right) (\hat{\boldsymbol{\theta}}_S - \mathbf{z}_0) - \hat{\boldsymbol{\theta}}_S \right)$$

$$= \mathbf{f}_{\hat{\boldsymbol{\theta}}}(\mathbf{x}_i) + \mathbf{J}_S(\hat{\boldsymbol{\theta}}_S, \mathbf{x}) \left( \mathbf{J}_S(\hat{\boldsymbol{\theta}}_S, \mathbf{X})^{\dagger} \left( \tilde{\mathbf{Y}} - \mathbf{f}_{\boldsymbol{\theta}}(\mathbf{X}) + \mathbf{J}_S(\hat{\boldsymbol{\theta}}_S, \mathbf{X}) \hat{\boldsymbol{\theta}}_S \right) + \left( \mathbf{I} - \mathbf{J}_S(\hat{\boldsymbol{\theta}}_S, \mathbf{X})^{\dagger} \mathbf{J}_S(\hat{\boldsymbol{\theta}}_S, \mathbf{X}) \right) (\hat{\boldsymbol{\theta}}_S - \mathbf{z}_0) - \hat{\boldsymbol{\theta}}_S \right)$$

$$= \mathbf{f}_{\hat{\boldsymbol{\theta}}}(\mathbf{x}_i) + \mathbf{J}_S(\hat{\boldsymbol{\theta}}_S, \mathbf{x}) \mathbf{J}_S(\hat{\boldsymbol{\theta}}_S, \mathbf{X})^{\dagger} \left( \tilde{\mathbf{Y}} - \mathbf{f}_{\boldsymbol{\theta}}(\mathbf{X}) \right) + \mathbf{J}_S(\hat{\boldsymbol{\theta}}_S, \mathbf{x}) \left( \mathbf{I} - \mathbf{J}_S(\hat{\boldsymbol{\theta}}_S, \mathbf{X})^{\dagger} \mathbf{J}_S(\hat{\boldsymbol{\theta}}_S, \mathbf{X}) \right) \mathbf{z}_0.$$

We now set a prior on $\mathbf{z}_0$, namely $\mathbf{z}_0 \sim \mathcal{N}(0, \eta^2 \mathbf{I})$. From this, we compute the distribution of the linear predictive samples

$$\tilde{\mathbf{f}}_{\hat{\boldsymbol{\theta}}}(\boldsymbol{\theta}_{S,\mathbf{z}}^{\star}, \mathbf{x}) \sim \mathcal{N}(\mu(\hat{\boldsymbol{\theta}}, \mathbf{x}), \boldsymbol{\Sigma}(\hat{\boldsymbol{\theta}}, \mathbf{x}))$$

$$\mu(\hat{\boldsymbol{\theta}}, \mathbf{x}) := \mathbf{f}_{\hat{\boldsymbol{\theta}}}(\mathbf{x}_i) + \lim_{\delta \to 0} \mathbf{K}_{\mathbf{x}^*}^T (\mathbf{K}_{\mathbf{X}} + \delta \mathbf{I})^{-1} \left( \tilde{\mathbf{Y}} - \mathbf{f}_{\boldsymbol{\theta}}(\mathbf{X}) \right)$$

$$\boldsymbol{\Sigma}(\hat{\boldsymbol{\theta}}, \mathbf{x}) := \eta^2 (\boldsymbol{\kappa}_{\mathbf{x}^*} - \lim_{\delta \to 0} \mathbf{K}_{\mathbf{x}^*}^T (\mathbf{K}_{\mathbf{X}} + \delta \mathbf{I})^{-1} \mathbf{K}_{\mathbf{x}^*}),$$

where

$$\kappa(\mathbf{x}_i, \mathbf{x}_j) = \mathbf{J}_S(\hat{\boldsymbol{\theta}}_S, \mathbf{x}_i) \mathbf{J}_S(\hat{\boldsymbol{\theta}}_S, \mathbf{x}_j)^T,$$

is the kernel function induced by the feature map $\mathbf{J}_S(\hat{\boldsymbol{\theta}}_S, \mathbf{x})$. Kernel notation then follows that of Section 2.3. We observe that this distribution is equal to that of (1), for $\tau \to 0$, and $\lambda = \eta^{-2}$. This concludes the proof of Lemma 2. For regression, we take $\tilde{\mathbf{Y}} = \mathbf{Y}$. For classification, we take $\tilde{\mathbf{Y}} = \mathbf{f}_{\boldsymbol{\theta}}(\mathbf{X})$. For the LL-GLM, we take $\boldsymbol{\theta}_S = \boldsymbol{\theta}_L$. For the DNN-GLM, we take $\boldsymbol{\theta}_S = \boldsymbol{\theta}$

## E. Sampling from LL-GLM

The LL-GLM sampling procedure from Lemma 2 takes the linearization of a DNN around the parameters in the final layer, and then trains an ensemble of these linear networks on the training data. In order to have an ensemble of predictors, we require variance across these new trained parameters. If the last-layer width is less than the number of training points, which it commonly is, then we require rank-deficiency in the last-layer feature matrix, i.e. the Jacobian with respect to the last-layer. In this section we firstly detail evidence that the Jacobian *should* be rank-deficient in infinite-precision. We then show that in finite precision the Jacobian contains many small, but non-zero, singular values; hence, it is only nearly rank-deficient. However, we note that the *LinearSampling* procedure only employs the inner product of the Jacobian with itself, which squares the singular values, thus taking the problem closer to numerical rank deficiency. This, coupled with convergence properties of gradient descent, results in the truncated SVD solution to the problem, destroying directions of the final layer feature space that do not contribute meaningfully to the predictions.

### E.1. Rank-Deficiency in Final Features

There have been many works which give evidence that neural networks learn low-dimensional representations of the training data. In Pope et al. (2021), the authors estimate the intrinsic dimension (ID) of the underlying data manifold of many common image datasets (MNIST, CIFAR-10, ImageNet etc.), and show that it is significantly smaller than the extrinsic dimension (ED, i.e. the embedded dimension). In Ansuini et al. (2019), the authors estimate the ID of the feature representations for intermediate layers of the neural network. They show that after training, early layers exhibit an increasing ID with layer, after which medium-level layers begin to decrease ID, with final layers exhibiting the smallest ID. This 'hunchback' shape (as coined by the authors) is consistent across architectures. The final feature layer contains an ID an order or more of magnitude smaller than its ED. Further, the authors found a clear negative correlation between ID and test accuracy; the more concise the summary of features, the better the generalization. Importantly, these phenomena did not occur when training on noisy labels. The work of Ma et al. (2018) showed similar findings. Specifically, during training the ID of the final layer is monotonically decreasing with epochs, and the test accuracy follows inversely to the behaviour of the ID. In contrast, when training on noisy labels, the ID initially decreases, and then increases as the network overfits; the test accuracy follows the inverse relationship. Further, Feng et al. (2022) showed theoretically that the rank of the intermediate feature representations (i.e. $X_l$, $l = 1, \ldots, L$) is monotonically non-increasing with layer $l$, though we note that this result is in contrast with 'hunchback' shape of Ansuini et al. (2019).

### E.2. Near Rank-Deficiency in Finite Precision

In finite precision, we find that for many common datasets and models, $\mathbf{J}_L(\hat{\boldsymbol{\theta}}_L, \mathbf{X})$ is not rank-deficient, yet contains many small, but non-zero, singular values. As per the previous discussion, we maintain that this arises due to the finite precision of computations. However, we now show that for feature matrices that are near rank-deficient, the sampling process the LL-GLM 'deletes' many directions that are close to the null-space, and that gradient descent instead finds the truncated SVD solution of (21).

During the gradient descent process to find the solution of (21), initialized at $\mathbf{z}$, iterates are of the form

$$
\begin{aligned}
\frac{\boldsymbol{\theta}_L^{(1)} - \boldsymbol{\theta}_L^{(0)}}{\eta} &= \nabla_{\boldsymbol{\theta}_L} ||\tilde{\mathbf{f}}_{\hat{\boldsymbol{\theta}}}(\boldsymbol{\theta}_L^{(0)}, \mathbf{X}) - \mathbf{Y}||_2^2 = \nabla_{\boldsymbol{\theta}_L} ||\mathbf{f}_{\hat{\boldsymbol{\theta}}}(\mathbf{X}) + \mathbf{J}_L(\hat{\boldsymbol{\theta}}_L, \mathbf{X})(\boldsymbol{\theta}_L^{(0)} - \hat{\boldsymbol{\theta}}_L) - \mathbf{Y}||_2^2 \\
&= \nabla_{\boldsymbol{\theta}_L} ||\mathbf{J}_L(\hat{\boldsymbol{\theta}}_L, \mathbf{X})\bar{\boldsymbol{\theta}}_L^{(0)} - \mathbf{r}||_2^2 \\
&= \mathbf{J}_L(\hat{\boldsymbol{\theta}}_L, \mathbf{X})^T \mathbf{J}_L(\hat{\boldsymbol{\theta}}_L, \mathbf{X})\bar{\boldsymbol{\theta}}_L^{(0)} - \mathbf{J}_L(\hat{\boldsymbol{\theta}}_L, \mathbf{X})^T \mathbf{r}
\end{aligned}
$$

where $\bar{\boldsymbol{\theta}}_L^{(0)} = \boldsymbol{\theta}_L^{(0)} - \hat{\boldsymbol{\theta}}_L$, $\mathbf{r} = \mathbf{f}_{\hat{\boldsymbol{\theta}}}(\mathbf{X}) - \mathbf{Y}$, and $\eta > 0$ is a learning-rate. Further, if the network is sufficiently over-parameterized, then the network will be able to interpolate the training data, and hence there exists $\mathbf{w} \in \mathbb{R}^{d_L}$, where $d_L$ is the width of the last-connected layer of the neural network, such that $\mathbf{Y} = \mathbf{J}_L(\hat{\boldsymbol{\theta}}_L, \mathbf{X})\mathbf{w}$. Further, note that $\mathbf{f}_{\hat{\boldsymbol{\theta}}}(\mathbf{X}) = \mathbf{J}_L(\hat{\boldsymbol{\theta}}_L, \mathbf{X})\hat{\boldsymbol{\theta}}_L$, as $\mathbf{J}_L(\hat{\boldsymbol{\theta}}_L, \mathbf{X})$ contains the learnt features, and $\hat{\boldsymbol{\theta}}_L$ is the weights in the final-connected layer. Thus, $\mathbf{J}_L(\hat{\boldsymbol{\theta}}_L, \mathbf{X})^T \mathbf{r} = \mathbf{J}_L(\hat{\boldsymbol{\theta}}_L, \mathbf{X})^T \mathbf{J}_L(\hat{\boldsymbol{\theta}}_L, \mathbf{X})\hat{\boldsymbol{\theta}}_L - \mathbf{J}_L(\hat{\boldsymbol{\theta}}_L, \mathbf{X})^T \mathbf{J}_L(\hat{\boldsymbol{\theta}}_L, \mathbf{X})\mathbf{w} = \mathbf{J}_L(\hat{\boldsymbol{\theta}}_L, \mathbf{X})^T \mathbf{J}_L(\hat{\boldsymbol{\theta}}_L, \mathbf{X})(\hat{\boldsymbol{\theta}}_L - \mathbf{w})$. So we see that the iterates of gradient descent depend only on $\mathbf{J}_L(\hat{\boldsymbol{\theta}}_L, \mathbf{X})^T \mathbf{J}_L(\hat{\boldsymbol{\theta}}_L, \mathbf{X}) = \mathbf{V}\Lambda^2\mathbf{V}^T$, where $\mathbf{J}_L(\hat{\boldsymbol{\theta}}_L, \mathbf{X}) = \mathbf{U}\Lambda\mathbf{V}^T$ is the singular value decomposition of $\mathbf{J}_L(\hat{\boldsymbol{\theta}}_L, \mathbf{X})$. Note that $\Lambda^2$ denotes the squaring of the singular values of $\mathbf{J}_L(\hat{\boldsymbol{\theta}}_L, \mathbf{X})$. We assume that $\mathbf{J}_L(\hat{\boldsymbol{\theta}}_L, \mathbf{X})$ is nearly-rank deficient. Hence, it has many very small singular, and a large condition number. The singular values of $\mathbf{J}_L(\hat{\boldsymbol{\theta}}_L, \mathbf{X})^T \mathbf{J}_L(\hat{\boldsymbol{\theta}}_L, \mathbf{X})$ will thus be even more extreme, with small values tending closer to zero, while large values will grow larger. Thus, $\mathbf{J}_L(\hat{\boldsymbol{\theta}}_L, \mathbf{X})^T \mathbf{J}_L(\hat{\boldsymbol{\theta}}_L, \mathbf{X})$ may become rank-deficient as per the numerical rank. By this, $\mathbf{J}_L(\hat{\boldsymbol{\theta}}_L, \mathbf{X})^T \mathbf{J}_L(\hat{\boldsymbol{\theta}}_L, \mathbf{X}) = \mathbf{J}_L(\hat{\boldsymbol{\theta}}_L, \mathbf{X}; r)^T \mathbf{J}_L(\hat{\boldsymbol{\theta}}_L, \mathbf{X}; r)$ under numerical precision, where $\mathbf{J}_L(\hat{\boldsymbol{\theta}}_L, \mathbf{X}; r)$ is $\mathbf{J}_L(\hat{\boldsymbol{\theta}}_L, \mathbf{X})$ with small singular-values 'cut-off' such that $r$ is the numerical rank of $\mathbf{J}_L(\hat{\boldsymbol{\theta}}_L, \mathbf{X})^T \mathbf{J}_L(\hat{\boldsymbol{\theta}}_L, \mathbf{X})$. Therefore, gradient descent on a nearly-rank deficient least-squares problem converges to the solution of the truncated least-squares solution of the rank-deficient problem.

### E.3. Singular Value Distribution for Trained Networks

We plot the singular value distribution of the final-layer features, for several regression and classification datasets, using a trained network. We plot the normalized distribution, where we have divided by the largest singular value for each dataset / model. We also show machine epsilon for the *float32* and *float16* datatypes. We see that the majority of the singular values for all datasets and models exists between the machine epsilon for these two datatypes. Thus, using these precision values will give the truncated SVD solution to the least-squares problem in (21), where the amount of truncation depends on the datatype.

### E.4. Convergence of Gradient Descent

Perhaps most importantly to our discussion, we note that for gradient descent, the number of epochs for the algorithm to converge for each direction is inversely proportional to the magnitude of the corresponding singular value of that direction. I.e. for directions with vanishingly small singular values, gradient descent will make no meaningful progress. For our case, we note that once the major eigen-directions have converged, the directions with small singular-values will remain unchanged; hence, early stopping will result in a truncated SVD solution, where the majority of the energy will have been captured by the dominating directions, and many directions will become null-directions, allowing for projections onto the null-space of a matrix which is close to the last-layer Jacobian.

Note that truncation of the spectrum of the CK Gram matrix may actually improve UQ performance; as per our previous discussion, it may be the case that many of these small eigen-directions should in fact be null-directions in infinite-precision. Exploring the effect of low-rank truncation on UQ performance is an interesting avenue for future work.

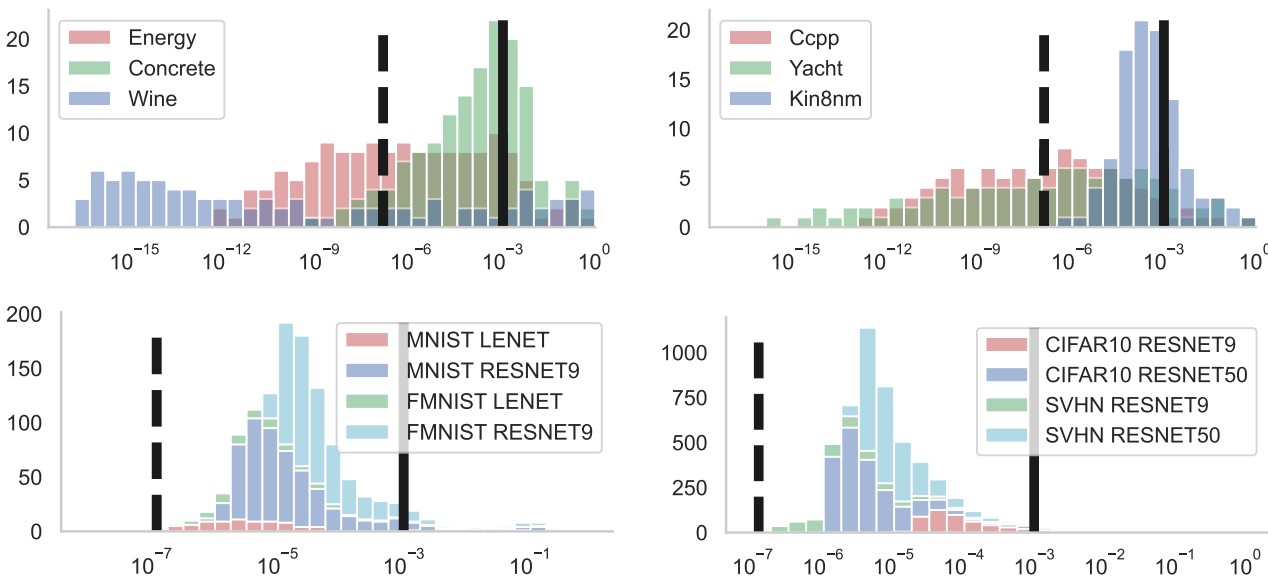

*Figure 13.* Normalized singular value distribution for final trained features, for several datasets and models. The top rows displays regression datasets, while the bottom row gives image classification datasets. We consistently see many small singular values. The dotted black line denotes machine epsilon for *float32*, while the solid black line denotes machine epsilon for *float16*.

## F. Justification for VARROC

In order to evaluate the performance of UQ methods for classification, we first highlight two settings for which we require uncertainty:

**1.** To detect when a network will make an incorrect prediction on a test point that is in-distribution.

**2.** To detect when a test point is out-of-distribution, and hence a correct prediction cannot be made.

The AUC-ROC score is the Area Under the Curve of the Receiver Operator Characteristic. It evaluates the performance of a binary classifier. We can understand this through a classifier that gives a score $\hat{y} \in [0, 1]$ for a test point $x$. If $\hat{y} \leq 0.5$, the classifier selects class 0, and if $\hat{y} > 0.5$, the classifier selects 1. For a test set $\mathbf{X}_{\text{test}}$, and hence a set of predictions $\hat{\mathcal{Y}}_{\text{test}}$, the Receiver-Operator Characteristic considers the ratio of the True Positive Rate (TPR) to the False Positive Rate (FPR), as the decision boundary, i.e. $\alpha$ where if $\hat{y} \leq \alpha$, $x$ is classified as coming from class 0, is taken from 0 to 1. This essentially measures the overlap of the predictions for the two sets; if all class 0 points have a predicted value less than $0.5$, and all class 1 points have a predicted value greater than $0.5$, then as $\alpha$ increases, the TPR will grow to 1, and then plateau at or before $\alpha = 0.5$, at which point the FPR will grow to 1. Hence the area under this curve will be 1. This is exactly what we would like from our uncertainty quantification method; the higher the AUC-ROC score, for a classifier that uses the variance for correct versus incorrect predictions, and for correct versus OoD predictions, the better we are at knowing when a network will give an incorrect prediction.

Specifically, the ROC curve plots the sensitivity of our variance detector as a function of the Type-I error. This suits our purpose, as for both detection of incorrectly predicted points, and out-of-distribution points, we want to maximize the number of points we will correctly predict on, while minimizing the points that will lead to incorrect/erroneous predictions. Hence, we define VARROC-ID and VARROC-OOD as two metrics to compare the performance of uncertainty quantification techniques on classification problems:

**1.** Firstly, we use the variance of the maximum softmax prediction as our uncertainty score. For a Bayesian method, we generally have a mean predictor $\mu : \mathbb{R}^d \to \mathbb{R}^c$ and a covariance function $\Sigma : \mathbb{R}^d \to \mathbb{R}^{c \times c}$, that output in the softmax space. For the variance of the maximum softmax prediction for a given test point $\mathbf{x}^\star$, we first find $\hat{c} = \text{argmax}_k \, \mu(\mathbf{x}^\star)_k$, where $\mu(\mathbf{x}^\star)_k$ denotes the $k$-th output of $\mu(\mathbf{x}^\star)$. That is, we find the class that the Bayesian method predicts, given $\mathbf{x}^\star$. We then take $\Sigma(\mathbf{x}^\star)_{\hat{c},\hat{c}} = \sigma^2(\mathbf{x}^\star)_{\hat{c}}$, that is, the variance of this prediction.

**2.** Secondly, we compute the AUCROC score for two settings: on an in-distribution test set, where we seek to detect

correctly predicted versus incorrectly predicted points (**VARROC-ID**), and using an OOD test set, where we seek to detect correctly predicted versus OOD points (**VARROC-OOD**).

Note that for VARROC-OOD, we do not want to include the entire ID test set, as this may include test points that are incorrectly predicted; if the model assigns these points high variance (which we would prefer), then this will reduce the ability of the method to differentiate between ID and OOD.

VARROC is similar to the uncertainty metric used in Miani et al. (2024); there, the uncertainty score was the variance of the logits, and the metric was AUC-ROC. However, we feel that variance in the softmax predictions is the more appropriate uncertainty score, as this is the space in which prediction occurs, not the logit space.

Instead of the variance in the softmax predictions, one may also use Mutual Information (MI) (de Jong et al., 2023), $I$. This gives the metrics VARROC-MI-ID and VARROC-MI-OOD. Derived from an information theoretic framework, we can estimate the MI as $\hat{I}$ using an MC-sampling framework. For a test point $\mathbf{x}^*$, and for a Bayesian method that outputs $T$ samples $\mathbf{p}(\mathbf{x}^*)_1, \ldots, \mathbf{p}(\mathbf{x}^*)_T$, where $\mathbf{p}(\mathbf{x}^*)_i \in \mathbb{R}^c$, and $\bar{\mathbf{p}}(\mathbf{x}^*)_{.,k}$ is the mean over samples for class $k$,

$$\hat{I}(\mathbf{x}^*) = \sum_{k=1}^{c} \left( \left( \frac{1}{T} \sum_{i=1}^{T} \mathbf{p}(\mathbf{x}^*)_{i,k} \log \mathbf{p}(\mathbf{x}^*)_{i,k} \right) - \bar{\mathbf{p}}(\mathbf{x}^*)_{.,k} \log \bar{\mathbf{p}}(\mathbf{x}^*)_{.,k} \right).$$

The choice of softmax variance versus MI for modeling epistemic uncertainty is still a topic of discussion (Wimmer et al., 2023). Interestingly, these quantities are equal up to leading order (Smith & Gal, 2018). Hence, we measure AUCROC using both softmax variance and MI; we see in practice they give similar results.

Finally, for probabilistic forecasting, one should use the log pointwise predictive density (LPPD) (Gelman et al., 1995). For our purposes, we do not want to simply estimate this using the mean probability prediction for each method. Instead, we incorporate the uncertainty by employing the generalized probit approximation (Gibbs, 1998),

$$p(\mathbf{y}^*|\mathbf{x}^*, \mathcal{D}) \approx \mathrm{softmax} \left( \left\{ \frac{\mu_{*,j}}{\sqrt{1 + \frac{\pi}{8} \Sigma_{*,j,j}}} \right\}_{j=1}^{c} \right),$$

where $\mu_{*,j}, \Sigma_{*,j,j}$ are the posterior predictive mean and covariance for class $j = 1, \ldots, c$. For every method, we take $\mu$ to be the pre-trained DNN; thus, we are measuring the alignment of the predictive variance values to the error in the predictive mean. Note that for both GLMs, scaling the initial prior scale is equivalent to scaling the predictive variance (see Appendix D). Hence, for this metric, we use a randomly selected validation set of 2000 from the test set (without replacement) to tune the scale of the predictive variance to minimize the maximize the validation LPPD. This scale is then used to scale the test predictions, when computing LPPD on the test set.

Finally, note that these metrics are not novel; they are simply existing metrics for model quality (AUCROC, LPPD), evaluated using existing uncertainty measures.

## G. Evaluation of Image Classification

### G.1. Distance-Aware Variance

As discussed in (Foong et al., 2019; Liu et al., 2020), we would prefer for a method to provide distance-aware variance; that is, the variance should be smallest closest to the training data, and then increase with distance. We now display that both the LL-GLM and DNN-GLM possess this property.

We consider a toy example, where we form $C = 3$ clusters of training points in $\mathbb{R}^2$. Each cluster is Gaussian distributed, with mean $\mu_1 = (-2.5, -2.5), \mu_2 = (0, 2.5), \mu_3 = (2.5, -2.5)$, and deviation $\sigma = 0.5$. We sample $n_i = 200$ training points from each cluster, and concatenate these points, as well as their cluster index, into the sets $\mathbf{X}_{\mathrm{train}}$ and $\mathcal{Y}_{\mathrm{train}}$. We then train a $1-$layer MLP with SiLU activation and width 200 on these points. We evaluate LL-GLM, DNN-GLM, DE, LLA, SWAG, SGLD (Welling & Teh, 2011), Spectral Normalized Gaussian Process (SNGP) (Liu et al., 2020) and MC-Dropout on this problem, and plot the variance of the logits (in log-scale) and the probabilities (summed over classes for each test example), for test points in a grid over the training domain ($\pm 1$ past the minimum and maximum values respectively in each

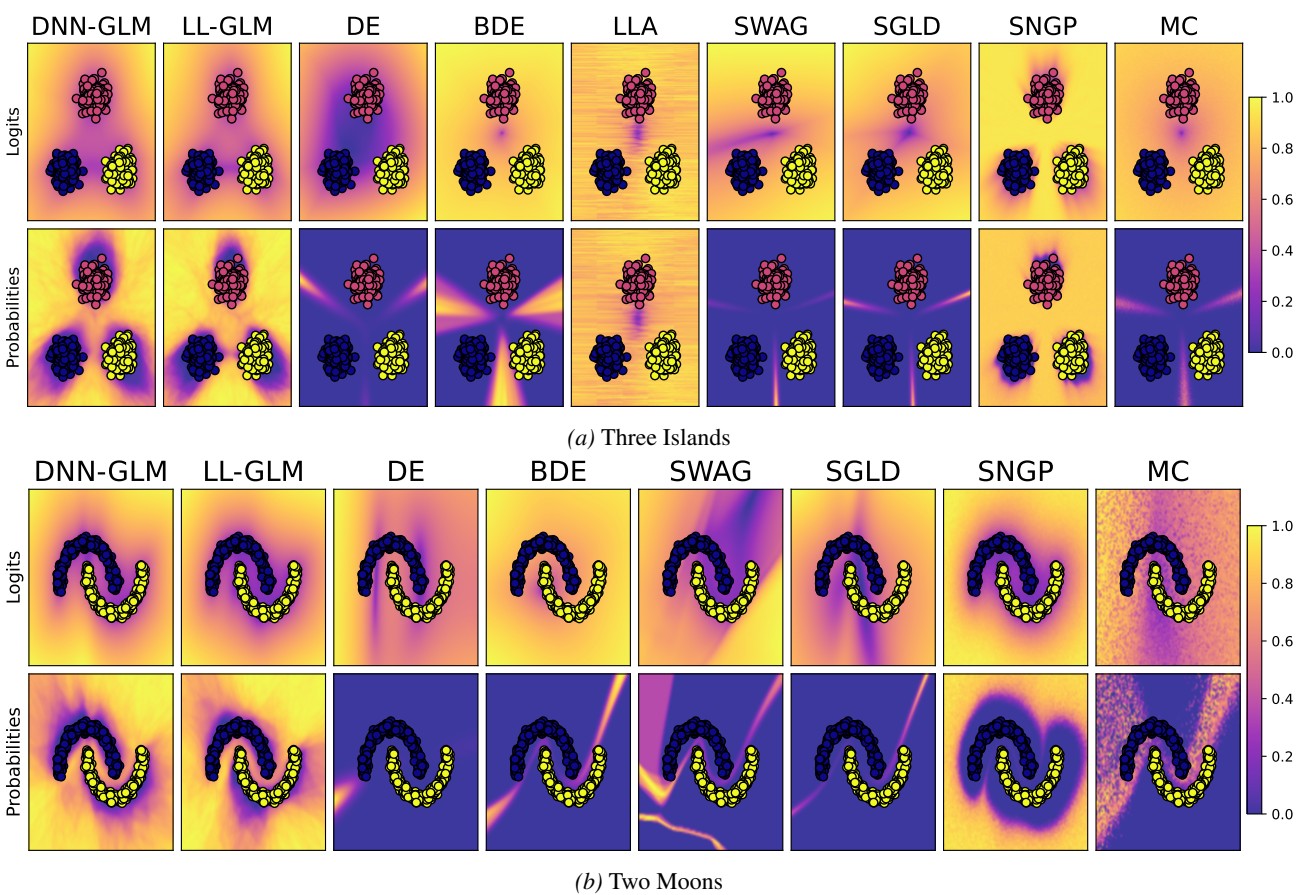

*Figure 14.* Evaluation of distance-aware property of Bayesian methods.

dimension of $\mathbf{x}_i \in \mathbf{X}_{\text{train}}$). All uncertainty values are normalized to $[0, 1]$ for each method.

We plot the results in Figure 14a. We see that LL-GLM, DNN-GLM and SNGP possess the distance-aware variance property in the logit space and probability space; this is impressive, considering that SNGP is not post-hoc, and involves adapting the structure and training of a network to imbibe the posterior with this distance-aware property.

We also compare the methods on the Two Moons dataset (Pedregosa et al., 2011). We use $n = 1000$ training points, and again employ a $1-$layer MLP with SiLU activation and width 200. We present the results in Figure 14b, where we see similar findings to Figure 14a. Note that we could not include LLA in this experiment as the current package for LLA, *laplace-torch* (Daxberger et al., 2021), does not have implementation for binary classification problems.

### G.2. Extended Image Classification Results

We present the LPPD, VARROC, VARROC-MI, execution time and maximum recorded memory results for LL-GLM, DNN-GLM, DE, LLA, SWAG, MC and SMS-UBU on variety of image classification tasks in Table 4. We observe that LL-GLM and DNN-GLM perform similarly across all tasks, and that they perform comparably to DE. We also provide the relative percentage difference between the GLMs, computed as (DNN-GLM $-$ LL-GLM/DNN-GLM $\times$ 100, in Table 5. Note that for smaller models, the time and memory are less influenced by model size, and are instead dominated by processes such as data-loading. We observer that the relative difference in performance is small across all tasks, with large increases in time and memory for the DNN-GLM, and that no GLM performs consistently better across all tasks.

*Table 4.* Results for LPPD, VARROC, MI-VARROC, runtime, and memory. Time is reported in hh:mm:ss and memory in GB.

| Method | LeNet MNIST | | | | | | | LeNet FashionMNIST | | | | | | |
|---|---|---|---|---|---|---|---|---|---|---|---|---|---|---|
| | V-ID | V-OOD | MI-ID | MI-OOD | LPPD | Time | Memory | V-ID | V-OOD | MI-ID | MI-OOD | LPPD | Time | Memory |
| DNN-GLM | 0.9867 | 0.9611 | 0.9853 | 0.9752 | -0.0299 | 89.15 | 0.2941 | 0.8791 | 0.9460 | 0.8717 | 0.9766 | -0.3035 | 85.59 | 0.2941 |
| LL-GLM | 0.9851 | 0.9284 | 0.9820 | 0.9213 | -0.0299 | 122.9 | 0.1216 | 0.8805 | 0.8871 | 0.8753 | 0.9247 | -0.3039 | 95.21 | 0.3382 |
| DE | 0.9826 | 0.9464 | 0.9802 | 0.9627 | -0.0312 | 1.804e+03 | 0.1284 | 0.8751 | 0.9465 | 0.8625 | 0.9709 | -0.3035 | 1.740e+03 | 0.2596 |
| LLA | 0.9864 | 0.9318 | 0.9847 | 0.9318 | -0.0299 | 14.46 | 0.2545 | 0.8786 | 0.9366 | 0.8720 | 0.9681 | -0.3037 | 14.52 | 0.2551 |
| SWAG | 0.9888 | 0.9587 | 0.9873 | 0.9618 | -0.0647 | 360.1 | 0.2226 | 0.8684 | 0.8322 | 0.8700 | 0.8735 | -0.3341 | 346.6 | 0.2232 |
| MC | 0.9788 | 0.9227 | 0.9767 | 0.9227 | -0.0378 | 12.63 | 0.1065 | 0.8520 | 0.8497 | 0.8488 | 0.9042 | -0.3088 | 12.78 | 0.1071 |

| Method | ResNet9 MNIST | | | | | | | ResNet9 FashionMNIST | | | | | | |
|---|---|---|---|---|---|---|---|---|---|---|---|---|---|---|
| | LPPD | V-ID | V-OOD | MI-ID | MI-OOD | Time | Memory | LPPD | V-ID | V-OOD | MI-ID | MI-OOD | Time | Memory |
| DNN-GLM | -0.0147 | 0.9880 | 0.9806 | 0.9879 | 0.9861 | 00:08:38 | 7.254 | -0.2077 | 0.9208 | 0.9765 | 0.9219 | 0.9849 | 00:08:36 | 7.254 |
| LL-GLM | -0.0149 | 0.9883 | 0.9825 | 0.9877 | 0.9851 | 00:00:36 | 1.699 | -0.2153 | 0.9188 | 0.9651 | 0.9179 | 0.9754 | 00:00:36 | 1.699 |
| DE | -0.0149 | 0.9908 | 0.9920 | 0.9901 | 0.9939 | 00:54:36 | 2.672 | -0.2264 | 0.9250 | 0.9793 | 0.9244 | 0.9979 | 00:54:46 | 2.672 |
| LLA | -0.0147 | 0.9848 | 0.9843 | 0.9831 | 0.9863 | 00:00:25 | 1.952 | -0.2730 | 0.9096 | 0.9640 | 0.9093 | 0.9788 | 00:00:25 | 1.952 |
| SWAG | -0.0147 | 0.9864 | 0.9777 | 0.9855 | 0.9797 | 00:13:26 | 2.511 | -0.2585 | 0.9198 | 0.9647 | 0.9196 | 0.9685 | 00:13:39 | 2.511 |
| MC | -0.0145 | 0.9871 | 0.9746 | 0.9868 | 0.9765 | 00:00:25 | 1.901 | -0.2681 | 0.9187 | 0.9668 | 0.9175 | 0.9735 | 00:00:25 | 1.901 |
| SMS-UBU | -0.0157 | 0.9843 | 0.9922 | 0.9837 | 0.9960 | 00:13:10 | 3.316 | -0.2496 | 0.9003 | 0.9187 | 0.9006 | 0.9988 | 00:13:11 | 3.316 |

| Method | ResNet50 CIFAR-10 | | | | | | | ResNet50 CIFAR-100 | | | | | | |
|---|---|---|---|---|---|---|---|---|---|---|---|---|---|---|
| | LPPD | V-ID | V-OOD | MI-ID | MI-OOD | Time | Memory | LPPD | V-ID | V-OOD | MI-ID | MI-OOD | Time | Memory |
| DNN-GLM | -0.2397 | 0.9063 | 0.8957 | 0.9134 | 0.9071 | 00:07:36 | 21.99 | -1.0520 | 0.7703 | 0.7917 | 0.8414 | 0.8868 | 00:07:23 | 22.16 |
| LL-GLM | -0.2427 | 0.9216 | 0.8955 | 0.9224 | 0.8998 | 00:00:34 | 10.5 | -1.0460 | 0.8260 | 0.8381 | 0.8555 | 0.8784 | 00:00:33 | 10.75 |
| DE | -0.2389 | 0.9201 | 0.9085 | 0.9223 | 0.9188 | 47:50:00 | 14.15 | -1.1150 | 0.7828 | 0.7971 | 0.8430 | 0.8914 | 46:56:40 | 14.35 |
| LLA | -0.2719 | 0.9110 | 0.8889 | 0.9134 | 0.8936 | 00:00:58 | 11.7 | -1.0700 | 0.7041 | 0.6785 | 0.8418 | 0.8761 | 00:03:24 | 38.58 |
| SWAG | -1.5630 | 0.4909 | 0.3252 | 0.5900 | 0.5009 | 05:06:50 | 15.14 | -1.3870 | 0.6313 | 0.6316 | 0.8127 | 0.8967 | 05:10:30 | 15.23 |
| MC | -0.2660 | 0.9069 | 0.8872 | 0.8989 | 0.8839 | 00:01:04 | 11.44 | -1.0490 | 0.8105 | 0.8203 | 0.8247 | 0.8333 | 00:01:02 | 11.57 |
| SMS-UBU | -0.2494 | 0.5608 | 0.6831 | 0.6991 | 0.8162 | 01:13:34 | 16.99 | -1.0550 | 0.2910 | 0.2945 | 0.6012 | 0.7497 | 01:15:10 | 17.12 |

| Method | ResNet50 SVHN | | | | | | | ResNet50 ImageNet | | | | | | |
|---|---|---|---|---|---|---|---|---|---|---|---|---|---|---|
| | LPPD | V-ID | V-OOD | MI-ID | MI-OOD | Time | Memory | LPPD | V-ID | V-OOD | MI-ID | MI-OOD | Time | Memory |
| DNN-GLM | -0.4331 | 0.9097 | 0.9218 | 0.9177 | 0.9347 | 01:31:48 | 63.56 | -0.9610 | 0.8130 | 0.7960 | 0.8340 | 0.8950 | 21:58:19 | 66.089 |
| LL-GLM | -0.4319 | 0.9154 | 0.9043 | 0.9227 | 0.9061 | 00:05:21 | 12.45 | -0.9630 | 0.8110 | 0.7730 | 0.8500 | 0.9180 | 10:43:51 | 14.818 |
| DE | -0.4533 | 0.8469 | 0.9252 | 0.8699 | 0.9565 | 17:31:40 | 16.06 | - | - | - | - | - | - | - |
| LLA | -0.4917 | 0.9151 | 0.9068 | 0.9261 | 0.9088 | 00:01:30 | 13.6 | - | - | - | - | - | - | - |
| SWAG | -0.4427 | 0.8866 | 0.9268 | 0.9031 | 0.9401 | 02:24:09 | 17.03 | - | - | - | - | - | - | - |
| MC | -0.4904 | 0.9081 | 0.9059 | 0.9139 | 0.9073 | 00:01:55 | 13.34 | -0.9650 | 0.5000 | 0.5000 | 0.5120 | 0.5340 | 01:25:28 | 18.657 |
| SMS-UBU | -0.4815 | 0.5136 | 0.7758 | 0.6357 | 0.8969 | 01:47:10 | 18.9 | - | - | - | - | - | - | - |

# H. Implementation Details

**BFE** The implementation details for all figures in Section 3 and Appendix A can be found in the following code.

**Toy Regression** We use $n = 20$ training points, and $n = 10000$ test points. The network is a $1-$layer MLP with width $50$ and *SiLU* activation function. We train this network for $10000$ epochs with a learning rate of $0.001$, using an *Adam* optimizer and a *Polynomial decay* learning rate scheduler. For the GLM posteriors, we train with $\eta = 1$, $S = 10$, epochs = $5000$, learning rate = $0.001$, momentum $\mu = 0.9$. Post-hoc, we scale the variance by $\eta = \sqrt{200}$ for the DNN-GLM, $\eta = \sqrt{2000}$ for the LL-GLM, to be in line with the scale of the problem. We are able to scale $\eta$ post-hoc by the distribution of the linear predictive samples. Note that this post-hoc scaling would usually be achieved through a validation set.

**UCI Regression** Each dataset used a random $70/15/15$ split for training/test/validation, for each repeat of each experiment. An MLP with *tanh* activation was used. Hyperparameters for experiments can be found in Table 6. Not that for DNN-GLM and LL-GLM, the full Jacobian could be stored and used for all datasets except for *Song*. A *PolyLR* scheduler refers to the *PyTorch Polynomial decay* learning rate scheduler. Note that for the GLM posterior inference, the GLMs were trained with $\lambda = 0.01$, to increase convergence; then, a ternary search algorithm was used to find the optimal $\lambda$ post-hoc (which scales the variance) that minimized the ECE on the validation set. See Wilson et al. (2025, Appendix F.1) for more details on this approach.

**Toy Classification** Details on the datasets used can be found in Appendix G.1. For the **Three Islands** dataset, a $1-$layer MLP with *SiLU* activation and width $200$ was trained for $1000$ epochs (with SGD w/ momentum $0.9$) with a learning rate of $0.01$. The DNN-GLM and LL-GLM used $100$ samples, and were trained for $1000$ epochs with learning rate $0.01$ and $0.03$, with $\eta = 12$ and $\eta = 27$ respectively. For the **Two Moons** dataset, a $1-$layer MLP with *SiLU* activation

*Table 5.* Relative difference comparison: each entry is $(\text{DNN-GLM} - \text{LL-GLM})/\text{DNN-GLM} \times 100$.

| Model / Dataset | LPPD (%) | V-ID (%) | V-OOD (%) | MI-ID (%) | MI-OOD (%) | Time (%) | Mem (%) |
|---|---|---|---|---|---|---|---|
| LeNet MNIST | 0.00 | 0.16 | 3.40 | 0.33 | 5.53 | -37.86 | 58.65 |
| LeNet FMNIST | -0.13 | -0.16 | 6.23 | -0.41 | 5.31 | -11.24 | -14.99 |
| ResNet9 MNIST | -1.16 | -0.03 | -0.19 | 0.02 | 0.10 | 93.02 | 76.58 |
| ResNet9 FMNIST | -3.66 | 0.22 | 1.17 | 0.43 | 0.96 | 93.02 | 76.58 |
| ResNet50 CIFAR-10 | -1.25 | -1.69 | 0.02 | -0.99 | 0.80 | 92.63 | 52.25 |
| ResNet50 SVHN | 0.28 | -0.63 | 1.90 | -0.54 | 3.06 | 94.17 | 80.41 |
| ResNet50 CIFAR-100 | 0.57 | -7.23 | -5.86 | -1.68 | 0.95 | 92.63 | 51.49 |
| ResNet50 ImageNet | -0.21 | 0.25 | 2.89 | -1.92 | -2.57 | 51.16 | 77.58 |
| **Average** | -0.70 | -1.14 | 1.20 | -0.60 | 1.77 | 58.44 | 57.32 |

and width 200 was trained for 3000 epochs (with SGD w/ momentum 0.9) with a learning rate of 0.1. The DNN-GLM and LL-GLM used 100 samples, and were trained for 1000 epochs with learning rate 0.01 and 0.01, with $\eta = 30$ and $\eta = 80$ respectively. These values of $\eta$ were chosen to display the distance-aware ability visually. For hyper-parameters of competing methods, please refer to the script in the referenced code repository. For plotting, the variance of the logits was passed through a *log* function. For the probabilities, we took the maximum variance of the softmax probabilities over the classes, for each prediction. Before plotting, both variance types were normalized to be in a $[0, 1]$ range.

**Image Classification** Details of the training procedure are found in Table 7. For CIFAR-10, we used a random horizontal flip and crop as a regularizer for training images. For the GLMs, DE and MC-Dropout, 10 posterior samples were used. A dropout probability of 0.1 was used for MC-Dropout. For SWAG, the posterior learning rate was kept the same as that for the DNN for the CIFAR-100 and SVHN datasets; otherwise, SWAG used $100\times$ the learning rate. Further, the posterior epochs were kept the same as the DNN epochs, and 100 posterior samples were used. For LLA, a last-layer approximation was used, as well as a KFAC approximation for all datasets. The hyper-parameters for LLA were chosen by maximizing the marginal likelihood. Note that for ImageNet, neither DE nor LLA were included, as the computational cost was too prohibitive. For SMS-UBU, we use a single chain and the full-subspace feature map. We use the hyperparameters listed in Paulin et al. (2024, Section 5 & Table 3). Further, for the SWA (Izmailov et al., 2018) component of SMS-UBU, we take the pre-trained DNN, run SWA (using Adam with weight decay equal to that for our original DNN training) for 5 epochs, and then run SMS-UBU from the averaged parameters. We take 40 posterior samples, and discard the first 10 as a burn-in.

**GPT-2** The trained GPT-2 weights were taken from *HuggingFace*; a classification head was then attached, and was trained on the binary IMDB movie dataset (25000 training points and 25000 test points). The model was trained using the full context length for GPT-2, with 10 epochs, a batch-size of 8, the *AdamW* optimizer with learning-rate 0.00002 and $\epsilon = 1e-8$, and a linearly decreasing learning rate scheduler. A training accuracy of 0.988 was achieved. For the GLMs, a learning rate of 0.00001 and scale $\eta = 0.01$ was used; 10 posterior epochs were used. Due to memory constraints on the DNN-GLM, a batch size of 4 was used for both the DNN-GLM and LL-GLM.

**Singular-Value Distribution** For the singular-value histograms in Appendix E.3, we take the Jacobian $\mathbf{J}_L(\hat{\boldsymbol{\theta}}_L, \mathbf{X})$, with respect to the parameters in the final connected layer, of a DNN that has been trained. For the UCI datasets, we take the training procedures in Table 6, and train the DNN on 5 random training splits of the dataset. We then plot the average singular-value histogram of $\mathbf{J}_L(\hat{\boldsymbol{\theta}}_L, \mathbf{X})^T \mathbf{J}_L(\hat{\boldsymbol{\theta}}_L, \mathbf{X}) \in \mathbb{R}^{p_L \times p_L}$, where the distribution has been normalized such that the maximum singular-value is 1. For image classification, we take DNNs that have been trained according to Table 7, and the plot again the normalized singular-value histogram of $\mathbf{J}_L(\hat{\boldsymbol{\theta}}_L, \mathbf{X})^T \mathbf{J}_L(\hat{\boldsymbol{\theta}}_L, \mathbf{X}) \in \mathbb{R}^{p_L \times p_L}$.

**Low-Rank Approximation** For this experiment, we take the DNN and training procedures for the *Energy* dataset, as detailed in Table 6. We then compute the predictive variance on the test set, for LL-GLM using $S = 100$, epochs = 5000, learning rate 0.1, and $\eta = 1$. We then take 10 evenly-spaced integers between 10 and 150 (the numerical rank of $\mathbf{J}_L(\hat{\boldsymbol{\theta}}_L, \mathbf{X})$ for this dataset). For each integer $j$, we compute LL-GLM, with $\mathbf{J}_L(\hat{\boldsymbol{\theta}}_L, \mathbf{X})$ replaced by $\mathbf{J}_{\hat{\boldsymbol{\theta}}_L}(\mathbf{X}; j)$, i.e. the approximation based off of the SVD decomposition that limits the rank of $\mathbf{J}_L(\hat{\boldsymbol{\theta}}_L, \mathbf{X})$ to $j$. We compute the predictive posterior for this rank-restriction, and compute the relative $l_2$ norm difference compared to the original full-rank predictive variance. This is repeated for 2 random repeats for each rank $j$. Importantly, we also compute this for the full-rank LL-GLM again, which gives us an idea of the inherent error in this system. We then plot the mean relative difference for each rank $j$.

*Table 6.* Training procedure for UCI regression results in Table 1.

| DNN | Energy | Concrete | Kin8nm | Naval | CCPP | Wine | Yacht | Protein | Song |
|---|---|---|---|---|---|---|---|---|---|
| Learning Rate | $10^{-2}$ | $10^{-2}$ | $10^{-2}$ | $10^{-2}$ | $10^{-2}$ | $10^{-2}$ | $10^{-2}$ | $10^{-2}$ | $10^{-2}$ |
| Epochs | 1500 | 1000 | 500 | 150 | 100 | 100 | 1000 | 250 | 50 |
| Weight Decay | $10^{-5}$ | $10^{-5}$ | $10^{-5}$ | $10^{-4}$ | $10^{-5}$ | $10^{-4}$ | $10^{-5}$ | $10^{-4}$ | 0 |
| Optimizer | Adam | Adam | SGD | SGD | Adam | SGD | Adam | SGD | SGD |
| Scheduler | PolyLR | PolyLR | None | None | PolyLR | None | PolyLR | None | None |
| MLP Size | [150] | [150] | [100, 100] | [150, 150] | [100, 100] | [100] | [100] | [150, 200, 150] | [1000, 1000, 500, 50] |
| **GLMs** | | | | | | | | | |
| Learning Rate | $10^{-2}$ | $10^{-2}$ | $10^{-2}$ | $10^{-2}$ | $10^{-2}$ | $10^{-2}$ | $10^{-2}$ | $10^{-2}$ | $10^{-3}$ |
| Epochs | 150 | 100 | 50 | 15 | 10 | 10 | 100 | 25 | 10 |
| **Experiment** | | | | | | | | | |
| No. experiments | 10 | 10 | 10 | 10 | 10 | 10 | 10 | 10 | 3 |

*Table 7.* Training procedure for image classification results in Figure 6 and Table 4.

| | LeNet5 MNIST | LeNet5 FMNIST | ResNet9 MNIST | ResNet9 FMNIST | ResNet50 CIFAR10 | ResNet50 CIFAR100 | ResNet50 SVHN | ResNet50 ImageNet |
|---|---|---|---|---|---|---|---|---|
| **DNN** | | | | | | | | |
| Learning Rate | 0.005 | 0.005 | 0.001 | 0.001 | 0.1 | 0.1 | 0.01 | 0.1 |
| Epochs | 35 | 35 | 10 | 10 | 200 | 200 | 50 | 10 |
| Weight Decay | 0.0001 | 0.0001 | 0.0001 | 0.0001 | 0.0005 | 0.0005 | 0.0005 | 0.0005 |
| Batch Size | 152 | 152 | 100 | 100 | 128 | 128 | 128 | 56 |
| Optimizer | Adam | Adam | Adam | Adam | SGD | SGD | SGD | - |
| Scheduler | Cosine | Cosine | Cosine | Cosine | Cosine | Cosine | Cosine | - |
| Accuracy | 99% | 90% | 99.5% | 93.5% | 92.5% | 75% | 87% | 76% |
| **DNN-GLM** | | | | | | | | |
| Learning Rate | 0.01 | 0.01 | 0.01 | 0.001 | 0.001 | 0.005 | 0.001 | 0.001 |
| Epochs | 10 | 10 | 2 | 2 | 2 | 2 | 3 | 2 |
| Batch Size | 152 | 152 | 152 | 152 | 152 | 152 | 152 | 56 |
| $S$ | 10 | 10 | 10 | 10 | 10 | 10 | 10 | 10 |
| $\gamma$ | 0.01 | 0.01 | 0.01 | 0.01 | 0.01 | 0.01 | 0.001 | 0.001 |
| **LL-GLM** | | | | | | | | |
| Learning Rate | 0.01 | 0.01 | 0.01 | 0.01 | 0.01 | 0.01 | 0.001 | 0.01 |
| Epochs | 20 | 20 | 2 | 2 | 2 | 2 | 3 | 3 |
| Batch Size | 56 | 152 | 152 | 152 | 152 | 152 | 152 | 56 |
| $S$ | 10 | 100 | 10 | 10 | 10 | 10 | 10 | 10 |
| $\gamma$ | 1.0 | 1.0 | 0.5 | 0.5 | 0.1 | 0.1 | 0.01 | 0.001 |

