# OpenReview forum: "Is the Last Layer Sufficient for Uncertainty Quantification?"
_ICML.cc/2026/Conference — ICML 2026 regular_

### Official Review · Reviewer_6SMn · 2026-02-25

**Soundness:** 2
**Presentation:** 3
**Significance:** 3
**Originality:** 3
**Overall Recommendation:** 4
**Confidence:** 2

**Summary:**

The work explores last-layer approximations for Bayesian Neural Networks (BNN). Typically BNNs are treated with all layers as Bayesian, but only treating the last layer as Bayesian is often considered as a computational “shortcut” with a presumed worsened ability to estimate epistemic uncertainty.

This paper specifically looks at this using Bayesian Generalized Linear Models as an approximation of Bayesian Neural Networks. Based on previous literature, this can be approached as a Gaussian Process, where the Conjugate Kernel represents the last-layer model, and the Neural Tangent Kernel represents the fully Bayesian treatment. The relationship between these kernels and the BNNs comes from previous work.

The paper then provides theoretical contributions (that I did not fully grasp) comparing these two kernels and showing that they have the ability to achieve the same performance (measured as Bayes Free Energy), are both robust, and that stronger expressivity of the Neural Tangent Kernel falls away after training both models.

The two conditions (last layer vs. full model) are then compared on 9 UCI regression tasks, where the fully Bayesian model does not clearly outperform the last-layer model. The methods are also evaluated in classification on their OOD-detection ability and ability to detect misclassifications, which are contextualised against various relevant models. Deep Ensembles still outperforms the methods in performance, but is computationally more expensive.

**Compliance With Llm Reviewing Policy:**

Affirmed.

**Final Justification:**

All of my main concerns have been addressed. Still, I am not sufficiently familiar with Neural Tangent Kernels as an approach to last-layer BNNs. Therefore, I cannot properly estimate the impact of the contribution.

I still have small concerns that the paper fails to find cases where last-layer approximations are insufficient, but this does not mean that there may be no such cases. It appears that in general, last-layer approximations are fine, with somewhat stronger evidence than previously presented.

I maintain a very low confidence review, but increase to an weak accept. Concerns about presentation were addressed well, and this also cleared up some confusion about the evaluation. Still, it is not clear whether this will have major impact, as there is already a commonly held belief that last-layer approximations are probably sufficient, and I am not convinced this paper will be readily interpretable to many readers (also considering the low confidence reviews).

**Key Questions For Authors:**

Q1: In Tables 1 and 2, how do you decide when there is a difference and when there is not? Additionally, what do the $\pm$ indicate? Std.dev, var, SEM? How do you conclude from Table 2 that there is no difference between DNN-GLM and LL-GLM? DNN-GLM has mean=0.84, while LLM-GLM has mean=0.82. The (presumably) standard deviation is 0.02. I do not see why this should not be considered a difference.

Q2: In Figure 5, why is such a small epsilon chosen? This makes it hard to observe the method’s response to aleatoric uncertainty.

**Limitations:**

A limitations section is missing. While I am not familiar enough with the presented method to identify which limitations should be described, I expect that some limitations must exist.

**Strengths And Weaknesses:**

**Strengths**

S1: The paper addresses a topic that has potential for large impact in UQ. Computational cost is commonly a bottleneck, and last-layer approximations have the potential to substantially reduce computational cost, while maintaining good performance. (Significance)

S2: The work has a strong theoretical component to support its empirical contribution. I do not fully grasp the theoretical component, but trust that it is done in good faith, adding theoretical underpinnings for observed outcomes (Soundness)

S3: The perspective of treating BNNs and last-layer BNNs as Gaussian Processes in function space is new to me, and seems promising to add theoretical support. However, I am not sufficiently familiar with earlier work to value novelty. (Originality)

**Weaknesses**

W1: The empirical evaluation appears limited. The evaluation does not distinguish between aleatoric and epistemic uncertainty, but seems to test them interchangeably (ECE in Table 1 is both, VARROC-OOD is epistemic, VARROC-ID is both). I will point to Mucsanyi et al for broader evaluations commonly done for UQ.

W2: The key claim appears to be that last-layer models are sufficient, and that fully Bayesian models are not needed. However, the comparison on regression examples appears to be mainly oriented around small lower dimensional datasets, and does not extensively evaluate epistemic uncertainty in these. Figure 6 represents results in comparison, but obscures differences between methods, as well as differences between the full and last-layer model. From the underlying data in Table 4 we see that there are several conditions in which the full model outperforms the last-layer model. In Table 2 results for a GPT2 model are presented, which does suggest stronger performance of the full model. Overall, the empirical evaluation does not attempt to “zoom in” on potential differences, and there are no definitions for what is considered significant or not. (Soundness)

W3: VARROC-ID and VARROC-OOD are proposed as novel evaluation methods, but these are actually well-established evaluations. VARROC-OOD is OOD detection with AUC-ROC as metric, VARROC-ID is abstinence with AUC-ROC as metric. (Originality)

W4: This work uses the variance of the max class probability as a measure of uncertainty, which appears to not consider the knowledge available on uncertainty measures over second-order-distributions. This appears ad-hoc while more established measures are available. See Wimmer et al., De Jong et al., and/or Smith et al. (Soundness)

W5: Authors claim that linearizing only the last layer is commonly thought to lead to decreased performance, but do not point to any work that expresses this thought. It’s possible that the assumption they’re rebutting is a strawman argument. The work would be presented stronger if you can point to papers that make claims or assumptions about last-layer approximations. (Presentation) Overall, the work does not explore any previous claims or research about last-layer approximations, making it hard to establish novelty, (Originality)


Smith, Lewis, and Yarin Gal. "Understanding measures of uncertainty for adversarial example detection." arXiv preprint arXiv:1803.08533 (2018).

Wimmer, Lisa, et al. "Quantifying aleatoric and epistemic uncertainty in machine learning: Are conditional entropy and mutual information appropriate measures?." Uncertainty in artificial intelligence. PMLR, 2023.

Mucsányi, Bálint, Michael Kirchhof, and Seong Joon Oh. "Benchmarking uncertainty disentanglement: Specialized uncertainties for specialized tasks." Advances in neural information processing systems 37 (2024): 50972-51038.

de Jong, Ivo Pascal, Andreea Ioana Sburlea, and Matias Valdenegro-Toro. "Uncertainty Quantification in Machine Learning for Biosignal Applications--A Review." arXiv preprint arXiv:2312.09454 (2023).

---

> ### Author Rebuttal · Authors · 2026-03-31
>
> We appreciate the in-depth review of our work by the reviewer. We are glad that they consider this topic has "potential for large impact in UQ", and that they applaud the "strong theoretical component".
>
> **Aleatoric vs. Epistemic**: We first clarify that the uncertainty that we compute is exactly the EU (variance of noiseless-GP); we do not estimate aleatoric uncertainty (AU). Note ECE computes the coverage error of prediction intervals. Prediction intervals use total uncertainty (AU + EU). We assume our GP is correctly-specified, but the noise-model is not. Then, by definition, improvements in ECE for intervals computed with $\sigma_{\text{ep}}$ are due to better EU. Note there will be a lower-bound on ECE due to no AU. Both GLMs will have the same lower-bound on ECE, thus comparing ECE between methods exactly measures epistemic UQ quality. Further, note that MNIST, FMNIST, CIFAR-10, SVHN, CIFAR-100 are generally considered as 'noiseless' datasets [6, 7]. Hence, improvement in VARROC-ID score will be due to epistemic UQ quality.
>
> **Small Regression, no EU**: Please see our previous response for EU evaluation. Further, note this suite of regression tasks is standard in the literature.
>
> **Definition of Significant Difference**: We appreciate the reviewers request for clarification on this matter. Firstly, we consider a difference in performance to be insignificant if the relative reduction in performance is small compared with relative reduction in computation time. This allows subjectivity for specific use-cases (i.e. infinite resources). We refer the reviewer to the [following table](https://anonymous.4open.science/r/ll_uq-6641/table2.png), where we provide the relative performance changes for LL-GLM vs. DNN-GLM. We see that the average relative change in performance across tasks is minuscule ($2.30$% decrease to $0.55$% increase), while computation time reduces by $91.89$%, and memory by $77.17$%. For GPT-2, we see a small relative change in performance of $2.26$%, with a large reduction in computation time of $89.8$% and reduction in memory of $70.56$%. Hence, our conclusion is that for the large gain in speed/memory, the relative change in performance is surprising. We will update the manuscript to reflect this more accurate examination of our results.
>
> **Novelty of VARROC**: It is the use of the SoftMax variance as a score that is novel.
>
> **SoftMax Variance is Ad-Hoc**: We thank the reviewer for providing these references. Works [3, 4] posit the mutual information (MI) for measuring EU for classification. However, [2] shows that MI violates several properties that a measure of EU should satisfy. Further, [4] shows that variance over the softmax probabilities is equal to MI up to leading order. We present results for VARROC-ID and VARROC-OOD using MI as the score in the [following table](https://anonymous.4open.science/r/ll_uq-6641/table3.png); as predicted, performance is very similar. Further, we emphasize that variance of softmax values is not ad-hoc; this is the epistemic uncertainty in the induced second-order Dirichlet distribution.
>
> **Last-Layer References**: See [8, 9, 10, 11, 12].
>
> **Small Epsilon**: In this figure, the noise signal is taken to be $9$, which is close to the average output-value over the space.
>
> ### Continued Discussion
> We are very appreciative of the reviewers thoughts; the amendments to our work will greatly improve the quality. We encourage further discussion, due to the limited space for the initial rebuttal.
>
> ## References
> [1] Mucsányi, Bálint, Michael Kirchhof, and Seong Joon Oh. "Benchmarking uncertainty disentanglement: Specialized uncertainties for specialized tasks." NeurIPS 2024.
>
> [2] Wimmer, Lisa, et al. "Quantifying aleatoric and epistemic uncertainty in machine learning: Are conditional entropy and mutual information appropriate measures?." PMLR, 2023.
>
> [3] de Jong, Ivo Pascal, Andreea Ioana Sburlea, and Matias Valdenegro-Toro. "Uncertainty Quantification in Machine Learning for Biosignal Applications-A Review." 2023.
>
> [4] Smith, Lewis, and Yarin Gal. "Understanding measures of uncertainty for adversarial example detection." 2018.
>
> [5] Yu, et. al. "Delving into Noisy Label Detection with Clean Data", ICML 2023.
>
> [6] Jindal, et. al. "Learning Deep Networks from Noisy Labels with Dropout Regularization" 2017.
>
> [7] Shi. et. al., ”A Theoretical Analysis on Feature Learning in Neural
> Networks: Emergence from Inputs and Advantage over Fixed Features”, ICML
> 2022
>
> [8] Calvo-Ordonez, et. al. "Richer Bayesian Last Layers with Subsampled NTK Features", 2026.
>
> [9] Fiedler, et. al. "Improved Uncertainty Quantification with Bayesian Last Layer", 2023.
>
> [10] Watson, et. al. "Latent Derivative Bayesian Last Layer Networks", AISTATS 2021.
>
> [11] Kristiadi, et. al. "Being Bayesian, Even Just a Bit, Fixes Overconfidence in ReLU Networks", ICML 2020.
>
> [12] Ortega, et. al. "Variational Linearized Laplace Approximation for Bayesian Deep Learning", ICML 2024.

---

> > ### Author Rebuttal · Reviewer_6SMn · 2026-04-01
> >
> > I will describe which concerns I still have and which have been addressed.
> >
> > - **W1 / Aleatoric vs. Epistemic:** The papers cited for considering these datasets noiseless do this lightly as an assumption, but do not provide evidence. https://labelerrors.com/ provides examples of noisy items these datasets. Note that ImageNet is (in my experience) not regarded as a noiseless dataset. I see that you aim to only model epistemic uncertainty, but find this an unusual narrowing of the scope of BNNs. BNNs for classification typically estimate both aleatoric and epistemic uncertainty (may or may not be disentangled).
> >
> > - **W2/ Definition of Significant difference:** "we consider a difference in performance to be insignificant if the relative reduction in performance is small compared with relative reduction in computation time". Significance is typically defined as statistical significance. This would also be what would be needed to support your claim in the abstract, that "last-layer approximation yields comparable UQ performance" or in conclusion that "we have found no specific regimes where
> > a last-layer approximation decreases model-fit", or in contributions "demonstrating equivalent performance across tasks.". With the proposed definition relative to the reduction in computation time, I think claims remain similar to the "fidelity-cost trade-off" previously proposed by Kristiadi.
> >
> > - **W3 / Novelty of VARROC:** resolved, but the current way it is described is probably unhelpful in the presentation. It is more intuitive to treat the variance of softmax as part of the "model" instead of as part of the "evaluation". Then evaluation remains consistent with well-known evaluations and is more easily comparable. I would also avoid calling this novel, since the variance of softmax is already well known (Smith et al.), and the evaluation methods are as-well.
> >
> > - **W4 / SoftMax Variance is Ad-Hoc**: resolved, but I would propose that softmax-variance is not immune to the limitations of MI as proposed by Wimmer et al.
> >
> > - **W5 / Last Layer references:** partially resolved. From the cited literature I see that previous works suggest that there should be some trade-off, this is therefore not a strawman argument. The work could still benefit from extensively describing how it differs from these existing work, in part because they also seem to suggest that last-layer BNNs are a large cost saving and provide good UQ results.
> >
> > - **limitations section is still missing**

---

> > > ### Author Response · Authors · 2026-04-07
> > >
> > > We appreciate the effort of the reviewer to engage in further discussion. Please see our responses to your concerns.
> > >
> > > **W1**: Though these datasets may have some aleatoric noise, the argument for use of VARROC-ID is the same as for use of ECE. Now, to sample from GLM predictive posteriors, we employ Algorithm 1., as it is currently the most scalable algorithm available (allows ImageNet, GPT-2). However, Algorithm 1. only allows sampling from the noiseless GLM predictive posterior, and so cannot model aleatoric uncertainty. Extension of Algorithm 1. to include models of noise over the data is a potential avenue for future work. We will include this restriction of the work to EU in the Limitations section (see below).
> > >
> > > **W2**: With respect to statistical tests, a two-tailed t-test on the GPT-2 results in table 2 gives a p-value of $0.1832$, hence the difference is not considered statistically significant. However, the underlying distributions are unlikely to be identical, so enough realizations could eventually result in a statistically significant difference being detected.
> > > We had chosen not to frame this as a negative result for statistical significance, since we view it as **misleading**. However, if the reviewer would prefer, we are happy to include this in the updated version of the paper, with the appropriate caveats explicitly mentioned.
> > > In our work, we consider **practical significance**, i.e. relative difference in performance compared to reduction in computation time/memory. In the previously attached tables, we observe practically insignificant differences in performance. More importantly, any detected improvement is not consistently in favor of one mode -- for certain tasks LL-GLM is **superior** to DNN-GLM. This novel result leads us to firmly reject the claim that there is a fidelity-cost trade-off.
> > > Note we will change our wording in the contributions to "demonstrating insignificant performance differences across tasks", rather than "equivalent performance", for clarity. Finally, the claim "we have found no specific regimes where a last-layer approximation decreases model-fit" is in regards to the theoretical component of the work, not the experimental results.
> > >
> > > **W3 / W4**: We agree with the reviewer on all these points, and will update our presentation of SoftMax variance, MI, EU, VARROC-ID and VARROC-OOD (with novelty claim removed). Further, we will include the experiments with both SV and MI for completeness, as well as a mention of [2,3,4] when introducing these measures of EU.
> > >
> > > **W5**: See the following proposed subsection of the Background:
> > >
> > > "Several works in the Bayesian UQ literature discuss the fidelity-cost trade-off of last-layer approximations. In [10], the authors posit that last-layer features can overfit the feature space, limiting predictive UQ ability. They propose to amend Type-II maximum likelihood overfitting with an augmented marginal likelihood. Further, [8] claims that the last-layer approximation misses "uncertainty induced by earlier layers"; a low-rank transformation between the full Jacobian and the last-layer features is proposed as a new ad-hoc set of features. Interestingly, a direct comparison of LLA (DNN-GLM) vs. last-layer LLA (LL-GLM) on a 1-$D$ toy-regression problem is provided in [12], the only direct comparison we are aware of in the literature. Not all works are pessimistic; in [9], the authors discuss the "promising compromise" of using Bayesian last-layer methods for DNNs, while [11] shows that a last-layer approximation "already gives desirable benefits". However, our work is the first to explicitly compare a last-layer approximation (LL-GLM) with the full-linearization in a large empirical evaluation, across a range of tasks."
> > >
> > > **Limitations & Future Works**: Our proposed **Limitations & Future Works** section is as follows:
> > >
> > > "Firstly, the conditions in Assumption 1. prevent direct connection to theoretical results for trained DNNs on realistic datasets. While there is evidence that the theory-practice gap may be small in certain regimes (see Remark), extending Theorem 1 & 2 to trained DNNs under practical assumptions, for example through the spectral distributions derived in [14], is an exciting, yet challenging, avenue for future work. Further, the sampling technique of Algorithm 1. only allows sampling from the posterior of a noiseless GLM, and hence in our work we only model the **epistemic** uncertainty of a DNN. Extending Algorithm 1. to include noise models on the data (allowing for **aleatoric** uncertainty) would enable a more complete comparison of the DNN-GLM vs. LL-GLM for UQ. Finally, use of MCMC techniques to sample from the last-layer posterior may bring further performance and computational improvements."
> > >
> > > ## References:
> > > [13] Wei. et. al., "More Than a Toy: Random Matrix Models Predict How Real-World Neural Representations Generalize", ICML 2022.
> > >
> > > [14] Hogkinson, et. al. "Models of Heavy-Tailed Mechanistic Universality" ICML 2025

---

### Official Review · Reviewer_8JUm · 2026-03-03

**Soundness:** 2
**Presentation:** 3
**Significance:** 3
**Originality:** 2
**Overall Recommendation:** 5
**Confidence:** 3

**Summary:**

The authors investigate whether full-network linearization provides superior epistemic uncertainty quantification compared to last-layer linearization in the context of Bayesian Deep Learning. This linearization effectively turns the networks into Bayesian GLMs. Theoretically, they employ Random Matrix Theory to compute the limiting BFE for both kernels at initialization. Further they propose a sampling framework called LinearSampling (Alg 1) to apply the ideas to large scale linearized networks. They provide a pytorch package and test their method on a range of different BDL tasks claiming parity in performance and gains in computational efficiency. They hypothesize that epistemic UQ relies more heavily on how features are used in the final layer rather than how they are formed throughout the network.

**Compliance With Llm Reviewing Policy:**

Affirmed.

**Final Justification:**

I am raising my score to accept as the authors have addressed my primary concerns. My recommendation hinges on the authors delivering on their promise to transparently integrate all the discussed limitations into the general framing **&** dedicated limitations section. The SMS-UBU baseline configuration is imo suboptimal and the associated implementation, which is provided, appears to require a more rigorous audit to ensure correctness. Nevertheless, the paper offers a high-utility contribution by challenging the necessity of full-network linearization/inference. Although the results feel at times oversold given the nuances of trained networks (please address this in the CRC - see above), the practical findings provide clear value to the Bayesian Deep Learning community.

**Key Questions For Authors:**

* [Q1] Please adress [W1]
* [Q2] Regarding [W3] and [W4], can you include evaluations using classical scoring rules (like LPPD) and provide standard metrics alongside your custom scores for all experiments? Please also justify the use of VARROC-OOD in light of the arguments presented in [1].
* [Q3] Regarding [W4], could you augment at least the setting in Table 1 and relevant classification experiments with more flexible posterior approximations? Baselines such as Bayesian Deep Ensembles, or SOTA MCMC on the last layer would provide a much stronger test of your central & concluding hypothesis about epistemic UQ capture.
* [Q4] Please mathematically clarify exactly what distribution Algorithm 1 is sampling from and how it differs from previous techniques, see [W2].
* [Q5] Can you explain the source of the massive memory footprint (Table 2) for the 124M parameter model?

**Limitations:**

The authors do not sufficiently acknowledge the theoretical and empirical limitations of their work. A dedicated limitations section should be added. This section should explicitly discuss the constraints of Assumption(s) 1, the disconnect between asymptotic BFE and finite-width predictive performance.

**Strengths And Weaknesses:**

## Strengths

* **[S1]** The paper addresses the relevant bottleneck of high computational cost in many approximate Bayesian Deep Learning methods. The demonstration that LL-GLMs match DNN-GLMs on sizable models provides strong practical utility in particular in combination with the provided package enhancing the reproducibility and the practical impact of the work.
* **[S2]** The theoretical derivations in the appendix backing the main theoretical claims of the paper seem to be mathematically correct and complete.
* **[S3]** The manuscript is generally very well-written, easy to follow, and visually clear.

## Weaknesses

* **[W1]** The theoretical developments strictly rely on assumption(s) 1 and seem to be restricted to fully connected feedforward neural networks with scalar output. Thus it is highly questionable how much explanatory power the theoretical framing really provides for trained realistic nets which are explored in the empirical section of the paper.

* **[W2]** Algorithm 1 is presented as a novel sampling framework. However, it is a straight forward application of well-established optimize-then-sample techniques widely used in the BDL literature. The algorithmic novelty is therefore rather on the marginal side. Also the work has substantial conceptual overlaps with prior works like Wilson et al. (2025) and Hodgkinson et al. (2023) limiting conceptual originality.

* **[W3]** The authors introduce VARROC-ID/OOD, in major parts dismissing standard metrics like NLL/ELPD and AUROC for classification. This is, in my opinion, a violation of proper evaluation. LPPD is the canonical proper scoring rule for probabilistic forecasts (see Gelman et al., 2014). Furthermore, evaluating OOD detection for Bayesian methods and then mixing results with epistemic UQ in the conclusion is highly problematic as the objectives are fundamentally misaligned (cf. [1]). Furthermore, the lack of standard metrics (LPPD, AUROC, ECE, Brier Score) makes it difficult to contextualize these results against the broader literature.

* **[W4]** The authors claim LL-GLM is sufficient for UQ, but they partly only compare it to DNN-GLM (table 1) or miss out relevant SOTA baselines. Because both are restrictive approximations, it is unclear if they are equally good or equally bad. The evaluation crucially lacks a comparison to flexible full-posterior approximations (e.g., warm-started Bayesian Deep Ensembles such as MILE [2] for the setting table 1 or SMS-UBU [3] for the setting in Fig 6.) to establish a quality baseline. Especially for the last layer setting one could easily build upon the subnetwork inference literature (see e.g. [4]) and deploy high fidelity MCMC in the reduced subspace for a better contextualization of performance and calibration.

* **[W5]** The memory requirements reported for DNN-GLM on GPT-2 finetuning (>70 GB) seem highly excessive. A 124M parameter model is by modern standards relatively small and standard full-parameter fine-tuning of 8B parameter models requires similar or even less memory on an H100. This suggests the DNN-GLM implementation could include inefficiencies, which artificially inflate the relative efficiency claims of LL-GLM. If no reductions, e.g., through Jacobian vector products, can be achieved, also the memory footprint of the LL-GLM method does not look too appealing.

## References

* [1] Li, Y. L., Lu, D., Kirichenko, P., Qiu, S., Rudner, T. G., Bruss, C. B., & Wilson, A. G. (2025). Out-of-Distribution Detection Methods Answer the Wrong Questions. ICML.
* [2] Sommer, E., Robnik, J., Nozadze, G., Seljak, U., & Rügamer, D. (2025). Microcanonical Langevin ensembles: Advancing the sampling of Bayesian neural networks. ICLR.
* [3] Paulin, D., Whalley, P. A., Chada, N. K., & Leimkuhler, B. (2025). Sampling from Bayesian neural network posteriors with symmetric minibatch splitting Langevin dynamics. AISTATS.
* [4] Daxberger, E., Nalisnick, E., Allingham, J. U., Antorán, J., & Hernández-Lobato, J. M. (2021). Bayesian deep learning via subnetwork inference. ICML.

---

> ### Author Rebuttal · Authors · 2026-03-31
>
> We thank the reviewer for their kind comments on the "strong practical utility" of the experimental section and "the provided package", and that they found the paper "generally very well-written, easy to follow, and visually clear."
>
> **Explanatory Power of Theory**: We refer the reviewer to our response to a similar question from Reviewer aCEK, where we outline extensions to trained networks and the significance of our theoretical results. To explicitly extrapolate our findings to trained neural networks, we provided a thorough experimental section, where we find that the theoretical conclusions still hold. Proving theoretical claims for trained neural networks is an interesting, yet challenging, avenue for future work.
>
> **Algorithmic Novelty \& Similarity to [3, 4]**: We agree that the novelty of Algorithm 1. and Lemma 2. is marginal, as it is simply an application of the work of [3] to the last-layer and classification setting. Furthermore, we agree that our work is a continuation and extension of the work of [3, 4], though we view this as a positive.
>
> **Violation of Proper Evaluation**: The suggestion to employ the LPPD metric is highly appreciated. We provide the LPPD score for all methods and models/datasets for image classification in the [following Table 1](https://anonymous.4open.science/r/ll_uq-6641/table1.png). We see that the LPPD score reinforces the conclusion of the paper. In regards to the inclusion of AUCROC, ECE, Brier Score, we respectfully disagree and kindly refer the reviewer to [3, Appendix C], for an in-depth discussion of why these metrics are inappropriate in this context.
>
> **EU for OOD Detection**: We agree with the reviewer, and [1], that for OOD detection, Bayesian predictive variance has no guarantees. Instead, we test the use of EU for OOD detection experimentally; we see that for both GLMs, EU is quite good at this task. Note that the key argument in [1] against EU for OOD detection is that as $n \to \infty$ (and number of parameters $p$ remain fixed), the posterior collapses to the MAP, and hence the predictive variance collapses to zero. However, this scaling case is not true for modern deep learning [5] ($p / n \to c \in (0, \infty)$). This could be a reason why predictive variance seems to perform well at OOD detection for the networks we employ.
>
> **Missing SOTA Baselines**: We kindly refer the reviewer to [3], where for both regression and classification tasks, DNN-GLM has been compared against a large array of competitive Bayesian methods (e.g. DE, SWAG, SNGP, SGLD, Bayesian Deep Ensembles, VaLLA etc.). In our image classification results, we provided comparison with the Bayesian methods we view to be the most compared to and well-known in the literature. We also provide a comparison with SMS-UBU [2] in the [amended image classification results](https://anonymous.4open.science/r/ll_uq-6641/table1.png). In relation to employing MCMC in the last-layer subspace, note that LL-GLM samples exactly from this posterior, and so employing MCMC is not needed.
>
> **Memory Cost**: The memory cost for LL-GLM is roughly proportional to the cost of a forward pass of the underlying model, and for the DNN-GLM is roughly $3 - 5 \times$ this. Hence, the large memory overhead in Table 2 arises from a naive implementation of GPT-2 (*float32*, context length $1024$). Improving the efficiency of this forward pass will reduce the memory footprint of both GLMs. We have also included the memory cost for the [image classification results](https://anonymous.4open.science/r/ll_uq-6641/table1.png). We will include mention of this in the manuscript.
>
> **Asymptotic vs. Finite-Width**: Please see Figure 7 in the Appendix of our work, where we observe excellent adherence of the limiting theoretical BFE with the mean BFE of small NNs.
>
> ### Continued Discussion
> We value the in-depth critique of our work by the review, and hope that we have sufficiently answered their concerns. The inclusion of LPPD and SMS-UBU will strengthen our findings significantly, and we will add an explicit limitations section. We encourage further discussion of our rebuttal points.
>
> ## References
> [1] Li, Y. L., Lu, D., Kirichenko, P., Qiu, S., Rudner, T. G., Bruss, C. B., & Wilson, A. G. (2025). Out-of-Distribution Detection Methods Answer the Wrong Questions. ICML.
>
> [2] Paulin, D., Whalley, P. A., Chada, N. K., & Leimkuhler, B. (2025). Sampling from Bayesian neural network posteriors with symmetric minibatch splitting Langevin dynamics. AISTATS.
>
> [3] Wilson, et. al. "Uncertainty Quantification with the Empirical Neural Tangent Kernel", NeurIPS 2025.
>
> [4] Hodgkinson, et. al. "Monotonicity and Double Descent in Uncertainty Estimation with Gaussian Processes", ICLR 2023.
>
> [5] Couillet, et. al. "Random Matrix Methods for Machine Learning", 2023.

---

> > ### Author Rebuttal · Reviewer_8JUm · 2026-04-02
> >
> > Thank you for the careful rebuttal.
> >
> > I appreciate the candid acknowledgement regarding W2 and consider W3 to be resolved with the inclusion of LPPD even though I am a little skeptical as you report ECE for regression (which can also be questioned) but then dismiss it for classification.
> >
> > In the following my follow-up questions:
> >
> > 1. It is not clear how SMS-UBU was configured. Was an ensemble of chains used, how many samples were drawn and was full or subspace inference performed?
> >
> > 2. The rebuttal states that "LL-GLM samples exactly from this posterior, and so employing MCMC is not needed." However if I understood the paper correctly LL-GLM is only approximate in this practical setting. Could you clarify in what sens this "exact" was meant?
> >
> > 3. I find the explainations regarding W1 (also the ones presented to aCEK) too high level and not too convincing. In the paper you state that epistemic UQ depends on how the features are "used" this seems to be at odds with the infinite limit arguments you present (also to aCEK and me now) where no feature learning happens anymore. Could you sketch more explicitly how you plan on adressing the in my view large gap between your theory and presented experiments? Note that I also agree with reviewer 6SMn that the results for the experiments mirror the known "fidelity-cost trade-off" previously discussed by Kristiadi et al. which makes it even harder to connect the results with your theoretical findings (again W1 and touching upon my original concerns regarding novelty/W2).
> >
> > I maintain my score for now but am open to adjusting based on the authors' responses.

---

> > > ### Author Response · Authors · 2026-04-07
> > >
> > > We are pleased that the reviewer is engaging in ongoing discussion; we hope our response answers their concerns.
> > >
> > > **SMS-UBU**: Firstly, only a single chain was run. Running an ensemble of posteriors will always improve performance, and computational cost, for multi-modal posteriors (see [6, Figure 5.]). We first look to compare the quality of the base methods first, and thus use a single chain to allow fair comparison with the GLMs. Secondly, we use the full-subspace for SMS-UBU. Finally, we have found errors in our implementation in our script, leading to incorrect results for SMS-UBU. Apologies! We have re-run our experiments, and have updated our results (see here). We use the hyperparameters in the second paragraph of Section 5, and those in Table 3 -- also note that we take our pre-trained DNN, run SWA (using Adam with weight decay equal to that for our original DNN training) for $5$ epochs, and then run SMS-UBU from the averaged parameters. We take $40$ posterior samples, and discard the first $10$ as a burn-in. The code repository has been updated with the SMS-UBU implementation, and the updated results can be found [here](https://anonymous.4open.science/r/ll_uq-6641/table1.png).
> > >
> > > **Exact Sampling**: If the last-layer Jacobian is rank-deficient, and the optimization scheme (GD/SGD) in Algorithm 1. is run for epochs $\to \infty$, then sampling is exact. However you are correct; in practical settings, the LL Jacobian may have many small eigenvalues that do not contribute meaningfully to prediction. For finite steps of GD/SGD in Algorithm 1., the trained parameters will not converge in these directions, and so the resultant samples will be from a GP using the CK, where singular values near zero have been truncated. It is unclear how these small singular values contribute to UQ. Hence, employing MCMC to sample from the non-truncated posterior may have some advantage. Though interesting, we leave this as a future direction for investigation (see our **Limitations & Future Work** section in response to **Reviewer 6SMn**).
> > >
> > > **Theory-Practice Gap**: Please see our final response to a similar query from **Reviewer aCEK**, where we highlight areas where the gap may be small. Further, see our **Limitations & Future Work** section, where we discuss this theory-practice gap.
> > >  In regards your concern on the use of the infinite-width argument, we first clarify that epistemic UQ for the DNN-GLM arises from uncertainty both in the features, and around how the features are used, as by an application of the chain rule, the NTK is equivalent to a linear combination of the CK at each layer. In the infinite-limit, feature learning does not occur; rather features stay random. Hence, in this regime, the NTK and the CK become equivalent for UQ (as measured by BFE), as they both become random feature (RF) models, and as we have shown the added depth of the NTK does not give any advantage for such RF models. Hence, the argument holds.
> > > Further, please see our second response to **reviewer 6SMn**, where we show that our results reject the fidelity-cost trade-off argument, as one mode is not consistently superior to the other in performance (which would be required for such a trade-off).
> > > We also emphasize that our work is **novel**:
> > >  - We have provided the first theoretical comparison of these GLMs, arising from a non-trivial extension of [4] to kernels derived from DNNs [6].
> > >  - We also provide the first direct experimental comparison of these GLMs, on large scale image classification and language modelling tasks.
> > >
> > > ### References
> > > [6] Wang, et. al. "Spectra of the Conjugate Kernel and Neural Tangent Kernel for Linear-Width Neural Networks", NeurIPS 2020.

---

### Official Review · Reviewer_vwRy · 2026-03-10

**Soundness:** 3
**Presentation:** 3
**Significance:** 3
**Originality:** 3
**Overall Recommendation:** 5
**Confidence:** 2

**Summary:**

This is a paper about Bayesian uncertainty quantification in NNs.  The authors consider an earlier linearisation approach and show how focusing only on the parameters of the last NN layer gives competitive results if compared against a linearisation of all the parameters, and this corresponds to a considerably faster quantification of the uncertainty.  This is proved by extensive experiments, but the paper contains also a technical part where the claim is supported by a theoretica analysis based on an equivalence with Gaussian processes whose Bayes free energies are considered.

**Compliance With Llm Reviewing Policy:**

Affirmed.

**Final Justification:**

In spite of my low confidence, after reading the other reviews and all the rebuttals, I am happy to confirm my positive opinion, and I think the paper can be safely accepted.

**Key Questions For Authors:**

With the only exception of the LLM experiments, the authors do not report information about execution times. Is there a reason for that?
Time can be a key descriptor to appreciate the advantages of the last-layer approach.

I dont' understand the relaxation of the i.i.d. assumption in S4.

Here the focus is purely on Bayesian UQ methods. What would have been the result of a comparison against some non-Bayesian approach either in term of performance and, most importantly, speed?

**Limitations:**

I don't see any risk of negative societal impact.
Regarding the limitations, the points I already raised are mostly a limited empirical analysis of the advantages of the last-layer approximation in terms of computational savings, and the impact of these results with respect to other, non-Bayesian, methods.

**Strengths And Weaknesses:**

Regarding soundness, I am not a specialist of GPs/BFEe and I didn't check the technical material in the appendixes. Yet, the claims of the authors seem to be quite reasonable and supported by examples (and later by experiments). The design of the experiments is in fact very reasonable in order to check the main claim of the paper and, in spite of the different benchmarks, the results are clearly supporting the point of the authors.

The paper is quite technical, as a non-expert, I might have missed some point, but I have found the presentation of both the theoretical part and the experiments sufficiently clear. This is also true with respect to the existing works.

The result seems to be quite significant and it might have an impact for the Bayesian UQ and not only. I am not sure this might have an impact outside the Bayesian UQ community, and whether some of these ideas might trigger similar results on non-Bayesian methods.

The work seems to be quite original, even if a dedicated section about the related works is not present.

These points motivated my (quite) positive recommentation. Of course, my low confidence should be taken into consideration.

---

> ### Author Rebuttal · Authors · 2026-03-31
>
> The reviewers comments on the 'reasonable' claims and experimental designs are appreciated. We are glad that they found the presentation 'sufficiently clear', and that they see the results as 'quite significant'.
>
> **Execution times**: We have now amended the results to include execution times for the image classification results (see [linked table](https://anonymous.4open.science/r/ll_uq-6641/table1.png)). We see that there is often a significant reduction in computation time for the LL-GLM vs. DNN-GLM. For smaller models, the computation times are dominated by PyTorch processes (e.g. data-loading), rather than algorithmic computations. However for the large ResNet50 models, we see that LL-GLM is often an order of magnitude faster, and two orders-of-magnitude faster than DE.
>
> **Relaxation of I.I.D in S4**: This is an error; here we meant that D no longer has to be a Gaussian distribution, and X and Y no longer have to be independent. We still assume that all inputs and outputs are i.i.d., as is common in machine learning, but we allow any distribution, and do not require that $p(x,y) = p(x)p(y)$. We will amend the text, and appreciate the reviewer highlighting this error.
>
> **Non-Bayesian Comparison**: A significant Frequentist method in the literature is the procedural-noise correcting (PNC) method [1], based on the infinite-width NTK results [2]. In [3], the DNN-GLM was compared with PNC in regression settings. The other important categories of non-Bayesian UQ methods are conformal predictions and distance-aware methods. We however restricted our attention in this paper to only Bayesian methods, due to the enticing frameworks of the predictive and posterior distributions. Comparison against such other methods is interesting, but challenging in our setting, as our experimental metrics (LPPD, VARROC) are based off of Bayesian variance values, and hence we leave this comparison for future work.
>
> ### Continued Discussion
> We will amend our document to reflect the correct loosening of assumptions, and will amend the evaluation to include the given execution times. Further, we will include the reference to the PNC method. If the reviewer has any further queries, we are hopeful that they will engage in further discussion.
>
> # References
> [1] Huang, et. al., "Efficient Uncertainty Quantification and Reduction for Over-Parameterized Neural Networks", NeurIPS 2023.
>
> [2] Jacot. et. al., "Neural tangent kernel: Convergence and generalization in neural networks", NeurIPS 2018.
>
> [3] Wilson, et. al. "Uncertainty Quantification with the Empirical Neural Tangent Kernel", NeurIPS 2025.

---

> > ### Author Rebuttal · Reviewer_vwRy · 2026-04-02
> >
> > I thank the authors for their clear rebuttal and happy to confirm my positive recommendation.

---

> > > ### Author Response · Authors · 2026-04-07
> > >
> > > Thank you!

---

### Official Review · Reviewer_aCEK · 2026-03-13

**Soundness:** 3
**Presentation:** 4
**Significance:** 4
**Originality:** 3
**Overall Recommendation:** 5
**Confidence:** 3

**Summary:**

The paper studies Bayesian uncertainty quantification for deep neural networks through two local linear approximations: a last-layer generalized linear model (LL-GLM) and a full-network linearized model (DNN-GLM). Viewing these linearized Bayesian models through their equivalent Gaussian process representation, the authors show that LL-GLM corresponds to the conjugate kernel (CK), while DNN-GLM corresponds to the neural tangent kernel (NTK). They then compare the two both theoretically, via the Bayes Free Energy (negative log marginal likelihood) across regression, classification, and language modeling tasks. The main theoretical result is that NTK does not have a broad inherent advantage over CK. Both behave similarly in many asymptotic regimes, and the clearest theoretical benefit for NTK appears only in a highly oversampled setting at random initialization.

**Compliance With Llm Reviewing Policy:**

Affirmed.

**Key Questions For Authors:**

1. A central question is how strongly the main conclusion depends on the fact that the theory is carried out at random initialization. The paper gives empirical evidence that the oversampled-regime gap can disappear after training, but this is also where the theory-to-practice bridge is weakest. Could the authors discuss more explicitly what theoretical properties they expect to survive training and which they believe are initialization-specific?

2. In this work, BFE is interpreted as a notion of robustness at the kernel level and also introduces VARROC-ID/OOD for empirical UQ evaluation. Could the authors clarify how they see the relationship between these notions and more standard robustness, and provide further evidence that the main LL-GLM vs DNN-GLM conclusions are not sensitive to this particular evaluation framing?

3. The paper presents strong empirical evidence that LL-GLM often matches DNN-GLM at substantially lower cost. Could the authors comment more explicitly on the settings where they would still expect full-network linearization to matter in practice, beyond the oversampled regime highlighted by the theory?

4. The theoretical analysis is developed for scalar-output networks. Could the authors comment on how strongly the main CK vs NTK conclusions are expected to depend on this restriction, and whether they anticipate similar behavior in multi-class settings?

**Limitations:**

The paper would benefit from a short, explicit limitations discussion covering the gap between initialization-based theory and trained-network practice, the narrow meaning of robustness used in the paper, and the dependence of some empirical conclusions on newly introduced evaluation metrics.

**Strengths And Weaknesses:**

The paper appears technically sound overall. The central comparison is well motivated, and the chain from linearized Bayesian GLMs to equivalent GP posteriors induced by CK and NTK is coherent and clearly central to the paper’s argument. The theoretical section is internally consistent. The authors derive limiting BFE expressions for randomly initialized CK and NTK under proportional growth of width and sample size, then use those limits to discuss robustness/descent behavior and the oversampled regime. The experimental section is also fairly broad, covering regression, image classification, and language modeling.

The strongest theory is at random initialization, not for trained networks. The paper is upfront about this and provides arguments for why initialization can still be informative, but this still weakens the directness of the theoretical support for the main practical conclusion. Second, the use of BFE as a measure of robustness is less convincing than its use as a measure of Bayesian model quality. The robustness terminology feels broader than what is actually established.

The paper is well written and well organized overall. Its main line of argument is coherent, but some of the core concepts are presented in a way that may be harder to access than necessary. In particular, the discussions of CK versus NTK, the equivalence between the GLM and GP views, and the interpretation of robustness through BFE could benefit from more intuitive presentation.

The significance is mainly a contribution to Bayesian UQ and linearized neural uncertainty approximations. But within that area, it is quite relevant. Its originality comes from a strong synthesis. It connects LL-GLM vs DNN-GLM to CK vs NTK through GP equivalence, analyzes their relative quality using limiting BFE, identifies a specific regime where NTK has a theoretical advantage, and then empirically argues that this advantage may be an initialization artifact. That combination of perspectives feels original and useful. The large-scale empirical comparison and the release of a lightweight sampling package also add value beyond the pure theory.

---

> ### Author Rebuttal · Authors · 2026-03-31
>
> We would like to thank the reviewer for their positive comments, specifically in regards to the 'technically sound' content, and the 'fairly broad' experimental section. We appreciate that the reviewer thought the work was "well written and well organized overall", and are glad that they see it as "quite relevant" and "original" in the Bayesian UQ area.
>
> **Theoretical Properties after Training**: We acknowledge that the assumptions for our analysis hinder the extrapolation of our findings to some trained neural networks. However, we can still provide comment on this. Firstly, the NTK theory of [1] means that findings will hold for trained neural networks as $d_l \to \infty$. Secondly, there has been some success at predicting generalization ability for trained neural networks using Marchenko-Pastur (MP) like structures [2]; our models also exhibit MP-type distributions. Further, a motivation for our work was [3], which suggested that there is an inherent deficiency in the initialized CK structure that prevented it from performing as well as the initialized NTK, as measured by robustness. By re-framing robustness in terms of BFE, we were able to show that there is not necessarily a degeneracy in the structure of the CK that could lead to poor performance. Explicit extrapolation of our theoretical findings to trained kernel matrices is a potential avenue for further work.
>
> **Relationship between BFE, VARROC, Robustness**: Robustness of a function is often defined as an imperviousness of the function to input manipulation; standard measurements are usually related to Lipschitz constant. We are unable to conclude the robustness of the mean predictive function from the BFE, however choose to delegate the task of robustness to the predictive variance, which acts as a detector for points for which the posterior functions are now 'uncertain' about. Specifically, as BFE decreases, each posterior function provides a better distributional fit to the training data. Accordingly, each posterior function will provide a worse fit for inputs from disparate distributions; the variance over such posterior functions will then be high for distributionally-shifted inputs. This will act as our detector, and allow our predictive-distribution to be more 'robust'; this is what VARROC-ID and VARROC-OOD are measuring.
>
> **Sensitivity to Evaluation Framing**: We have added further comparison of LL-GLM vs. DNN-GLM through the log pointwise predictive density (LPPD) metric ([see Table 1](https://anonymous.4open.science/r/ll_uq-6641/table1.png)). This measures the alignment of the predictive variance values to the error in the predictive mean. We see that the conclusions of the work also hold under this new metric.
>
> **When Full-Linearization Matters**: From the attached image classification results, we see that for smaller models (LeNet5, ResNet9), where feature learning is less-powerful, modelling the uncertainty in the features results in slightly better performance for the DNN-GLM. This relative improvement in performance reduces as model size grows (e.g. ResNet50).
>
> **Multi-Class Setting**: For outputs that are independent (as is assumed for Laplace Approximations for classification [5, 6]), the multi-class BFE will simply reduce to the BFE multiplied by the number of classes. Hence, the theory extends to this multi-class case. This can be seen in Figure 2, where we consider the multi-output datasets MNIST and CIFAR-10 (after whitening). We will add this note on the extension to multi-class settings to the theoretical discussion. For outputs with anisotropic covariance structures, the theory becomes much more complex, and requires the machinery of Free Probability [7]. Unfortunately, this is outside the scope of the current work.
>
> ## Continued Discussion
> We hope that our responses have provided answers to your critiques and queries. We will update our manuscript to include the new LPPD results, add a discussion of when full-linearization matters, and revise the explanation of the connection between BFE, robustness and VARROC. Further, we will add an explicit limitations section.
>
> ## References
> [1] Jacot. et. al., "Neural tangent kernel: Convergence and generalization in neural networks", NeurIPS 2018.
>
> [2] Wei. et. al., "More Than a Toy: Random Matrix Models Predict How Real-World Neural Representations Generalize", ICML 2022.
>
> [3] Bombari. et. al. "Beyond the Universal Law of Robustness: Sharper Laws for Random Features and Neural Tangent Kernels", ICML 2023.
>
> [4] Shi. et. al., "A Theoretical Analysis on Feature Learning in Neural Networks: Emergence from Inputs and Advantage over Fixed Features", ICML 2022.
>
> [5] Rasmussen et. al. "Gaussian Processes for Machine Learning", 2006.
>
> [6] Immer. et. al. "Improving predictions of Bayesian neural nets via local linearization", AISTATS 2021.
>
> [7] Anderson, et. al. "An Introduction to Random Matrices", 2005.

---

> > ### Author Rebuttal · Reviewer_aCEK · 2026-04-02
> >
> > Thank you for the thoughtful rebuttal. I appreciate the added clarification on the relationship between BFE, robustness, and VARROC, and the planned additions of LPPD results.
> >
> > The rebuttal addresses several of my questions usefully, especially regarding evaluation framing and the multi-class setting. The added LPPD comparison is helpful, and the discussion of smaller versus larger models makes the practical takeaway clearer.
> >
> > My concern regarding random initialization remains only partially resolved. The theory is still developed at random initialization, while the practical claims concern trained networks. The rebuttal gives plausible reasons why initialization-based analysis may still be informative, but it does not clearly identify which of the paper’s main conclusions are expected to persist after training and which may be initialization-specific. In that sense, the theory-to-practice gap is narrowed, but not fully resolved.
> >
> > I also still think the paper should use the term robustness carefully and avoid overstating what is established. The evidence appears strongest for Bayesian model quality and uncertainty-based detection of distributional shift, rather than robustness in the standard predictive-function sense.

---

> > > ### Author Response · Authors · 2026-04-07
> > >
> > > We appreciate the reviewers continued engagement in our work.
> > >
> > > **Theory-Practice Gap**:
> > > Firstly, we acknowledge that the theory-practice gap is not fully closed. However, we are able to provide specific cases where the gap may not be as large as one may think.
> > >  - By [8, Corollary 5.3], for a 2-layer DNN undergoing gradient descent for a student-teacher dataset, with a sufficiently small step-size (with similar assumptions to ours), the kernel matrices $K_{X, t}^{\text{CK}}, K_{X, t}^{\text{NTK}}$ at gradient step $t$ converge to $K_{X, 0}^{\text{CK}}, K_{X, 0}^{\text{NTK}}$ (i.e. kernels at initialization) under the double-asymptotic regime we consider in our work. We can then consider the BFE for the trained kernel matrices on a down-stream task, i.e. the data setting we currently consider in Assumption 1. In this setting, as the spectral distributions after training are equivalent to those at initialization in the limiting regime, almost surely, our theoretical results would remain unchanged.
> > >  - This theory does not cover SGD/Adam. However, we can extend the result in Figure 4 to UCI datasets, trained with Adam for $1500$ epochs with a learning rate of $0.01$, with $\lambda = 1, \tau = 0.01$. We see in the [following image](https://anonymous.4open.science/r/ll_uq-6641/tabular_mean_free_energy.png) that for each dataset, the free energy for the CK eventually decreases *below* the free energy for the NTK, during training.
> > >  - Finally, for Theorem 2, note that for both MNIST and FMNIST, the VARROC scores for both GLMs improve as we move from the smaller LeNet to the larger ResNet9, indicating BFE descent.
> > >
> > > Hence, in specific regimes, the gap between our initialization theory and that for practical networks may not be as large as we think. We will add a discussion of these points, as well as the reference to [2] and the infinite-width regime, to a formal Remark in the work. Further, in our proposed **Limitations & Future Work** section, given in the final response to **Reviewer 6SMn**, we acknowledge this theory-practice gap, and state that extending our theoretical results to practical trained networks is a potential avenue for future work.
> > >
> > > **Robustness**: We would like to emphasize that we are only considering the robustness of a kernel function through its induced UQ ability -- we do not measure robustness of the mean predictor of the GLM/GP (as is done in [9])., nor equate our definition of robustness to the previous Lipschitz definitions. This can be seen in the boxed text and proceeding paragraph in Section 2.4. However, we concede that the comparison with [9, 10] in Section 3.1 is less clear due to these differing definitions. We propose the following text:
> > >
> > > "... both the CK and the NTK approach the lower-bound on the BFE for appropriate regularization. For the CK, this result differs from the result given in Bombari et al. (2023), i.e. CK models can never be robust. This is due to the differing definition of robustness, that is, robustness of the kernel function as measured by the induced UQ ability (corresponding to BFE), rather than robustness of the mean predictor. Further, as training points grow large proportionally to width, the NTK can also approach the lower-bound on BFE for appropriate regularization. Hence, equipped with this new view of robustness, we diverge from the “universal law of  robustness” for mean predictors given in Bubeck & Sellke (2021): kernel functions do not need to arise from over-parameterized DNNs to be robust."
> > >
> > > ** References:
> > >
> > > [8] Wang, et. al. "Spectral Evolution and Invariance in Linear-width Neural Networks", NeurIPS 2023.
> > >
> > > [9] Bombari, et. al. "Beyond the universal law of robustness: Sharper laws for random features and neural tangent kernels." PMLR 2023.
> > >
> > > [10] Bubeck, et. al. "A universal law of robustness via isoperimetry", NeurIPS 2021.

---

### Decision · Program_Chairs · 2026-04-30

**Decision:**

Accept (regular)

**Comment:**

This paper studies local linearization techniques applied to the last layer of Bayesian neural networks (BNNs), in the spirit of part of the literature advocating for last-layer inference. The paper contains a good mix of methodological development and theory that I believe would be interesting for the BNN community. The paper has received some positive feedback from the reviewers; however, the reviewers also point to some limitations in the experiments and in the theory-to-practice gap (effect of the initialization). The rebuttal was mostly convincing and the discussion phase confirmed that the reviewers would be happy to see this paper accepted, provided that the authors make a good effort in strengthening the experimental validation according to what they promise in their rebuttal and clearly point to the limitations raised by the reviewers.